# Multiple Descents in Unsupervised Auto-encoders: The Role of Noise, Domain Shift and Anomalies

## Abstract

The phenomenon of *double descent* has recently gained attention in supervised learning. It challenges the conventional wisdom of the bias-variance trade-off by showcasing a surprising behavior. As the complexity of the model increases, the test error initially decreases until reaching a certain point where the model starts to overfit the train set, causing the test error to rise. However, deviating from classical theory, the error exhibits another decline when exceeding a certain degree of over-parameterization. We study the presence of double descent in unsupervised learning, an area that has received little attention and is not yet fully understood. We conduct extensive experiments using under-complete auto-encoders (AEs) for various applications, such as dealing with noisy data, domain shifts, and anomalies. We use synthetic and real data and identify model-wise, epoch-wise, and sample-wise double descent for all the aforementioned applications. Finally, we assessed the usability of the AEs for detecting anomalies and mitigating the domain shift between datasets. Our findings indicate that over-parameterized models can improve performance not only in terms of reconstruction, but also in enhancing capabilities for the downstream task.

## 1 Introduction

In recent years, there has been a surge in the use of extremely large models for both supervised and unsupervised tasks. This trend is driven by a desire to solve challenging machine-learning tasks. However, this pursuit contradicts the well-known bias-variance trade-off, which suggests that larger models tend to overfit the training data and perform poorly on the test set (Hastie et al., 2009). Despite this, many over-parameterized models have been able to generalize well (Krizhevsky et al., 2012; He et al., 2016). This challenges common assumptions regarding the generalization capabilities of models (Zhang et al., 2021; Advani et al., 2020; Neyshabur et al., 2018), as over-parameterized models often exhibit significantly superior performance compared to smaller models, even when interpolating the training data (Belkin et al., 2019b; 2018).

Recently, Belkin et al. (2019a) conducted a study on the bias-variance trade-off for large, complex deep neural network models. They discovered an interesting phenomenon called double descent. Initially, as the complexity of the model increases, the test error decreases. Specifically, as the complexity continues to increase, the variance term starts to dominate the test loss, resulting in an increase, which is known as the classical bias-variance trade-off. However, at a certain point, termed the "interpolation threshold" (Nakkiran et al., 2021), the test loss stops increasing and begins to decline again in the over-parameterized regime, yielding a curve with two decent regimes.

The phenomenon of double descent has been observed in many frameworks in supervised learning (see a survey in (Dar et al., 2021)). Model-wise double descent was demonstrated in (Spigler et al., 2018), while (Nakkiran et al., 2021; Gamba et al., 2022; Hastie et al., 2022) explore the impact of label noise and Signal-to-Noise ratio (SNR) on the double descent curve respectively. Nakkiran et al. (2021); Gamba et al. (2022) also demonstrated the phenomenon to epoch-wise and sample-wise double descent. Multiple descents were discussed in (Adlam & Pennington, 2020; Liang et al., 2020; Chen et al., 2021), and d'Ascoli et al. (2020) revealed that the interpolation threshold depends on the linearity and non-linearity of the model. However, the existence of double descent in core tasks in unsupervised learning is not yet fully understood.

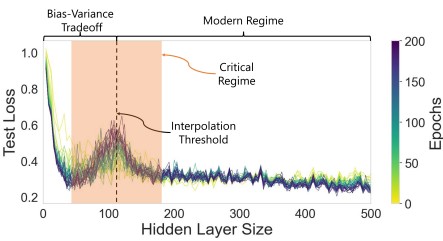

Figure 1: Demonstration of the double descent phenomenon in unsupervised learning. We present the test loss for varying epochs and hidden layer sizes for an under-complete AE.

In this study, we analyze the double descent phenomenon and its implications for crucial unsupervised tasks such as domain adaptation, anomaly detection, and robustness to noisy data, utilizing under-complete AEs. AEs are general architectures that have been used in unsupervised learning for numerous tasks, including denoising (Vincent et al., 2008; 2010), manifold learning (Duque et al., 2020; Wang et al., 2014), clustering (Song et al., 2013; Yang et al., 2019), anomaly detection (Zhou & Paffenroth, 2017; Sakurada & Yairi, 2014), feature selection (Han et al., 2018; Gong et al., 2022), domain adaptation (Deng et al., 2014; Yang et al., 2021), segmentation (Baur et al., 2021; Myronenko, 2019), and generative models (Kingma, 2013; Doersch, 2016), making them a prominent use case when studying double descent in the field of unsupervised learning. We utilized under-complete AEs to ensure the model learns meaningful representations in the latent space, as over-complete AEs might simply learn the identity function and fail to capture the underlying data structure.

We present extensive empirical evidence showing that double and triple descent phenomena occur in unsupervised AEs when the data is contaminated with noise. Our findings reveal that "memorization" plays a critical role, as models can overfit the noise rather than capture the underlying signal, leading to poor test performance. However, we also discovered that sufficiently large models can still achieve superior test performance on clean data, even when fitting noisy training samples. This suggests that these models successfully extract the true signal despite the presence of noise. Through experiments on both synthetic and real-world data, we show that double and triple descent curves manifest in under-complete AEs exposed to various types of data contamination. Specifically, we show that different levels of sample noise, feature noise, domain shift, and the proportion of outliers all significantly influence the shape of the double descent curve. We observe model-wise, sample-wise, and epoch-wise double descent patterns across these settings. For instance, in Figure 1, we illustrate a double descent curve obtained by training an AE on data generated using the "sample noise" model described in Subsection 3.1. We follow (Nakkiran et al., 2021) in categorizing models as under-parameterized or over-parameterized, referring to those on the left or right of the critical regime, respectively. In these regimes, increasing the model size leads to a reduction in test loss. The critical regime is characterized by models for which changes in size can either decrease or increase test loss.

Our results have important implications for real-world applications, particularly in unsupervised learning tasks. We demonstrate that over-parameterized models trained on source domain data can adapt more effectively to target domains, even under distributional shifts. Additionally, we uncover non-monotonic behavior in anomaly detection performance as model complexity increases, further underscoring the practical relevance of our findings when noise and domain shifts are prevalent.

## 2 RELATED WORK

The discovery of double descent for neural networks (NNs) has led to extensive research aimed at understanding the behavior of generalization errors. It has also provided insights into why larger models perform better than smaller or intermediate ones. Most studies have been conducted in a supervised learning setting, as detailed in (Belkin et al., 2019a; Nakkiran et al., 2021; Dar et al., 2021; Spigler et al., 2018; Gamba et al., 2022; Adlam & Pennington, 2020; Liang et al., 2020; Chen et al., 2021; Xia et al., 2022; Kausik et al., 2023). Recent studies (Nakkiran et al., 2021; Hastie et al., 2022; Bartlett et al., 2020; Li et al., 2020) have introduced label noise, feature noise, and different levels of SNRs and demonstrated that large over-parameterized NNs can "memorize" the noise while still generalizing better than smaller models.

The phenomenon of double descent has not been extensively studied in the context of unsupervised learning, and there are some contradictions in the literature regarding its presence. Principal Component Analysis (PCA) (Shlens, 2014b) and Principal Component Regression (PCR) (Massy, 1965), which are special types of linear AEs, are widely used unsupervised and supervised learning

models respectively and can serve as an interesting case study for exploring double descent. Gedon et al. (2022) argued that there is no double descent in PCA while (Xu & Hsu, 2019; Teresa et al., 2022) showed evidence for double descent in PCR. Lupidi et al. (2023) used a specific subspace data model and argued that there is no sign of model-wise double descent in both linear and non-linear AEs. Sonthalia & Nadakuditi (2023) and Dubova (2022) demonstrated sample-wise double descent for denoising AEs with different SNR values. Zhang et al. (2023) used a self-supervised learning framework for signal processing and found epoch-wise double descent for different levels of noise.

Our analysis of double descent differs from previously published studies in three significant ways. Firstly, when trained on noisy data, we demonstrate that standard under-complete AEs experience double and even triple descent at the model-wise, sample-wise, and epoch-wise levels. We have also partitioned the model's size into bottleneck and other hidden layer dimensions to understand the phenomenon better. Secondly, we show that the noise magnitude and the number of noisy samples affect the double descent curve. Thirdly, we show that double descent also occurs in common realistic contamination settings in unsupervised learning, such as source-to-target domain shift, anomalous data, and additive sample and feature noise. Finally, we demonstrate the implications of multiple descents in unsupervised learning tasks using real-world data, extending beyond reconstruction.

## 3 DATA MODEL

This section outlines the data and contamination models used to study double descent.

### 3.1 LINEAR SUBSPACE DATA

We utilized the synthetic dataset of Lupidi et al. (2023) to challenge their assertion that "double descent does not occur in self-supervised settings". First, we sample $N$ random i.i.d. Gaussian vectors, each of size $d$, representing random features in a latent space, $z_i \sim \mathcal{N}(0, I_d)$. Next, we embed the vectors $\{z_i\}_{i=1}^N$ into a higher dimensional space of size $n$ by multiplying each $z_i$ by $D$ of size $n \times d$, $Dz_i$, where $D_{ij} \sim \mathcal{N}(0, 1)$. This setting can be thought of as measuring $\{z_i\}_{i=1}^N$ with a measurement tool $D$, resulting in higher-dimensional data. Our dataset differs from (Lupidi et al., 2023) in several ways, and we will investigate four scenarios as part of our study:

**Sample Noise.** We aim to investigate the impact of the number of noisy training samples on the test loss curve. In contrast to Lupidi et al. (2023), which adds noise to all samples, we vary the number of noisy training samples to identify memorization. We do this by introducing a new variable, $p$, representing the probability of a sample being noisy. Thus, $p \cdot 100\%$ represents the percentage of noisy samples in the data. As noise is added, we control the SNR by defining the parameter $\theta$. Another significant change from Lupidi et al. (2023) is the chosen values of SNR, which can be found in Appendix A, table 2, along with its calculation to derive $\theta$, in Appendix B. This leads to the following equation, which describes our model for the sample noise scenario:

$$x_i = \begin{cases} \theta Dz_i + \epsilon_i, & \text{with probability } p, \\ \theta Dz_i, & \text{with probability } 1 - p, \end{cases}$$

where $\epsilon_i \sim \mathcal{N}(0, I_n)$ is an additive white Gaussian noise (AWGN), representing the noise added to samples with probability $p$. This setting can be likened to using a noisy measurement device. To illustrate this generation we present in Appendix A, Figure 14 a visualization of the data model.

**Feature Noise.** We further study the impact of the number of noisy training features on the test loss curve. In this scenario, each sample $\{\theta Dz_i\}_{i=1}^N$ is affected by noise in certain features. We denote the probability of a feature being noisy by $p$, controlling each sample's noisy features. We simulate a scenario where we have $n$ measuring tools, each measuring a different feature. To introduce noise, we select the same set of features to be noisy across all samples. This mimics a situation where $\lfloor n \cdot p \rfloor$ of the measuring tools are unreliable or noisy. The SNR calculation is explained in Appendix B and Appendix A, Figure 14 depicts the data generation for this setting.

**Domain Shift.** We aim to explore how the test loss curve behaves when there is a domain shift between the train and test datasets. First, we partition the vectors in the latent space $\{z_i\}_{i=1}^N$ to train and test vectors, denoted as $z_{train}^i$ and $z_{test}^i$ respectively. Then, the train vectors are projected to a higher dimensional space with the matrix $D$, and the test vectors are projected with a different matrix $D''$, modeling a domain shift. To control the shift, we define $D'' = D + s \cdot D'$, where $D$ is

the matrix multiplying the train vectors and $D'_{ij} \sim \mathcal{N}(0,1)$ is a new random matrix added to cause perturbations at each entry of $D$ and the parameter $s > 0$ controls the shift between $D$ and $D''$.

$$x_i = \begin{cases} Dz^i_{train}, & \text{if } train, \\ D'' z^i_{test}, & \text{if } test. \end{cases}$$

Since $D$ and $D'$ are i.i.d., $D''$ follows a normal distribution $\mathcal{N}(0, (1 + s^2)I)$. To obtain the same norm in the test data, we divide $D''$ by $\sqrt{1 + s^2}$. This scenario is similar to the case where two different measuring instruments (i.e., $D, D''$) are measuring the same phenomenon. This data model is illustrated in Appendix A, Figure 15, and the definition of the SNR is detailed in Appendix B.

**Anomalies.** We conduct an experiment to investigate the impact of anomalies in the training set on the test loss curve. To represent clean samples, we utilize $\{\theta D z_i\}^N_{i=1}$. For generating anomalies, we sample from a normal distribution $\mathcal{N}(0, I_n)$. We introduce a metric termed Signal-to-Anomaly ratio (SAR), which regulates the magnitude ratio between the clean and anomaly samples through the parameter $\theta$. Subsequently, we substitute $p \cdot 100\%$ of the normal samples with anomalies. This generation is illustrated in Appendix A, Figure 16.

## 3.2 SINGLE-CELL RNA DATA

We utilized single-cell RNA sequencing data from (Tran et al., 2020) to illustrate our findings using real-world data. The data exhibits diverse domain shifts across different laboratory environments and measurement technologies. This dataset is crucial for assessing the impact of domain shifts on the test loss curve. Since this data is from a real-world setting, we are unable to control the shifts between the training (source) and testing (target) datasets, as explained in Subsection 3.1. We also use this dataset to show double descent when noise is injected to the samples and features (i.e., sample and feature noise) manually. We refer to Appendix A for more details.

## 3.3 CELEBA DATA

We incorporate real-world data to investigate anomaly detection across various model sizes. Specifically, we leverage the CelebA attributes dataset used in (Han et al., 2022), comprising over 200K samples and 4,547 anomalies, each characterized by 40 binary attributes. We sub-sample 3000 clean samples and replace $p \cdot 100\%$ of them with anomalies to create the training set.

## 4 RESULTS

Our results presented in the main text are primarily based on a multi-layer perceptron (MLP) under-complete AEs. However, we have also conducted additional evaluations using convolutional NNs (CNNs). These findings are presented in Appendix E.2. Each of the reported results is based on 5 to 15 random seeds. Complete implementation details and discussion on the high computational load that each figure requires can be found in Appendix A. All models are trained using contaminated datasets and tested on clean data. Consequently, the test loss serves as an indicator of whether the model has memorized the noise (high test loss) or learned the signal (low test loss). Over-complete AEs are beyond the scope of this discussion, as they can learn the identity function, leading to trivial and uninformative data learning. Our emphasis is on standard (unsupervised) AEs, where in the training process, we minimize the mean squared error (MSE) between the input and the model's output. Train loss figures corresponding to all test losses depicted in this section are provided in Appendix C and more results with a non-linear synthetic dataset are presented in Appendix E.3.

### 4.1 MODEL-WISE DOUBLE DESCENT

This section analyzes the test loss with increasing model sizes. For AEs, we break down the well-known "double descent" phenomenon into two interconnected variations: "hidden-wise" and "bottleneck-wise" and show how both contribute to the double descent behavior in the test loss. We also study the influence of several contaminations described in Section 3 and conclude that the inter-polation threshold location and value can be manipulated by these factors. We also found that double descent typically occurs with high levels of sample noise and low SNR values. In these settings, the

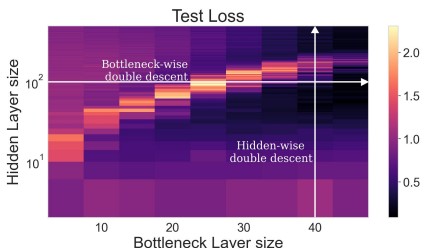 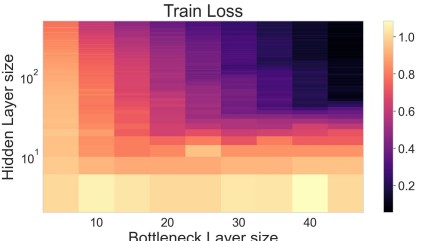

Figure 2: Test and train losses as a function of model size. The test loss demonstrates clear double descent when varying the bottleneck or hidden layer size. The AEs were trained on the linear subspace model with sample noise = 90% and SNR = -15 [dB] (see details in Subsection 3.1).

noise predominates the training set, leading models in the critical regime to focus on interpolating the noise rather than learning the underlying signal to reduce the training loss. This, in turn leads to higher test loss. This explains why Lupidi et al. (2023) did not observe the phenomenon, as they used only high SNR levels (10 dB) in the model-wise sample noise scenario.

In Figure 2, we provide visual evidence of the bottleneck-wise and hidden-wise double descent. This not only helps to distinguish between various model sizes but also underscores the significance of our different architectural choices. The training loss consistently decreases as the dimensions of the model increase. In contrast, both the bottleneck and hidden layers exhibit the characteristic double descent curve, as seen in the decrease in test loss, followed by an increase and then another decline. This demonstrates that AEs trained on contaminated data can exhibit double descent.

**Sample and feature noise.** Interestingly, Figure 3a shows that the height of the test loss increases and the interpolation threshold location shifts towards larger models as the level of sample noise increases. This can be clarified by the observation that increased noise adversely affects model learning. Moreover, we need a bigger model to overfit the noisy samples. The absence of double descent for 0-20% sample noise can be attributed to the insufficient number of noisy samples in the training data. In Figure 3b, we demonstrate triple descent using single-cell RNA data, where we notice a similar behavior for the test loss, specifically for each of the two peaks. Evidence for double descent using the feature noise data model introduced in Subsection 3.1 is presented in Appendix D.

In addition, we observed the phenomenon of final ascent, characterized by double descent following a final increase in the test loss, initially discussed in the case of supervised learning (Xue et al., 2022). We present the results of final ascent for unsupervised AEs with the single-cell RNA data for 0-20% sample noise in Appendix E.4, Figure 55b. We also show how double and triple descent patterns emerge under different types of noise, such as Laplacian noise, and find double descent for sparse AEs, as detailed in Appendix E.5.

**SNR.** We observed that the SNR plays a crucial role in the test loss, which in turn affects its height. A higher SNR value reduces the impact of noise, allowing the model to learn the underlying signal from the training set, resulting in a lower test loss. Conversely, a lower SNR value amplifies the influence of noise, disrupting the model's ability to learn the signal leading to inferior results in the test loss. In Figure 4a, for SNR = 0 [dB], the double descent curve is absent because it prevents models in the critical regime from memorizing the noise, as the noise is not sufficiently dominant. Figure 4 and Appendix D, Figure 28 present results for the sample and feature noise settings respectively.

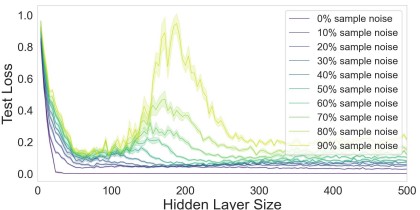 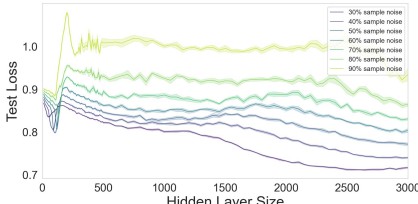

(a) **Linear subspace data**. SNR = -15 [dB].   (b) **Single-cell RNA data**. SNR = -17 [dB].

Figure 3: Test loss exhibits model-wise double and triple descents for the case of varying **sample noise**. Train losses are depicted Appendix C, Figure 19.

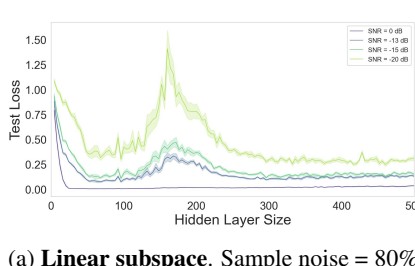
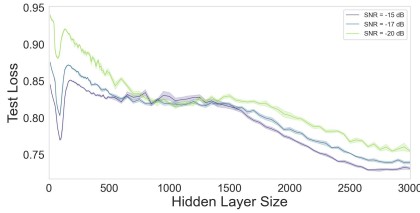

(a) **Linear subspace**. Sample noise = 80%.    (b) **Single-cell RNA**. Sample noise = 40%.

Figure 4: The effect of SNR for the case of **noisy samples** on the test loss curve. Train losses are illustrated in Appendix C, Figure 20.

**Domain shift.** We study the existence of double descent when the distribution of the training (source) data, differs from that of the testing (target) data. We investigate the impact of the model size on learning shared representations for both source and target datasets and reducing the shift between them. By training the model on the source data and testing it on different targets, we unveil non-monotonic behavior for the linear subspace dataset, shown in Figure 5. Additionally, the test loss rises as the shift is more dominant, and for lower levels of domain shift (shift = 0.1, 0.5), non-monotonic behavior does not occur since the source and target domains are closely aligned. In these cases, even models that interpolate the source data and learn domain-specific representations perform well on the target data, preventing an increase in the test loss. Furthermore, we identify that over-parameterized models result in lower test loss, leading to improved target data reconstruction. Subsection 5.1, Figure 12 presents double and triple descent results for the single-cell RNA data and further insights about the connection of model size and domain adaptation utilizing real-world data.

**Anomaly detection.** We also identify double descent occurring when anomalies, deviating from the expected behavior of the data are introduced into the training set. Following the unsupervised setting, we consider the scenario where there is no anomaly-free dataset available for training, making it more challenging to differentiate between normal and anomalous data (Cheng et al., 2021). In particular, we use the anomaly dataset mentioned in Subsection 3.1 with high number of anomalies in the train set, akin to (Lindenbaum et al., 2024; Lerman & Maunu, 2018b;a), which included anomalies with up to 99.5% for the subspace recovery setting and (Han et al., 2022), which includes up to 40% anomalies in the case of unsupervised anomaly detection. We study the test loss curves by varying the amounts of anomalous training samples. We used a common method where data points with reconstruction loss surpassing a defined threshold are identified as anomalies as discussed in (Lindenbaum et al., 2024; Malhotra et al., 2016; Borghesi et al., 2019).

We evaluate the anomaly detection capabilities using the receiver operating characteristic area under the curve (ROC-AUC) metric. This metric employs the reconstruction error to measure the model's ability to distinguish between clean and anomalous data (anomalies are identified as data points with errors crossing a defined threshold). A higher ROC-AUC value signifies superior performance. The models in the critical regime depicted in Figure 6a result in higher test loss of the clean samples, complicating the differentiation between clean and anomaly data, leading to worse ROC-AUC results. Scaling up the model size results in a secondary descent in the test loss of the clean data. This double descent curve is particularly evident under conditions of low SAR and a high number of anomalies in the training set, similar to the results of the sample noise scenario (the anomalies play the role of the noise). This secondary descent facilitates the model's ability to differentiate between clean and anomalous data, resulting in performance comparable to that of the under-parameterized models in terms of ROC-AUC, while learning meaningful embedding for both clean data and outliers, resulting in lower test losses. Figure 6b demonstrates the absence

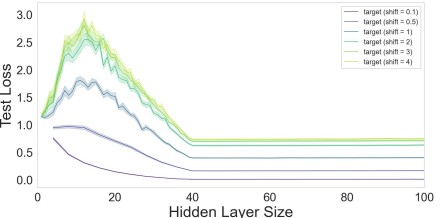

Figure 5: **Linear subspace data** exhibits model-wise non-monotonic behavior for varying **domain shifts**. Appendix C, Figure 21 shows the train loss behavior.

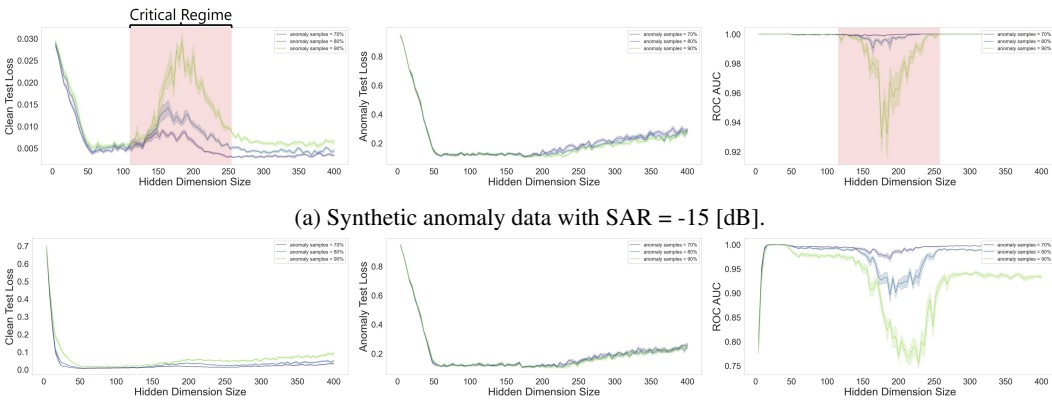

(a) Synthetic anomaly data with SAR = -15 [dB].

(b) Synthetic anomaly data with SAR = 0 [dB].

Figure 6: **Left:** test loss of the clean samples. A double descent pattern emerges for low SARs and high anomaly presence in the training data. **Middle:** test loss of the anomaly data. **Right:** Non-monotonic behavior of the ROC-AUC.

of double descent due to high SAR. However, similar to Figure 6a, intermediate models exhibit poorer ROC-AUC performance compared to under and over-parameterized models. We present more insights on anomaly detection utilizing real-world data in Subsection 5.2.

In conclusion, as contamination setups become more severe, such as higher noise levels, significant domain shifts, many anomalies, or low SNR, the double descent phenomenon becomes more pronounced, and the test loss increases. In some instances, these noise levels also cause the critical regime to shift to the right. In Appendix A, Table 1, we compare the existence of double descent between unsupervised and supervised learning for varying contamination setups and conclude that they result in similar behaviors.

### 4.2 EPOCH-WISE DOUBLE DESCENT

In this section, we explore the presence of double descent versus the number of epochs. This study represents the first unsupervised investigation of this kind, expanding on similar research carried out by (Nakkiran et al., 2021) for supervised learning. Figure 7 and Appendix D, Figure 29 show the impact of the number of noisy samples and features in the train set on the test loss curve respectively. Increasing noise makes it harder for the model to learn the signal, leading to a higher test loss. A similar effect is obtained when varying the SNR, where as it decreases, the noise becomes more dominant, resulting in an increase in the test loss. This is illustrated in Figure 8 and in Appendix D, Figure 30 for the case of sample and feature noise respectively. Epoch-wise double descent is also present when there is a domain shift between the train and test sets, as illustrated in Figure 9. 9a shows that the stronger the shift, the higher the test loss.

### 4.3 SAMPLE-WISE DOUBLE DESCENT

In this section, we study the impact of the number of training samples on the test loss curve. The complexity of a model and the number of samples it is trained on both play a crucial role in determining whether the model is over (small sample size) or under-parameterized (large sample size).

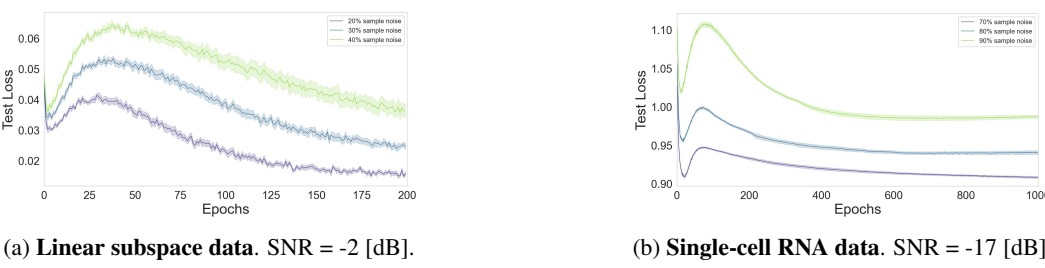

(a) **Linear subspace data**. SNR = -2 [dB].  (b) **Single-cell RNA data**. SNR = -17 [dB].

Figure 7: Epoch-wise double descent influenced by the number of **noisy samples**. Train losses are depicted in Appendix C, Figure 22.

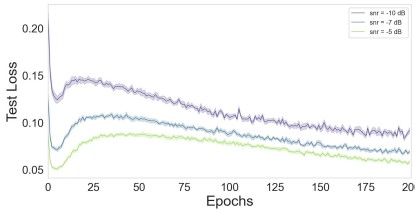 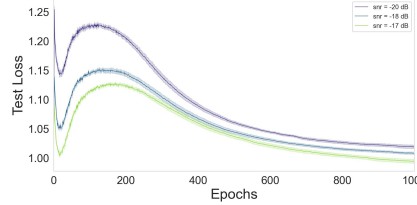

(a) **Linear subspace**. Sample noise = 40%.      (b) **Single-cell RNA**. Sample noise = 90%.

Figure 8: Epoch-wise double descent for the case of **sample noise** influenced by the SNR. Train losses are exhibited in Appendix C, Figure 23.

This causes the interpolation threshold's location to change, as shown in Figure 10. This adjustment can sometimes result in a model that performs worse than a model trained on a smaller set of training samples. A similar phenomenon was demonstrated in (Nakkiran et al., 2021) in a supervised setting.

We also investigate the impact of gradually increasing the number of training samples on the test loss curve while keeping the model's size fixed. Remarkably, we identify a non-monotonic trend in the test loss curve at Figures 11b, 11c, and Appendix D, Figures 31a and 31b, which sometimes results in double descent as noticed in 11a. The emergence of non-monotonic behavior is defined by a phase where an increased number of samples negatively impacts performance, resulting in higher test loss. Appendix C, Figure 26 depicts the training loss plotted against the number of samples in the scenario of sample noise. Figure 11 showcases only the results from the linear subspace dataset due to the insufficient amount of samples in the single-cell RNA dataset. More results regarding sample-wise double descent can be found in Appendix E.2, E.3. The impact of the noise level, SNR, and the domain shift on the test loss is consistent with the analyses conducted in Subsections 4.1 and 4.2.

## 5 REAL WORLD APPLICATIONS

In this section, we demonstrate how our findings can be applied to important tasks in machine learning, such as domain adaptation and anomaly detection. Our objective is to emphasize the significance of model size selection rather than to compete with state-of-the-art techniques.

### 5.1 DOMAIN ADAPTATION

Many frameworks in machine learning are exposed to domain shifts. The difference in distribution between the training and testing data can lead to inferior results when the model is employed on new, unseen data. Numerous domain adaptation methods have been proposed for both supervised and unsupervised settings (Zhou et al., 2022; Peng et al., 2019; Chang et al., 2019; Rozner et al., 2023; Yampolsky et al., 2023) to minimize the shift between the source and target domains. This is an ongoing challenge in biology, where researchers attempt to integrate datasets collected under different environmental conditions that cause distribution shifts. Many studies have been conducted to develop strategies to mitigate this shift, known in biology as "batch effect" (Tran et al., 2020).

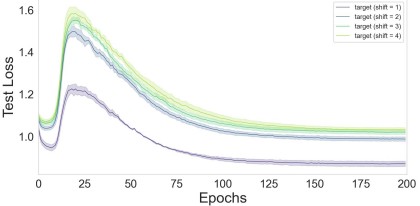 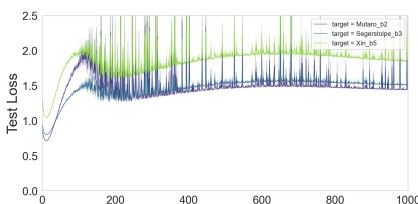

(a) **Linear subspace data** with 5% noisy samples and SNR = 20 [dB].      (b) **Single-cell RNA data**. The 'Wang' batch was excluded due to noisy results.

Figure 9: Epoch-wise double descent influenced by the amount of **domain shift**. **Left:** we introduce some noise to emphasize the double descent curve. Appendix C, Figure 24 shows the train losses.

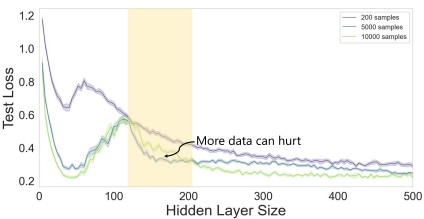

Figure 10: Model-wise double descent for the **linear subspace data** with different number of training samples. In the yellow interval, models trained with 10000 samples perform worse compared to those trained with 5000 samples. Sample noise = 70% and SNR = -15 [dB]. Train loss results are shown in Appendix C, Figure 25.

In this section, we study the relation between model size and its ability to alleviate distribution shifts in real world single-cell RNA data 3.2. Our work is the first to show the advantage of over-parameterized models in unsupervised tasks under real domain shifts through the emergence of double descent. Tripuraneni et al. (2021) and Kausik et al. (2023) focused on supervised learning and showed related results, each under their own set of assumptions. We visualize the source and target datasets using UMAP embeddings (McInnes et al., 2018) in Appendix E.1, Figure 32b. The top two sub-figures in Figure 12 present the test and train losses respectively for models trained on source and tested on target datasets. We observed that testing models on the 'Wang' dataset results in triple descent, while all other targets result in double descent.

We used the KL-divergence (KLD) (Shlens, 2014a) metric to calculate the distribution shift between our source data ('Baron') and the target datasets ('Segerstolpe', 'Xin', 'Mutaro', and 'Wang'). Our findings show that as the shift between the source and target increases (higher KLD), the test loss rises, inline with the results of the simulated experiment yielding Figure 5. To evaluate how different models perform in terms of domain adaptation, we measure how much of the shift was removed by analyzing the bottleneck representations of the AEs. Precisely, we compute the $k = 10$ nearest neighbors of each bottleneck vector and determine the proportion belonging to the same biological batch as mentioned in (Schilling, 1986), Section 3. We call this metric "k-nearest neighbors domain adaptation test" (KNN-DAT), indicating the extent of mixing between different domains. KNN-DAT of 1 implies complete separation, while a lower value indicates better mixing of different domains. That is, lower values of KNN-DAT imply that the embedding of samples from the target domain is more similar to the embedding of samples from the source domain.

The bottom row in Figure 12 presents the UMAP representations based on the embeddings of the learned AEs. For under-parameterized models, KNN-DAT results are better compared to the models in the critical regime. However, they achieve lower KNN-DAT at the expense of learning the source data inadequately (high train loss). The model in the critical regime learns domain-specific features, resulting in high KNN-DAT. We find that over-parameterized models yield the best KNN-DAT results, achieving a score of 0.75. They also lead to reduced test loss, resulting in improved reconstruction of the target data. This suggests that *over-parameterized models facilitate the transition between source and target datasets, serving as a viable domain adaptation strategy*. We also display results for the linear subspace dataset in Appendix E.1.

## 5.2 ANOMALY DETECTION

Unsupervised anomaly detection is a crucial task in machine learning. It has various applications across scientific fields, and many studies have utilized AEs for anomaly detection (Chandola et al., 2009; Lindenbaum et al., 2024; Chen et al., 2018; Rozner et al., 2024). We train our AEs on both

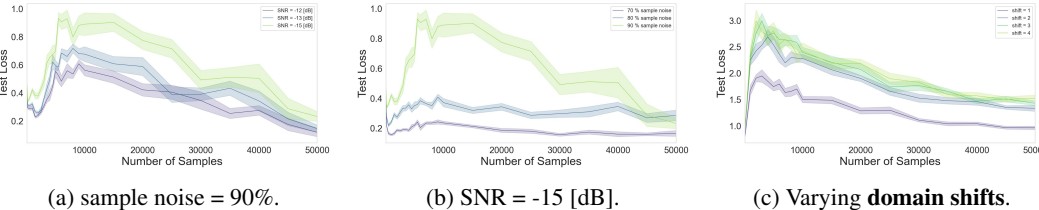

(a) sample noise = 90%.    (b) SNR = -15 [dB].    (c) Varying **domain shifts**.

Figure 11: **Sample-wise** non-monotonicity and double descent for the **linear subspace data**. Results for the feature noise scenario are presented in Appendix D, Figure 31 and the training losses in Appendix C, Figure 26.

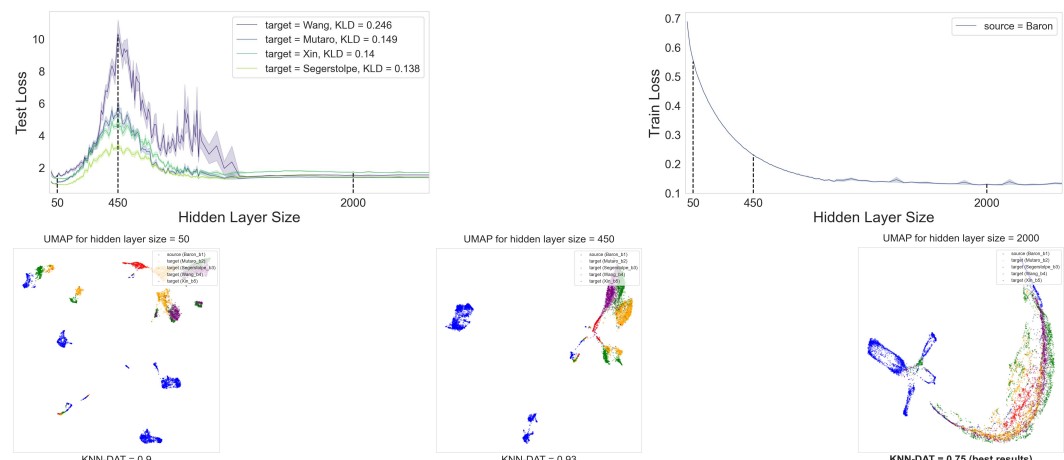

Figure 12: Top: test and train losses of different model sizes utilizing the **single-cell RNA** dataset. Training is done on the source data while testing on the target data. Bottom: UMAP of bottleneck vectors extracted from the encoder's output and KNN-DAT results for different model sizes.

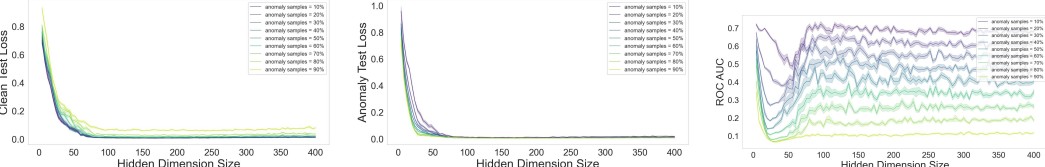

Figure 13: **Left, middle:** test loss of clean and anomaly data respectively. **Right:** non-monotonic behavior of the ROC-AUC for the celebA dataset.

normal and anomaly data and detect anomalies based on the reconstruction loss as detailed in Subsection 4.1. We used the CelebA attributes dataset and conducted an experiment similar to the one in Subsection 4.1 to investigate how the model's size affect its ability to detect anomalies. As expected, in line with the findings in (Han et al., 2022), small models outperform larger models in anomaly detection (the highest ROC-AUC is achieved by the smallest models in Figure 13). Since we do not control the SAR value in this data, which is positive, we do not observe a double descent in the test loss curves. Nonetheless, we identify a non-monotonic behavior of the ROC-AUC curve. Initially, it decreases for intermediate models, followed by an increase for over-parameterized models. In conclusion, when employing a model for unsupervised anomaly detection, *it is recommended to avoid selecting intermediate models, as their anomaly detection performance is inferior to under and over-parameterized models*.

## 6 CONCLUSIONS

In our study, we identified various instances of multiple descents and non-monotonic behaviors in unsupervised learning. These phenomena occur at the model-wise, epoch-wise, and sample-wise levels. We used under-complete AEs to investigate these phenomena and found compelling evidence for their robustness across diverse datasets, model types, and experimental scenarios. We examined four distinct use cases: sample noise, feature noise, domain shift, and anomalies. Our experiments revealed multiple instances of consecutive descents, with most of them resulting in improved (lower) test loss. Additionally, we found a connection between the model's size and its real-world performance. Specifically, over-parameterized models can serve as effective domain adaptation strategies when there is a distribution shift between the source and target data. In the realm of anomaly detection, we find that it is important to avoid selecting intermediate models that yield lower ROC-AUC outcomes.Our work was limited by computational resources, preventing us from using larger datasets for training. However, we hope that our findings will benefit research groups with greater computational capabilities, enabling them also to explore other frameworks in unsupervised learning, such as generative models. Another exciting direction for future research is developing theoretical frameworks that explain our findings, using similar ideas such as in (Curth et al., 2024; Curth, 2024).

## REPRODUCIBILITY

Please refer to Appendix A for all the necessary information for reproducing the results. This includes a detailed explanation of the datasets, which is also discussed in Section 3. Additionally, Appendix A covers more datasets which are used in Appendix E, model types, hyperparameters, and the loss function used during training. The SNR calculations are provided in Appendix B.

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

# A IMPLEMENTATION DETAILS

In this section, we provide complete implementation details for all experiments conducted in the paper. Illustrations of the linear subspace dataset generation introduced in Subsection 3.1 for the scenarios of sample and feature noise, domain shift, and anomalies are displayed in Figures 14, 15, and 16 respectively. We also provide Table 1 for comparison between the double descent results of supervised and unsupervised regimes.

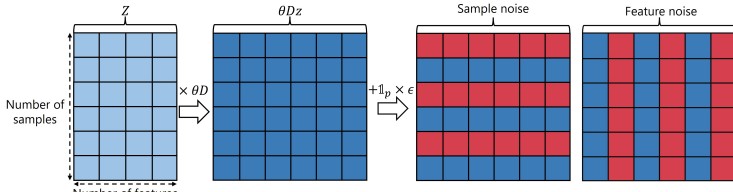

Figure 14: Data generation for the scenarios of sample and feature noise with $p = 0.5$. The first (leftmost) matrix depicts the latent vectors $Z$. The second matrix illustrates the latent vectors being projected into a higher dimensional space, and the rightmost matrices contain clean (blue) and noisy (red) samples / features respectively.

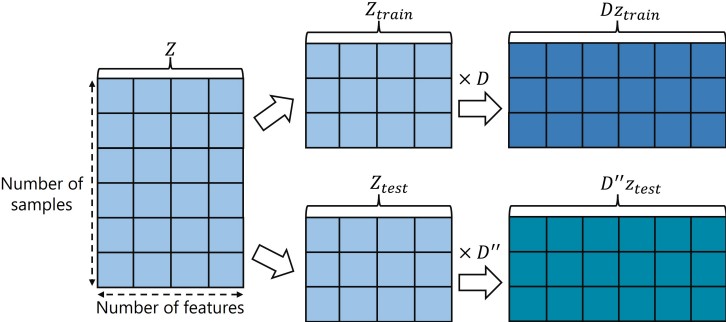

Figure 15: Data generation for the scenario of domain shift. The matrix on the left depicts the latent vectors $Z$ and the two middle matrices represent the separation to source ($Z_{train}$) and target ($Z_{test}$). The two rightmost matrices illustrate the latent vectors of the train and test data being projected into a higher dimensional space with different matrices ($D, D''$), resulting in a domain shift.



Figure 16: Data generation for the case of anomalies and $p = 0.5$. The matrix on the left depicts the anomalous data, the middle matrix represents the clean data, and the rightmost matrix contains both clean (blue) and outlier (red) samples.

**Parameters.** Table 2 details the hyper-parameters and other variables for the training process with the linear subspace, non-linear subspace (Appendix E.3), single-cell RNA, CelebA, and MNIST (Appendix E.2) datasets. The training optimizer utilized was Adam (Kingma & Ba, 2014), and the loss function for reconstruction is the mean squared error, which is mentioned in this Section.

**Data.** For the **linear subspace**, **non-linear subspace**, and **MNIST** datasets, we generate 5000 samples for training and 10000 for testing across all scenarios (sample noise, feature noise, domain shift, and anomaly detection). Regarding the **single-cell RNA** data, we have focused on dataset number 4 from (Tran et al., 2020), which includes 5 distinct domains (biological batches) named 'Baron', 'Mutaro,' 'Segerstolpe,' 'Wang,' and 'Xin', each representing 15 different cell types. Each cell (sample) in this dataset contains over 15000 genes (features). To facilitate the training of deep

Table 1: The existence of double descent in unsupervised and supervised learning.

| Setup | Unsupervised | Supervised | Behavior | Notes (unsup.) |
|---|---|---|---|---|
| Sample noise | exists (our paper) | exists (see Sections 1, 2) | similar | low SNR needed |
| Feature noise | exists (our paper) | exists (see Section 2) | similar | low SNR needed |
| Domain shift | exists (our paper) | exists (see Subsection 5.1) | similar | |
| Anomalies | exists (our paper) | not explored | - | |

Table 2: Parameters and hyper-parameters

| Parameters | Linear/ non-linear Subspace | RNA | CelebA | MNIST |
|---|---|---|---|---|
| Model | MLP | MLP | MLP | CNN |
| Learning rate | 0.001 | 0.001 | 0.001 | 0.001 |
| Optimizer | Adam | Adam | Adam | Adam |
| Epochs | 200 | 1000 | 200 | 1000 |
| Batch size | 10 | 128 | 10 | 128 |
| Data's latent size ($d$) | 20 | - | - | - |
| Number of features ($n$) | 50 | 1000 | 40 | 784 |
| Train dataset size | 5000 | 5000 | 3000 | 5000 |
| SNR/ SAR [dB] | -20, -15, -10, -7, -5, -2, 0, 2 | | | |
| Sample/ feature noise ($p$) | 0, 0.1, 0.2,...,1 | | | |
| Domain shift scale ($s$) | 1, 2, 3, 4 | - | - | - |
| Bottleneck layer size | 25, 30, 45 | 20, 100, 300 | 25 | 10, 30, 50, 500 |
| Hidden layer size | 4 - 500 | 10 - 3000 | 4-400 | - |
| Channels | - | - | - | 1-64 |

models while preserving the domain shift, we have retained the top 1000 prominent features. We utilize the 'Baron' biological batch as our source data for the scenario of domain shift, comprising 5000 training samples, while the target batches are 'Mutaro' (2122 samples), 'Segerstople' (2127 samples), 'Wang' (457 samples), and 'Xin' (1492 samples). As for the sample and feature noise scenarios, we use the 'Baron' domain for both sample and feature noise scenarios due to its largest sample size (8569). We allocate 5000 samples for training and introduce additive white Gaussian noise (AWGN) to specific samples and features, as described in subsection 3.1. The calculations of the SNR for both sample and feature noise cases are provided in Section B. The reserved 3569 samples are for testing. Please be aware that all the domains in this dataset are inherently noisy, reflecting their real-world nature. Therefore, even when no additional noise is applied ($p = 0$), the data remains noisy. This may account for why the test loss does not decrease monotonically as the model size increases for cases with low noise levels, as shown in Appendix E.4, Figure 55b.

For the **celebA** dataset, we sub-sample 3000 clean samples and replace $\lfloor 3000 \cdot p \rfloor$ of them with anomalies to ensure that $\sim p \cdot 100\%$ of the data is contaminated with anomalies. Due to the limited availability of anomaly data (4547 samples), the test set includes $\lfloor (1 - p) \cdot 4547 \rfloor$ anomalies along with an equal number of clean samples.

**Models.** All experiments, including the linear subspace, non-linear subspace, single-cell RNA, and celebA datasets are conducted using the same MLP AE architecture. To facilitate the exploration of double descent in both bottleneck layer size and hidden layer size, we employ a simplified model mentioned in (Lupidi et al., 2023) consisting of a single hidden layer for both the encoder and decoder, as depicted in Figure 17. We also utilize a CNN AE architecture consisting of three convolution layers in the encoder part, followed by a bottleneck layer, and then a decoder part consisting of three deconvolution layers trained on the MNIST dataset as illustrated in Figure 18 (results for the CNN AE are reported in Appendix E.2).

We work with under-complete AEs to encourage the acquisition of a meaningful embedding in the latent space and prevent the model from learning the identity function. The size of these models is determined by the sizes of the hidden layers (for MLP), the number of channels (for CNN), and the bottleneck layer, while the width of these models remains constant.

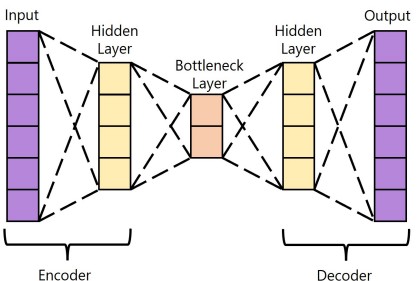
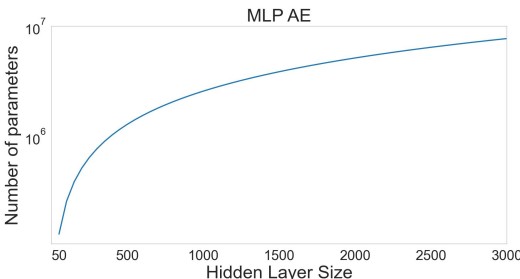

Figure 17: **Left:** Demonstration of the MLP-based AE model structure. **Right:** model's number of parameters for single-cell RNA settings (bottleneck layer size = 300 and input size = 1000 features).

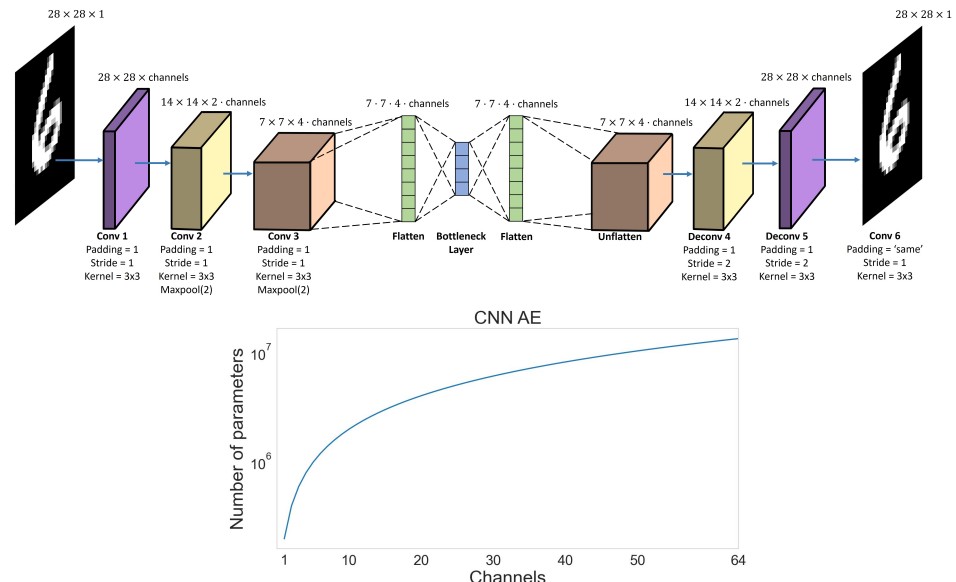

Figure 18: **Upper:** Demonstration of CNN AE model structure. **Lower:** number of parameters.

**Loss function.**  All AEs are trained with the mean squared error (MSE) loss function:

$$\text{MSE} = \frac{1}{n}\sum_{i=1}^{n}(y_i - \hat{y}_i)^2.$$

Where $n$ is the number of data samples, $y_i$ is the true value, and $\hat{y}_i$ is the predicted value. Due to contamination in the training dataset, the norm of train samples tends to be higher than that of the clean test samples. As the MSE loss is not scale-invariant, we opt to normalize both train and test losses only after the training process is complete, using $\frac{1}{n}\sum_{i=1}^{n}(y_i - \bar{y})^2$, Where $\bar{y}$ is the mean of $\{y_i\}_{i=1}^{n}$. This strategy enables us to continue utilizing the MSE loss function while facilitating a fair and meaningful comparison between train and test losses.

**Results.**  Ensuring the robustness of the findings across various model initializations and enhancing their reliability, all figures combine several results of different random seeds. The bolded curves in each figure represent the average across the results of different seeds, and the transparent curve around each bolded curve represents the $\pm 1$ standard error from the mean.

**Environments and Computational Time.**  All experiments were conducted on NVIDIA RTX 6000 Ada Generation with 47988 MiB, NVIDIA GeForce RTX 3080 with 10000 MiB, Tesla V100-SXM2-32GB with 34400 MiB, and NVIDIA GeForce GTX 1080 Ti with 11000 MiB.
Each result in Figure 3 represents an average over 10 seeds. The hidden layer sizes for the linear subspace data range from 4 to 500 with a step size of 4, and for the single-cell RNA data, they range from 10 to 500 with a step size of 10, and from 500 to 3000 with a step size of 50. This

results in 125 and 110 models trained for each dataset, respectively. Figure 3a illustrates 10 different sample noise levels, requiring the training of $125 \times 10 \times 10 = 12,500$ models. Similarly, Figure 3b depicts 4 different sample noise levels, corresponding to $110 \times 10 \times 4 = 4,400$ trained models. In total, 16,900 models, each with up to 8 million parameters trained on 5,000 data points were needed to obtain the results. In Appendix E.2, Figure 34, we present results for CNNs including between 1 to 64 channels trained on 5,000 images from the MNIST dataset including 5 levels of sample noise and 6 levels of feature noise for 10 different seeds. These experiments require training $64 \times 11 \times 10 = 7,040$ models with up to 13 million parameters. Each evaluation of a specific experiment takes several days if trained on the NVIDIA RTX 6000 and weeks if trained on the other mentioned GPUs to obtain the results.

## B  SNR CALCULATIONS

In this section, we will outline our approach for calculating the signal-to-noise ratio (SNR) for all experiments involving the addition of noise. Initially, we convert the SNR from decibels to linear SNR using the formula:

$$\text{SNR} = 10^{\left(\frac{\text{SNR[dB]}}{20}\right)}. \tag{1}$$

We have a closed-form equation for the linear subspace dataset to determine the scalar $\theta$ required to multiply the train samples and achieve the desired linear SNR value. We use the fact that both train and noise are sampled from an i.i.d. normal distribution and calculate $\theta$ for the sample noise, feature noise, domain shift, and anomalies.

**Notations**:
$z - d \times 1$ vector. Represents a vector in a latent space of size $d$.
$D - n \times d$ matrix. Represents a random matrix to project $z$ from a $d$ dimensional space into a higher-dimensional space ($n > d$).
$\epsilon - n \times 1$ vector. Represents the noise added to a vector with $n$ dimensions.

For the scenario of sample noise, where a particular sample is affected by noise across all its features:

$$\text{SNR}^2 = \frac{E[\|\theta Dz\|_2^2]}{E[\|\epsilon\|_2^2]} = \frac{E[\theta^2 z^T D^T D z]}{E[\epsilon^T \epsilon]} = \frac{\theta^2 E_z[E_{D|z}[z^T D^T D z|z]]}{E\left[\sum_{i=1}^{n} \epsilon_i^2\right]} \underbrace{=}_{(a)} \tag{2}$$

$$\frac{\theta^2 E_z[z^T E_{D|z}[D^T D]z]}{n} \underbrace{=}_{(b)} \frac{\theta^2 E_z[z^T n \cdot I_{d \times d} z]}{n} = \frac{\theta^2 \cdot n \cdot E_z[z^T z]}{n} = \theta^2 E\left[\sum_{i=1}^{d} z_i^2\right] \underbrace{=}_{(a)} \theta^2 \cdot d.$$

Isolating $\theta$, we get that $\theta = \frac{\text{SNR}}{\sqrt{d}}$.

(a) Given a vector $a \sim \mathcal{N}(0, I_n)$ of $n$ i.i.d. samples, $E\left[\sum_{i=1}^{n} a_i^2\right] = \sum_{i=1}^{n} E[a_i^2] = \sum_{i=1}^{n} 1 = n$.

(b) Given a matrix $M \sim \mathcal{N}(0, I_n)$ of size $n \times n$ where all entries are i.i.d., then

$$E[M^T M] = E\begin{bmatrix} M_{1,1}^2 + \cdots + M_{n,1}^2 & \cdots & M_{1,1}M_{1,n} + \cdots + M_{n,1}M_{n,n} \\ \vdots & \ddots & \vdots \\ M_{1,1}M_{1,n} + \cdots + M_{n,1}M_{n,n} & \cdots & M_{1,n}^2 + \cdots + M_{n,n}^2 \end{bmatrix} =$$

$$\begin{bmatrix} n & \cdots & 0 \\ \vdots & \ddots & \vdots \\ 0 & \cdots & n \end{bmatrix} = n \cdot I_{n \times n}.$$

For the scenario of feature noise, each train sample has only $n \cdot p$ noisy features, meaning the noise vector contains values for only $n \cdot p$ entries. Consequently, $\theta$ is determined by $\sqrt{\frac{p}{d}} \cdot \text{SNR}$. For practitioners who want to explore the scenario involving domain shift, where the source and target are noisy, note that the matrix responsible for projecting $z_{test}$ into a higher-dimensional space is denoted as $D'' = D + s \cdot D'$ where $D'$ is sampled from a standard normal distribution $\mathcal{N}(0, I)$ and both $D$ and $D'$ are i.i.d. Consequently, $D''_{ij} \sim \mathcal{N}(0, 1 + s^2)$. Substituting $D$ with $D''$ in equation equation 2, we find that $E_{D''|z}[D''^T D''] = n \cdot (1 + s^2) \cdot I_d$, leading to $\text{SNR}^2 = (1 + s^2) \cdot \theta^2 \cdot d$,

therefore $\theta = \frac{SNR}{\sqrt{(s^2+1)d}}$. In other words, since the covariance matrix of $D''$ is $(1+s^2)I$, we need to make sure we first normalize the matrix by $\sqrt{1+s^2}$ to maintain the identity covariance matrix.

For other datasets, such as the single-cell RNA dataset, we normalize each sample $x$ by its norm $\|x\|$, and similarly normalize each noise vector $n$, yielding: $\hat{x} = \frac{x}{\|x\|}$ and $\hat{n} = \frac{n}{\|n\|}$. This ensures that the ratio $\frac{\hat{x}}{\hat{n}}$ equals 1. By employing equation equation 1, we attain the intended linear SNR factor $\theta$, and then scale down $\hat{n}$ by $\theta$, yielding $\hat{n}_{scaled} = \frac{\hat{n}}{\theta}$. This guarantees that the linear SNR is $\frac{\hat{x}}{\hat{n}_{scaled}} = \theta$.

## C   TRAIN LOSS RESULTS

In this section, we provide the train loss figures corresponding to each of the test losses mentioned in the main paper.

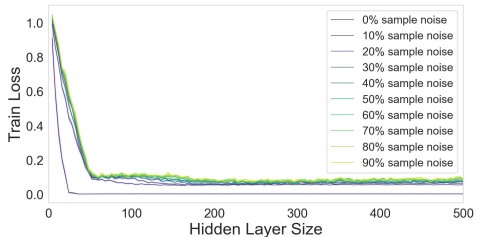
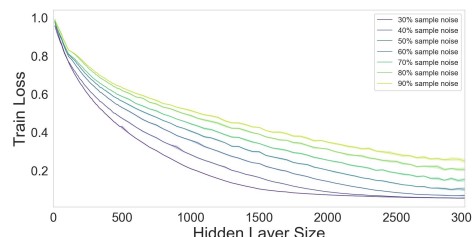

(a) **Linear subspace data** with SNR = -15 [dB].   (b) **Single-cell RNA data** with SNR = -17 [dB].

Figure 19: Model-wise train losses for the case of varying **sample noise**.

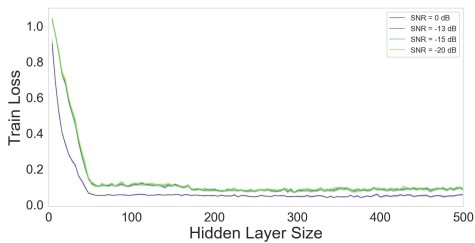
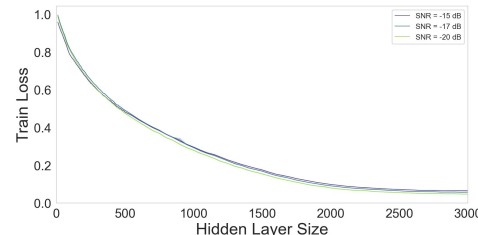

(a) **Linear subspace data**. Sample noise = 80%.   (b) **Single-cell RNA data**,. Sample noise = 40%.

Figure 20: Model-wise train losses for the case of **noisy samples** and varying SNR.

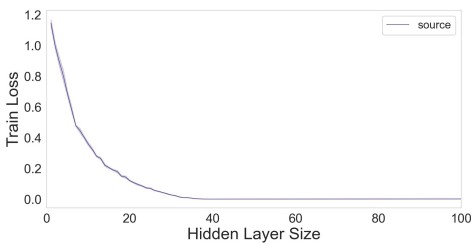

Figure 21: Train loss of source data for the **linear subspace data**.

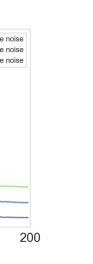
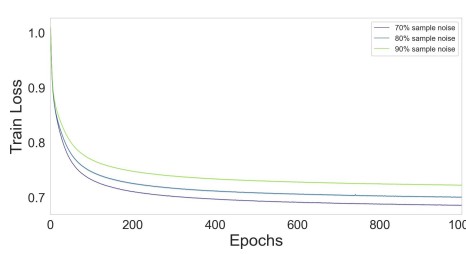

(a) **Linear subspace data** with SNR = -2 [dB].  (b) **Single-cell RNA data** with SNR = -17 [dB].

Figure 22: Epoch-wise train losses for varying number of **noisy samples**.

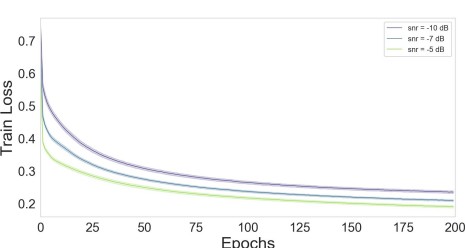
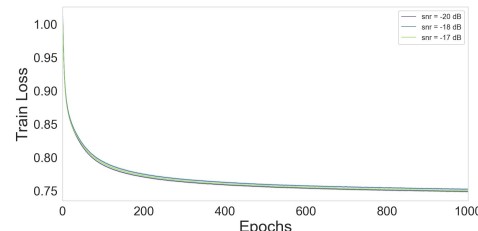

(a) **Linear subspace data**. Sample noise = 40%.  (b) **Single-cell RNA data**. Sample noise = 90%.

Figure 23: Epoch-wise train losses for the case of **sample noise** influenced by the SNR.

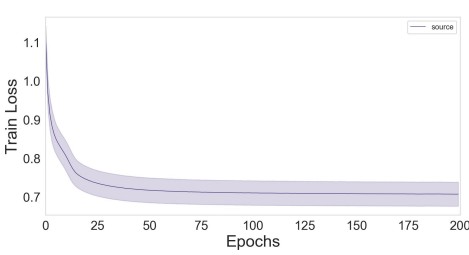
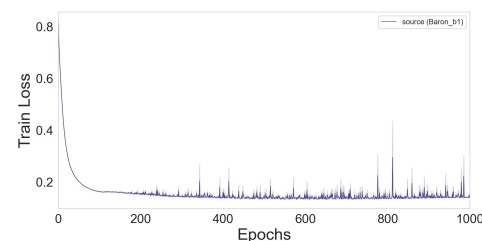

(a) **Linear subspace data** with 5% noisy samples and SNR = 20 [dB].  (b) **Single-cell RNA data** trained on the 'Baron' batch.

Figure 24: Epoch-wise train loss influenced by the amount of **domain shift**.

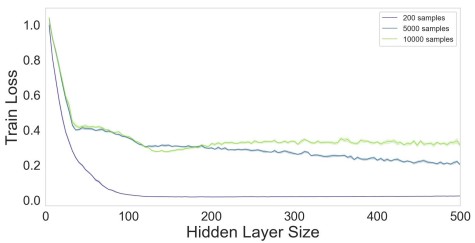

Figure 25: Train loss of the **linear subspace data** with sample noise = 70% and SNR = -15 [dB] for different number of training samples.

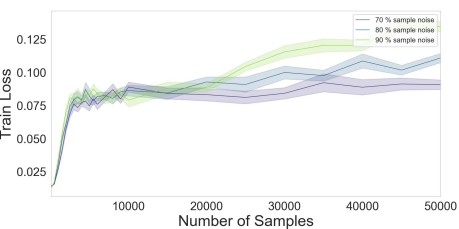

Figure 26: Train loss rises as the number of samples increase for the scenario of sample noise. Similar behavior exists for feature noise and domain shift scenarios. The rise occurs due to the model entering the under-parameterized regime for larger number of training samples. However, despite the increase in training loss with a higher number of samples, the test loss decreases, suggesting improved reconstruction of the test samples.

## D RESULTS FOR FEATURE NOISE

In this section, we present the results for the feature noise scenario. Feature noise adds complexity since each sample contains noise in some of its features. As a result, the model never encounters samples with entirely clean features, making it unable to isolate and focus on clean data. Consequently, the model experiences difficulty in learning the correct data structure. Surprisingly, increasing feature noise actually leads to a decrease in the test loss for the single-cell RNA dataset (Figure 27b). This can also be observed in Appendix E.2, Figure 34b and Appendix E.3, Figure 44b. Moreover, the peak shifts left as the number of noisy features rise in Figure 27a.

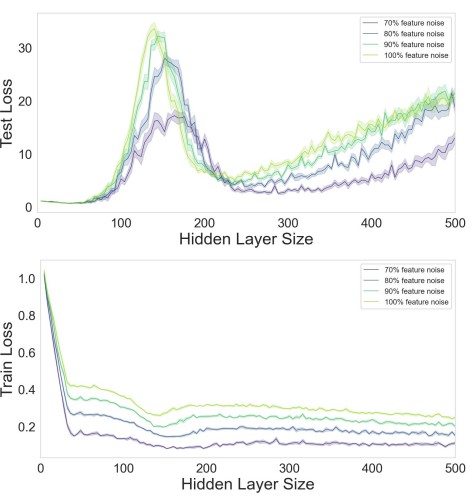
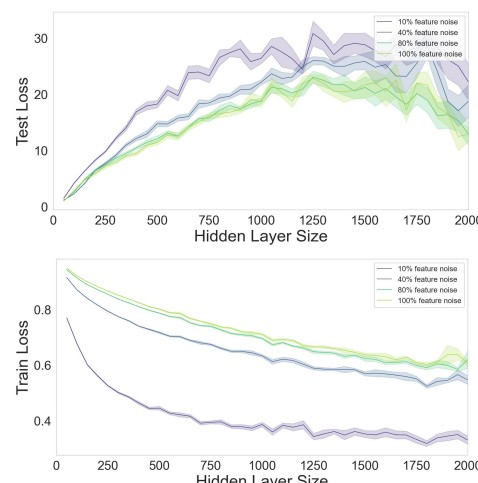

(a) Double descent for the **linear subspace data** trained with SNR = -13 [dB]. The model also exhibits the final ascent phenomenon (Xue et al., 2022).

(b) Non-monotonic behavior for the **Single-cell RNA data** trained with SNR = -12 [dB]. Beyond a hidden layer size of 2000, the test loss continues to decrease, while the train loss increases.

Figure 27: Test loss exhibits model-wise double descent and non-monotonic behaviors for the case of varying **feature noise**.

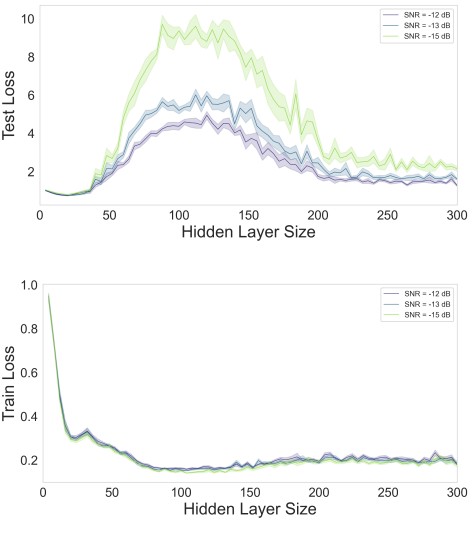
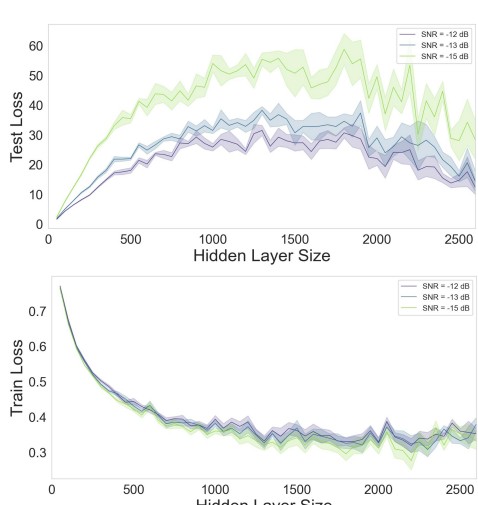

(a) **Linear subspace data** with 40% noisy features. Beyond hidden layer of size 300, the test loss rises.

(b) **Single-cell RNA data** with 10% noisy features. Beyond a hidden layer size of 2600, the test loss continues to decrease, and the train loss increases.

Figure 28: The effect of SNR for the case of **noisy features** on the test loss curve.

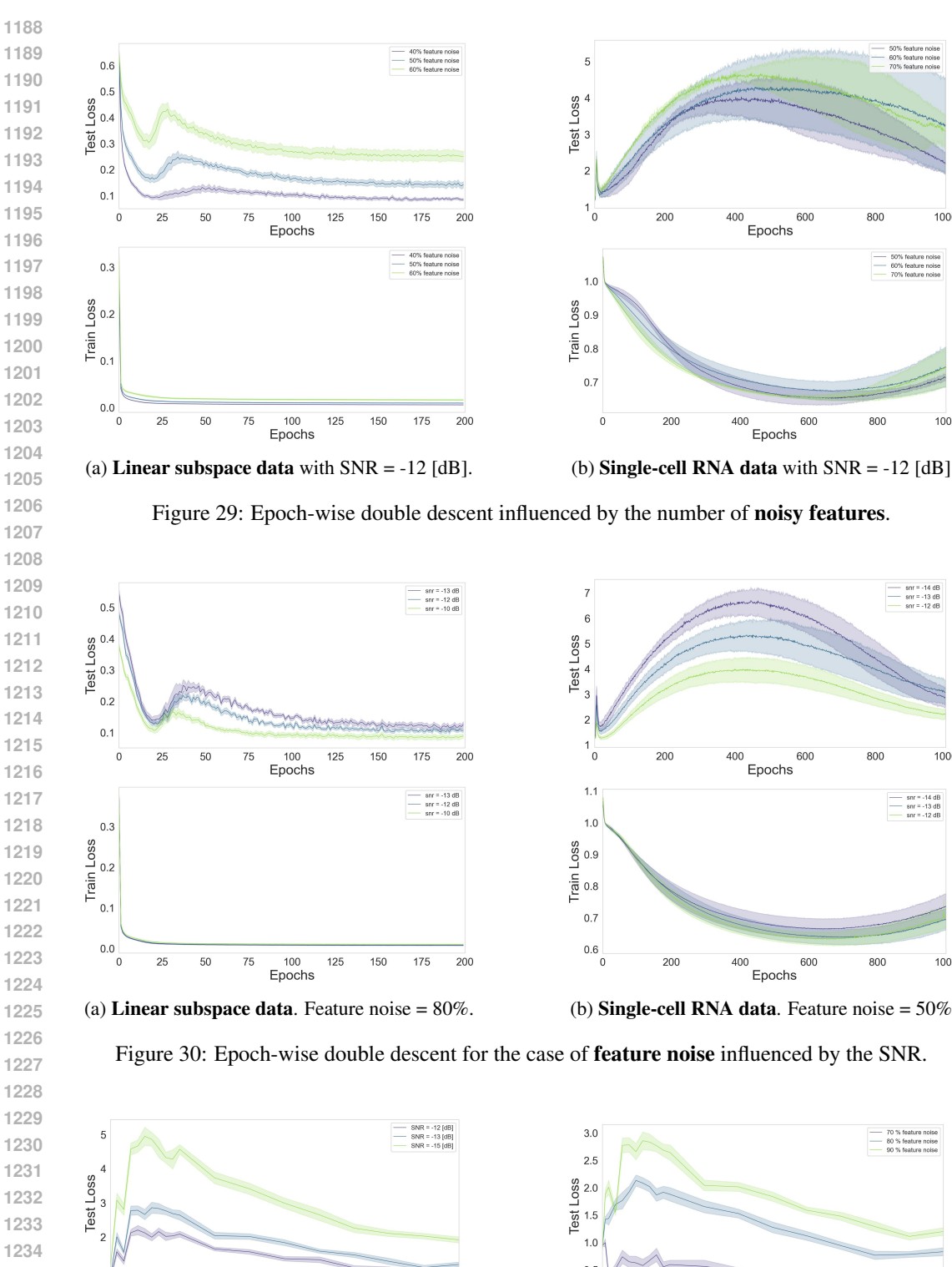

(a) **Linear subspace data** with SNR = -12 [dB].

(b) **Single-cell RNA data** with SNR = -12 [dB].

Figure 29: Epoch-wise double descent influenced by the number of **noisy features**.

(a) **Linear subspace data**. Feature noise = 80%.

(b) **Single-cell RNA data**. Feature noise = 50%.

Figure 30: Epoch-wise double descent for the case of **feature noise** influenced by the SNR.

(a) Sample-wise non-monotonicity for varying **SNR**s in the scenario of **feature noise** = 90 %.

(b) Sample-wise non-monotonicity for varying number of **noisy features**. SNR = -13 [dB].

Figure 31: **Sample-wise** non-monotonicity pattern for the **linear subspace data** for the scenario of feature noise.

# E  ADDITIONAL EXPERIMENTS

## E.1  MORE RESULTS FOR DOMAIN ADAPTATION

This section presents the UMAP visualizations of the different domains for both the linear subspace and single-cell RNA data in Figure 32. Results for different model sizes trained on the linear subspace dataset are also reported in Figure 33.

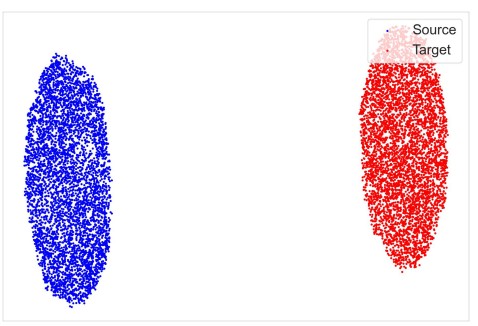
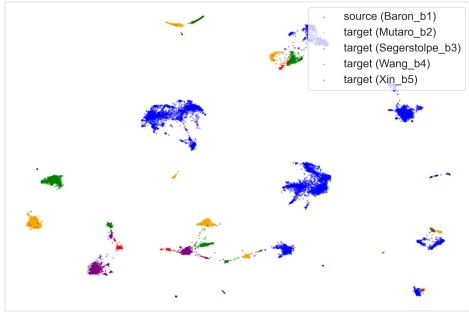

(a) The UMAP representation shows a clear domain shift between the source and target datasets.

(b) Clusters represent different cell types. Different domains are represented by different colors.

Figure 32: UMAP representations of source and target datasets for the linear subspace dataset (left) and single-cell RNA dataset (right).

Figure 33 illustrates the results based on a similar experiment conducted in Section 5.1 for the linear subspace data. As expected, the interpolating models exhibit the poorest KNN-DAT outcomes. Over-parameterized models introduce a decrease in the test loss indicating an improved reconstruction of the target data. In this scenario, we noticed that smaller models perform better than over-parameterized models based on KNN-DAT results. We think that the small size of the hidden layer (4) and the high dimensionality of the dataset (50 features) result in significant information loss in these layers. This could lead to closely clustered vectors in the embedding space, ultimately causing low KNN-DAT results. However, a hidden layer of size 4 indicates insufficient capacity to represent the signal, as shown by the high values of test and train losses in Figure 33.

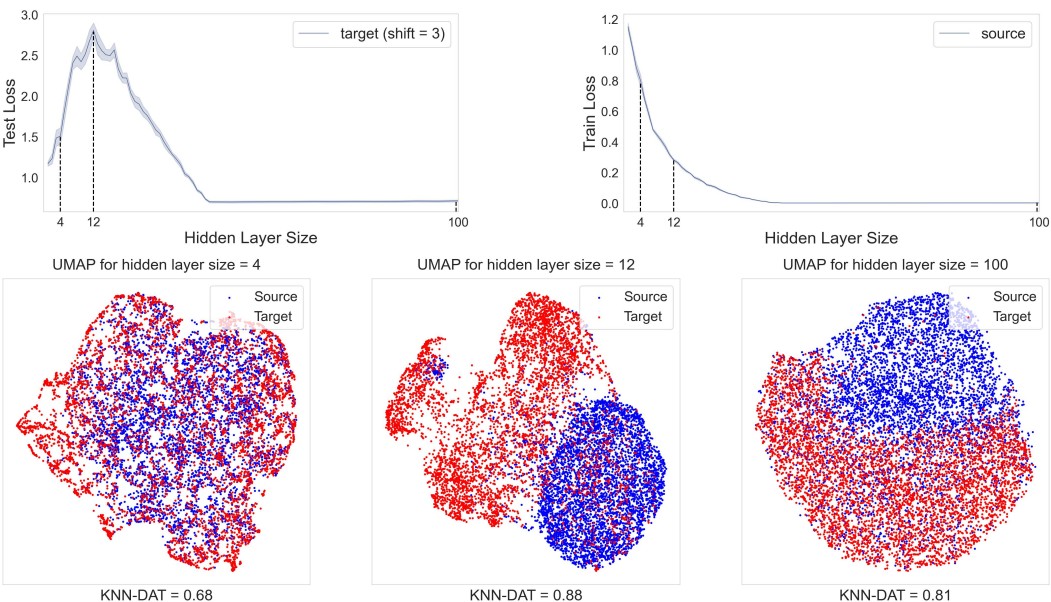

Figure 33: UMAP of the latent (bottleneck) vectors with a size of 45 and KNN-DAT results for different model sizes trained on the **linear subspace** dataset for a shift of 3.

## E.2 DOUBLE DESCENT RESULTS FOR CNNS TRAINED ON MNIST

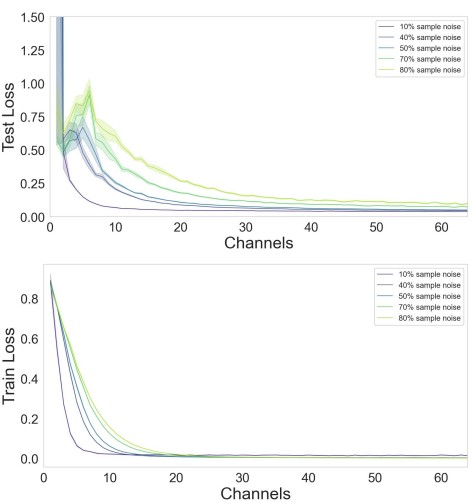
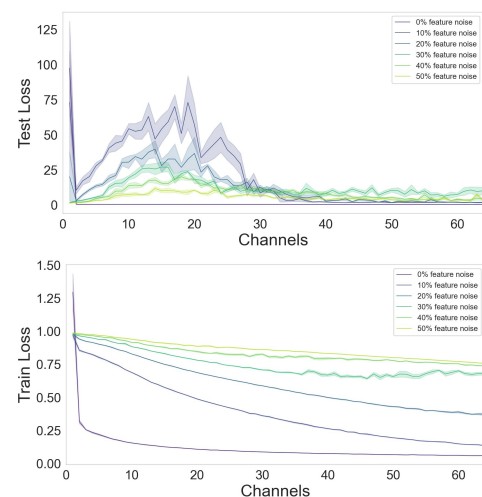

(a) Sample noise scenario with SNR = -17 [dB].

(b) Feature noise scenario with SNR = -20 [dB].

Figure 34: Model-wise double descent for CNNs trained on MNIST with varying levels of sample noise (left) and feature noise (right).

In this section, we demonstrate that the double descent phenomenon can be reproduced in other unsupervised AE architectures. We employed the MNIST dataset (LeCun et al., 1998) and trained under-complete CNNs as detailed in Figure 18. For the case of sample noise, the noise is added to $p \cdot 100\%$ of the images, and for the feature noise scenario, noise is introduced to $p \cdot 100\%$ of the pixels of each image. To demonstrate the phenomenon with the presence of domain shift, the model is trained on the MNIST-M and MNIST datasets and tested on MNIST and MNIST-M, respectively. Results for model-wise double descent for varying amounts of sample and feature noise cases are presented in Figure 34.

In Figure 35, we show the test and train loss results (top two sub-figures) for three different models trained on MNIST with 50% sample noise and an SNR of -15 dB and find out that over-parameterized models can reduce the noise levels in an image. The smallest model, with 3 channels, is under-parameterized. The second model, within the critical regime, with 5 channels, performs poorly, while the third is over-parameterized, containing 60 channels. Interestingly, We noticed that even though our AE was not trained to remove noise (as in denoising AEs (Vincent et al., 2008; 2010)), over-parameterized models were able to reduce noise to some extent. In contrast, models within the critical regime performed significantly worse.

After training, we evaluated each model by feeding it images with varying SNR values and examining the reconstructed outputs (bottom sub-figure in Figure 35). The over-parameterized model produced the best-quality reconstructed images. Following that, the under-parameterized model performed moderately well, and the model in the critical regime generated the noisiest images. This is because the critical model focused on memorizing the noise during training instead of learning the underlying signal, resulting in consistently noisy outputs. In contrast, the over-parameterized model had enough capacity to memorize the noise and learn the signal. While the under-parameterized model cleans the images better than the critical model, it still distorts some details compared to the over-parameterized model due to its limited capacity.

To quantify noise reduction, we used the Peak Signal-to-Noise Ratio (PSNR), a metric that assesses signal quality by comparing the original image to its noisy version. PSNR measures the ratio between the maximum possible value of a signal ($R^2$) and the power of the noise (MSE). Higher PSNR values indicate better quality, meaning less noise. The formula for PSNR is

$$PSNR = 10 \cdot \log\left(\frac{R^2}{MSE(x, f(x+n))}\right),$$

where $x + n$ represents the noisy image ($n$ is the noise), and $x$ is the clean version. This metric, expressed in decibels, allows us to evaluate how well each model cleans the images. As shown,

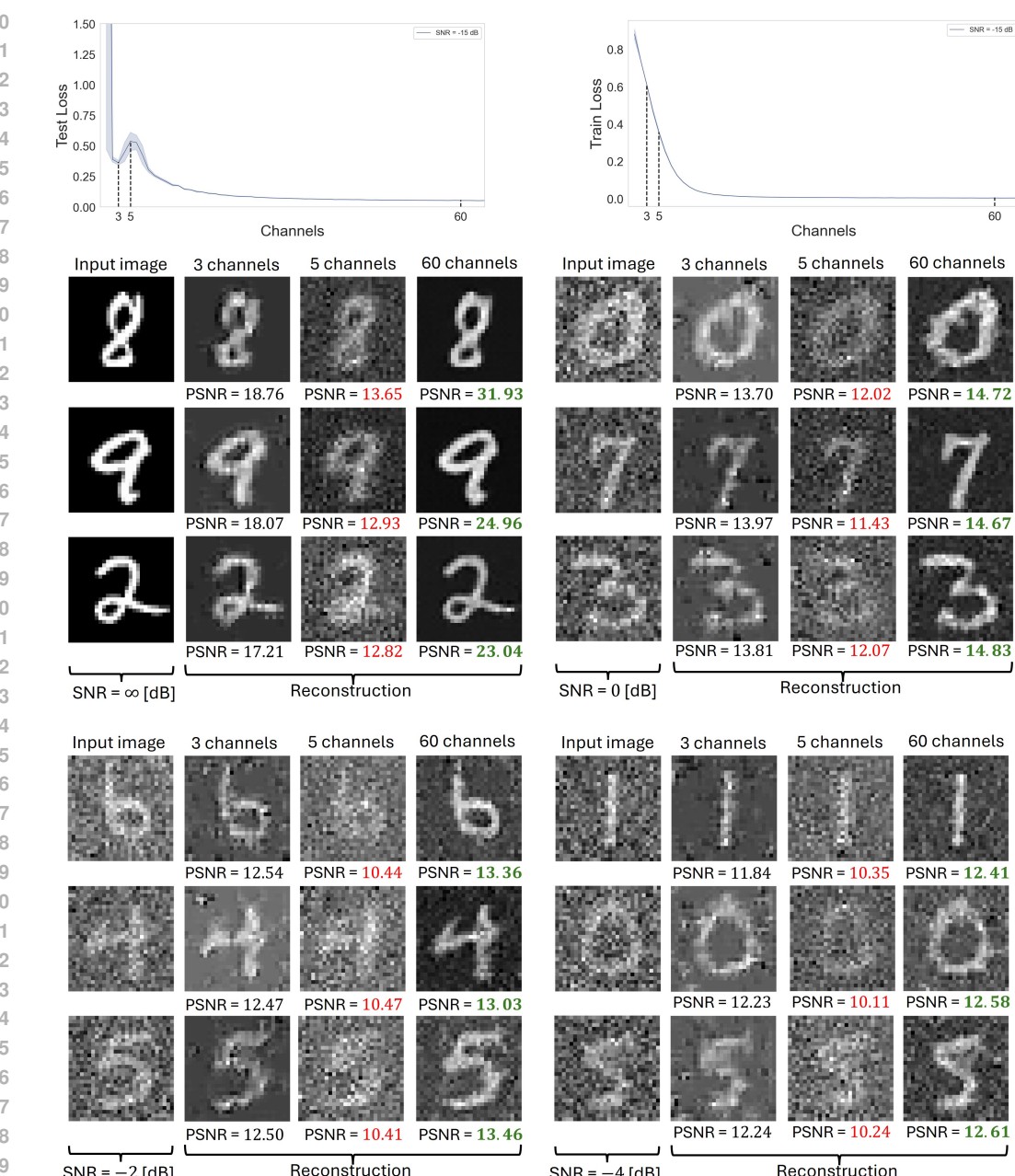

Figure 35: Models trained on 50% noisy MNIST images with SNR = -15 [dB] and tested on MNIST images with different values of SNR.

the over-parameterized model consistently achieves the highest PSNR values (highlighted in bold green), while the poorly interpolating model, which primarily memorized noise, produces the lowest PSNR values (in red). In conclusion, *over-parameterized models are capable of reducing noise when trained on noisy data, even without being explicitly tasked to do so.*

We proceed by illustrating the impact of SNR on the test loss curve for both sample and feature noise scenarios in Figure 36. As expected, the test loss increases for low SNR values. We then investigate the effect of domain shifts between the training and testing datasets in two cases. First, models are trained on the MNIST dataset and tested on the MNIST-M dataset, as shown in Figure 37a. Second, models are trained on MNIST-M and tested on MNIST, as seen in Figure 37b. In both cases, the model-wise double descent curve is observed.

We further illustrate this phenomenon along the epochs axis, displaying non-monotonic behavior and double descent under different levels of sample and feature noise (Figure 38) and showing the

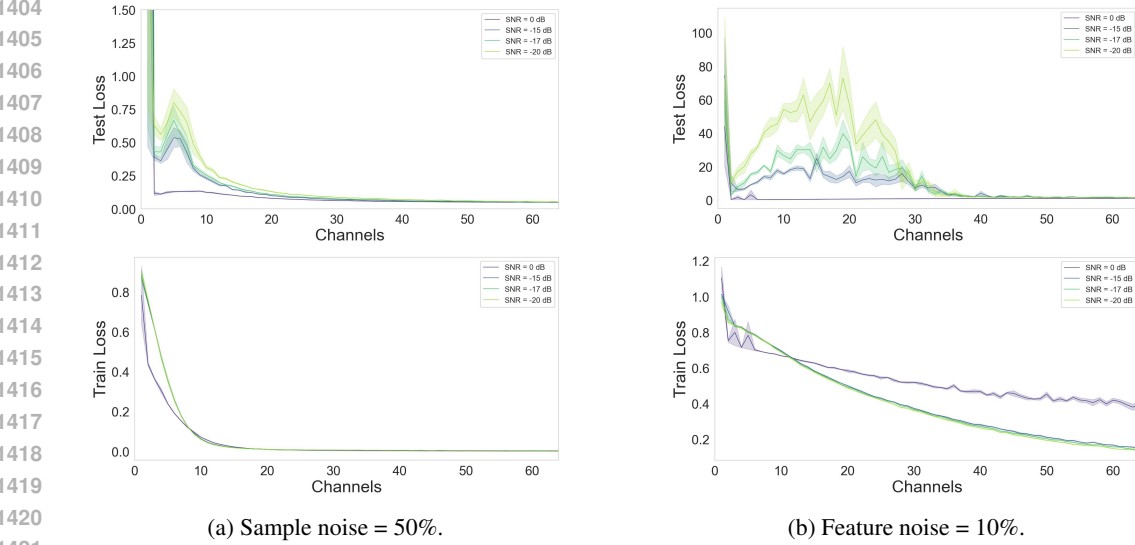

(a) Sample noise = 50%.                         (b) Feature noise = 10%.

Figure 36: Model-wise double descent for CNN trained on MNIST with varying levels of SNRs. Left: sample noise, right: feature noise.

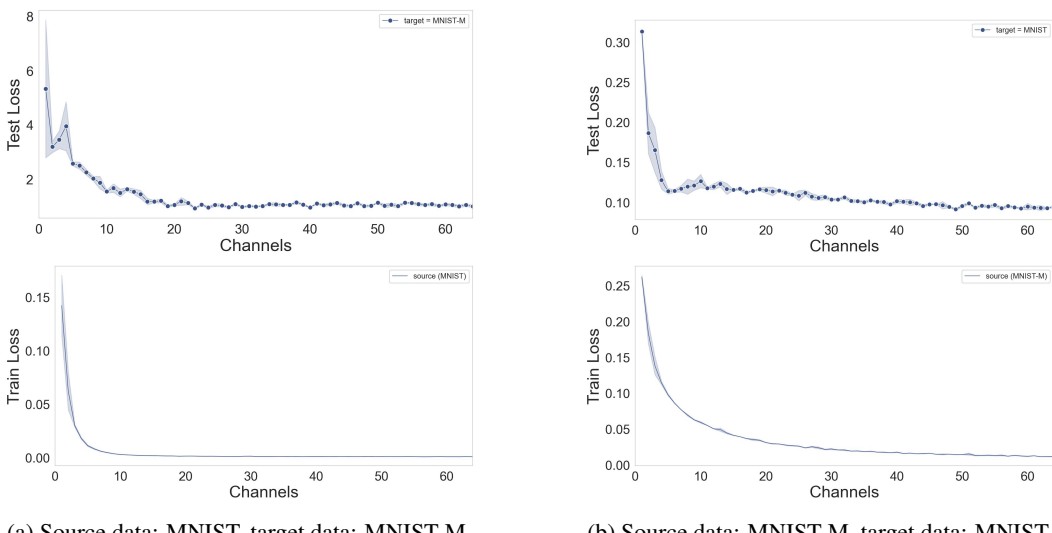

(a) Source data: MNIST, target data: MNIST-M.        (b) Source data: MNIST-M, target data: MNIST.

Figure 37: Model-wise double descent for CNN trained and tested on different domains.

impact of SNR variation (Figure 39). Additionally, we provide similar results under domain shift conditions between the train and test datasets (Figure 40). Sample-wise double descent and non-monotonic behavior is observed as well in all contamination setups. The cases of varying levels of sample noise and feature noise are displayed in Figure 41 and for varying SNRs for both scenarios in Figure 42. Sample-wise double descent is also illustrated in Figure 43 for when a domain shift is present between the training data (MNIST) and the testing data (MNIST-M).

### E.3 DOUBLE DESCENT RESULTS FOR THE NON-LINEAR SUBSPACE DATASET

Building on the linear subspace dataset discussed in Subsection 3.1, we have developed a new dataset with non-linear characteristics to investigate the double descent phenomenon in more complex scenarios. Although the single-cell RNA dataset is already non-linear, we have created this dataset to demonstrate the reproducibility of the double descent phenomenon across various datasets.

As in the linear subspace model discussed in Subsection 3.1, we sample $N$ latent vectors $\{z_i\}_{i=1}^{N}$ from a normal distribution and project them to a higher dimension using a random matrix $D_1$. The key difference is the inclusion of non-linear components $z_i^2$ and $z_i^3$, each projected to a higher

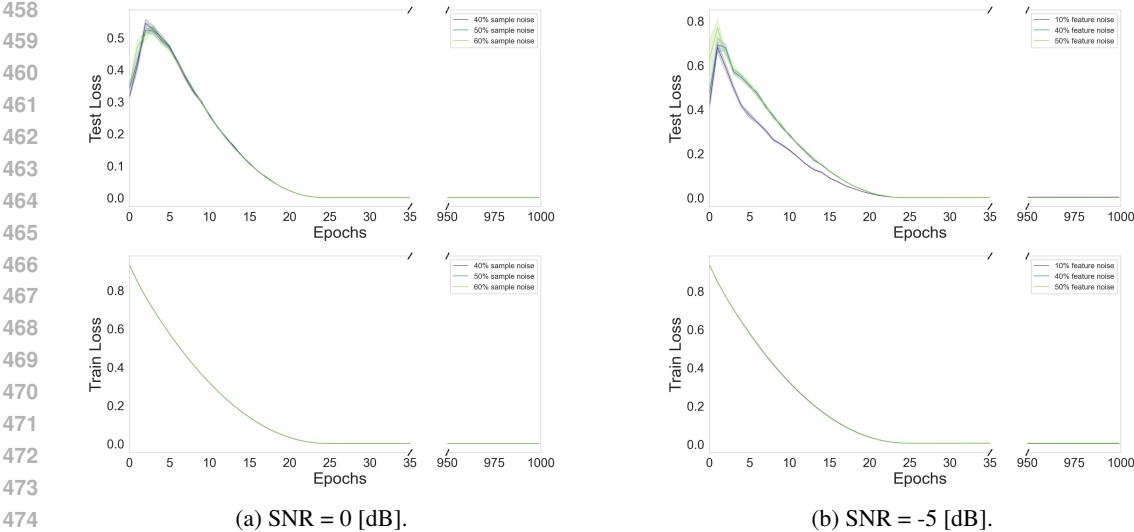

(a) SNR = 0 [dB].
(b) SNR = -5 [dB].

Figure 38: Epoch-wise non-monotonic behavior for varying levels of sample noise (left) and feature noise (right).

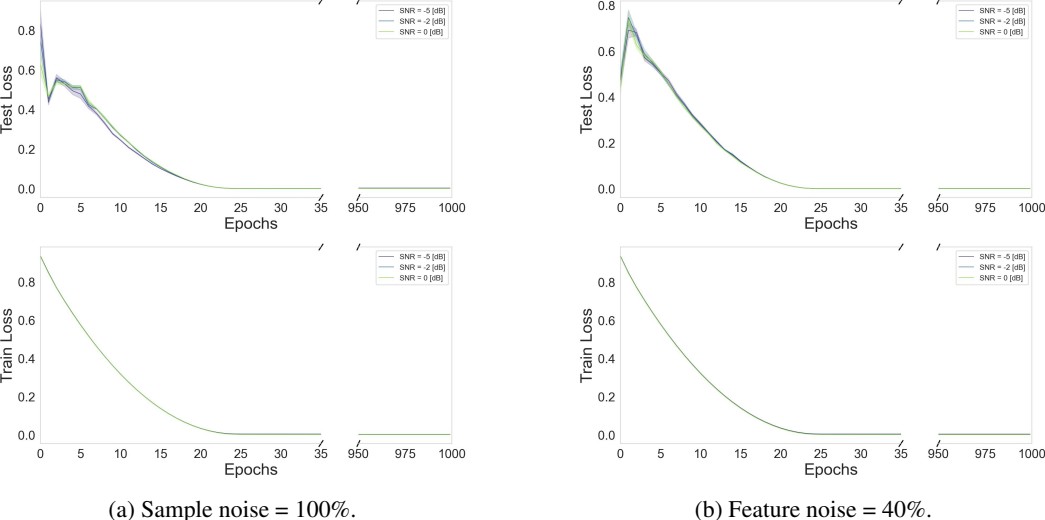

(a) Sample noise = 100%.
(b) Feature noise = 40%.

Figure 39: Epoch-wise double descent and non-monotonic behavior for varying SNRs. Left: sample noise, right: feature noise.

dimensional space with different random matrices $D_2$ and $D_3$. To create contaminated setups of sample and feature noise, noise is added to $p \cdot 100\%$ of the data where $\theta$ controls the SNR:

$$x_i = \begin{cases} \theta(D_1 z_i + D_2 z_i^2 + D_3 z_i^3) + \epsilon_i, & \text{with probability } p, \\ \theta(D_1 z_i + D_2 z_i^2 + D_3 z_i^3), & \text{with probability } 1 - p, \end{cases}$$

For the domain shift scenario, we divide the latent vectors into training and testing sets and use the same parameter 's' as described in Subsection 3.1 to control the shift between the train and test sets in the following manner: $D_i'' = D_i + s \cdot D'$ for $1 \leq i \leq 3$ and get:

$$x_i = \begin{cases} D_1 z_{train}^i + D_2 (z_{train}^i)^2 + D_3 (z_{train}^i)^3, & \text{if } train, \\ D_1'' z_{test}^i + D_2'' (z_{test}^i)^2 + D_3'' (z_{test}^i)^3, & \text{if } test. \end{cases}$$

For anomaly detection, clean samples are represented by $\theta(D_1 z_i + D_2 z_i^2 + D_3 z_i^3)$, with $p \cdot 100\%$ of them replaced by anomalies sampled from a normal distribution, as detailed in Subsection 3.1.

We start by presenting results for the sample and feature noise scenarios as depicted in Figure 44. As shown, the test loss results for the case of sample noise (Figure 44a) resemble those of the linear

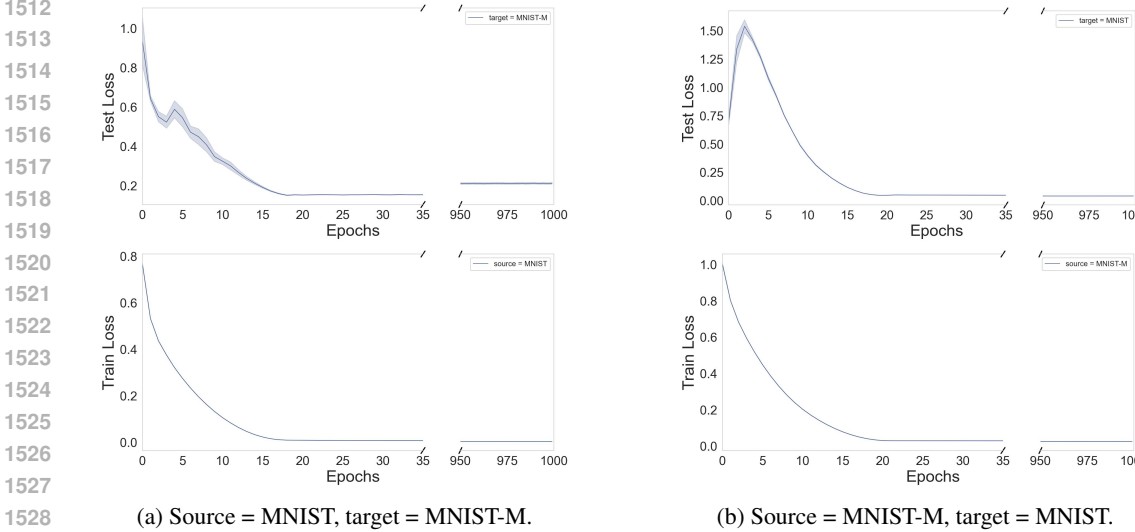

(a) Source = MNIST, target = MNIST-M.  (b) Source = MNIST-M, target = MNIST.

Figure 40: Epoch-wise double descent and non-monotonic behavior for domain shift.

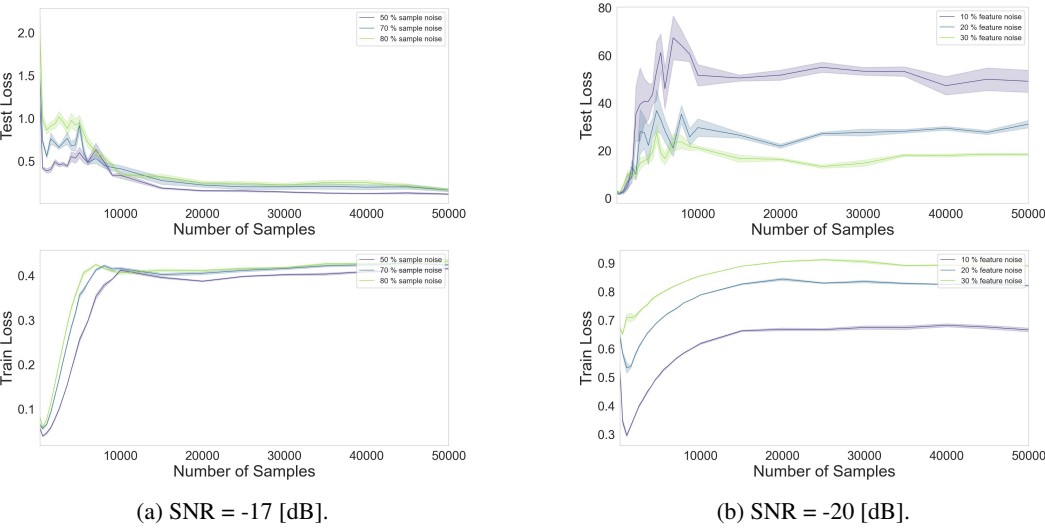

(a) SNR = -17 [dB].  (b) SNR = -20 [dB].

Figure 41: Sample-wise double descent and non-monotonic behavior for varying levels of sample noise (left) and feature noise (right).

subspace data model presented in Figure 3a. Figure 44b demonstrates the model-wise final ascent phenomenon for the case of feature noise as elaborated in Appendix E.4. Figure 45 shows how the SNR affects the test loss curve for both sample and feature noise cases. As observed, the test loss increases with decreasing SNR. Additionally, the final ascent in the test loss is depicted in 45b for the feature noise scenario, where the slope becomes steeper as the SNR decreases. We also demonstrate the double descent and final ascent results regarding the domain shift scenario in Figure 46 and the anomaly detection capabilities in Figure 47.

We also observed epoch-wise double descent and non-monotonic behavior for this dataset, as shown in Figure 48 for different percentages of sample and feature noise and in Figure 49 for varying SNR levels under the same noise conditions. Additionally, epoch-wise double descent is also observed when a domain shift is present between the train and test sets, as depicted in Figure 50. Instances of sample-wise double descent and non-monotonic curves are also reported and displayed in Figure 51 for varying levels of sample and feature noise, Figure 52 for varying levels of SNR, and in Figure 53 for domain shift.

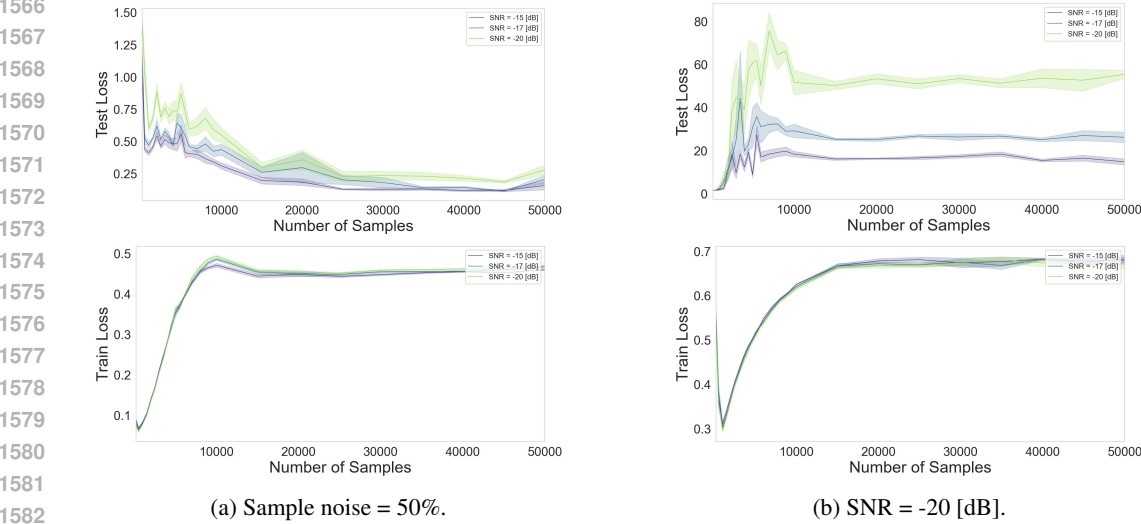

(a) Sample noise = 50%.  (b) SNR = -20 [dB].

Figure 42: Sample-wise double descent and non-monotonic behavior for varying levels of SNR. Left: sample noise, right: feature noise.

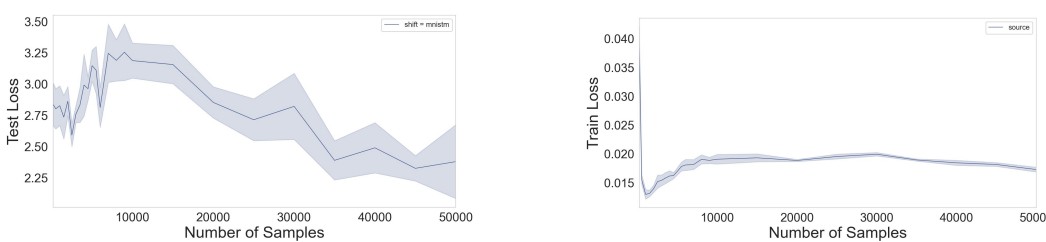

Figure 43: Sample-wise double descent for models trained on the MNIST dataset and tested on the MNIST-M dataset.

### E.4 FINAL ASCENT PHENOMENON

While training various models on different datasets contaminated with sample and feature noise at different SNR levels and domain shifts between train and test sets, we observed a final ascent phenomenon characterized by a pattern of decreasing-increasing-decreasing-increasing test loss. The phenomenon was first observed in Xue et al. (2022) in supervised learning with label noise. We suspect a potential connection to this phenomenon in unsupervised learning, which we have yet to fully analyze. We refer to Figure 55a, which illustrates the final ascent results for the linear subspace dataset under extreme conditions of 100% sample noise, as a continuation of Figure 3a. We also present the final ascent results for the single-cell RNA dataset in Figure 55b. Another instance of final ascent with the presence of varying feature noise is illustrated in Figure 27a for the linear subspace dataset and in Figures 44b, 45b for the non-linear subspace dataset. Results are also replicated using the non-linear subspace dataset under various domain shifts, as observed in Figure 46.

### E.5 MULTIPLE DESCENTS UNDER DIFFERENT NOISE TYPES AND SPARSE AEs

This section explores the emergence of double and triple descent for noise distributions beyond Gaussian noise and sparse AEs. Figure 55 illustrates the phenomenon for the linear subspace and single-cell RNA datasets when subjected to Laplacian noise. The experimental setup mirrors that of Figure 3. As shown, both datasets exhibit similar results under these conditions.

We extend our research to recent applications of AEs, including sparse AEs, which are increasingly utilized in explainable AI (XAI) (Gao et al., 2024) and have been adopted by Google in their Gemini project. Using sparse CNN AEs, we trained models on the MNIST dataset containing 80% noisy samples and observed the emergence of double descent. The models were configured with a bottleneck layer of size 550, and the parameter $k$, determining the top $k$ highest bottleneck values to retain, was set to 500. The results are illustrated in Figure 56.

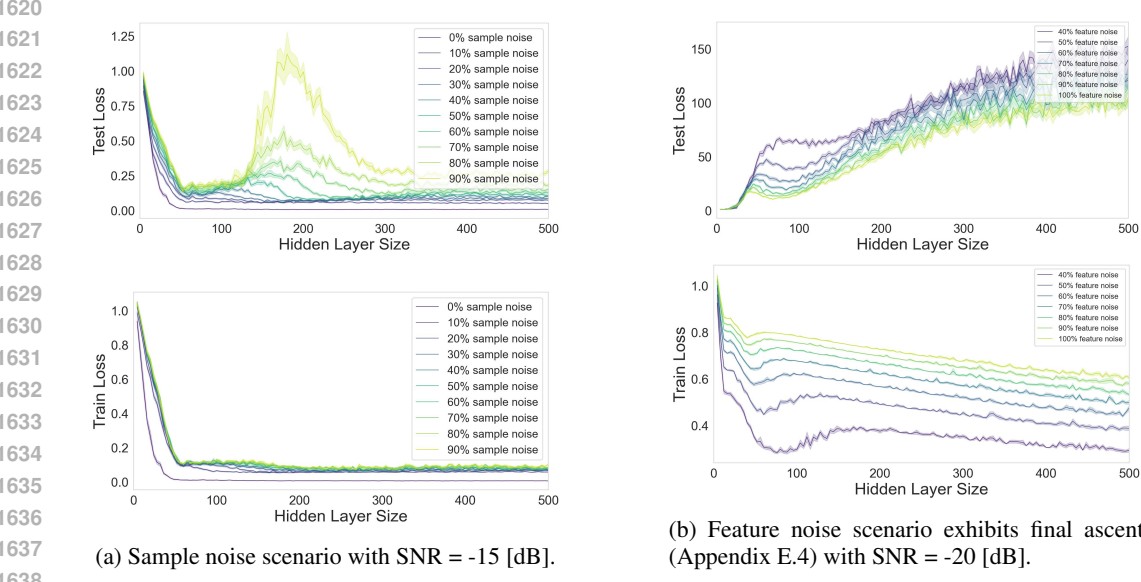

(a) Sample noise scenario with SNR = -15 [dB].

(b) Feature noise scenario exhibits final ascent (Appendix E.4) with SNR = -20 [dB].

Figure 44: Model-wise double descent for the non-linear subspace data with varying levels of sample noise (left) and final ascent with varying levels of feature noise (right).

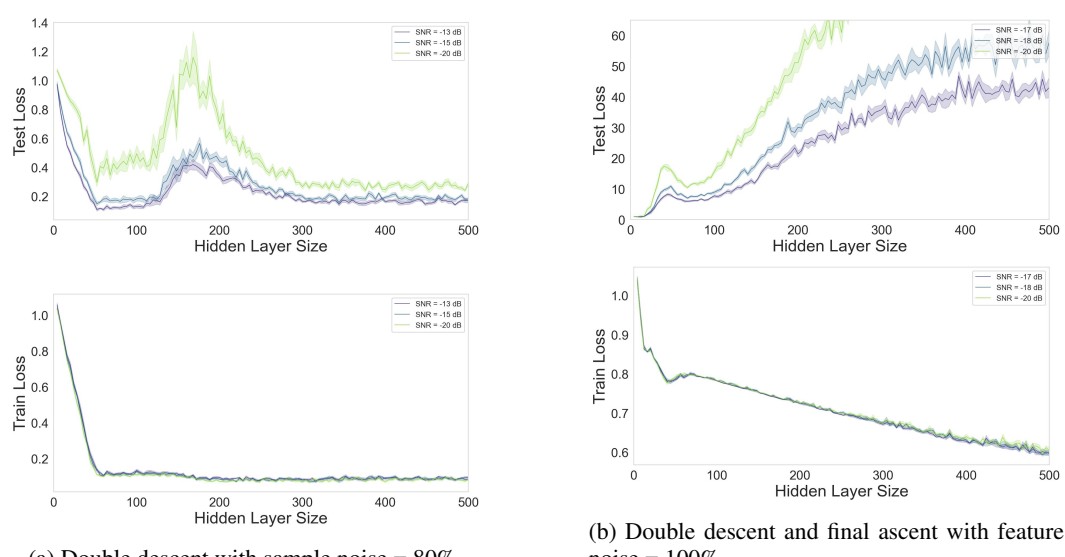

(a) Double descent with sample noise = 80%.

(b) Double descent and final ascent with feature noise = 100%.

Figure 45: Effect of SNR on the test loss curve as a function of model size. Left: **sample noise** scenario. Right: **feature noise** scenario.

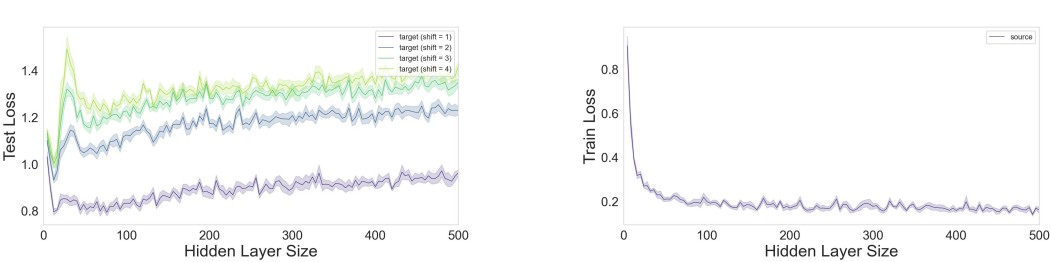

Figure 46: Model-wise double descent and final ascent for the scenario of domain shift.

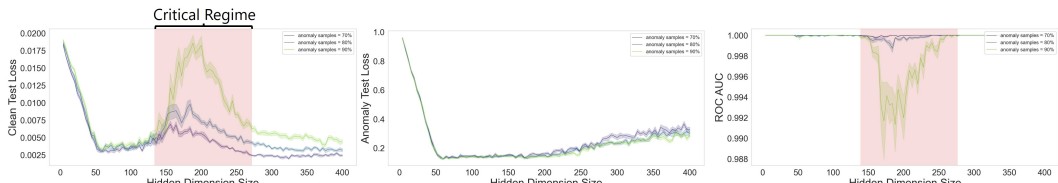

Figure 47: Non-linear anomaly data with SAR = -15 [dB]. **Left:** test loss of the clean samples. A double descent pattern emerges for low SARs and high anomaly presence in the training data. **Middle:** test loss of the anomaly data. **Right:** Non-monotonic behavior of the ROC-AUC.

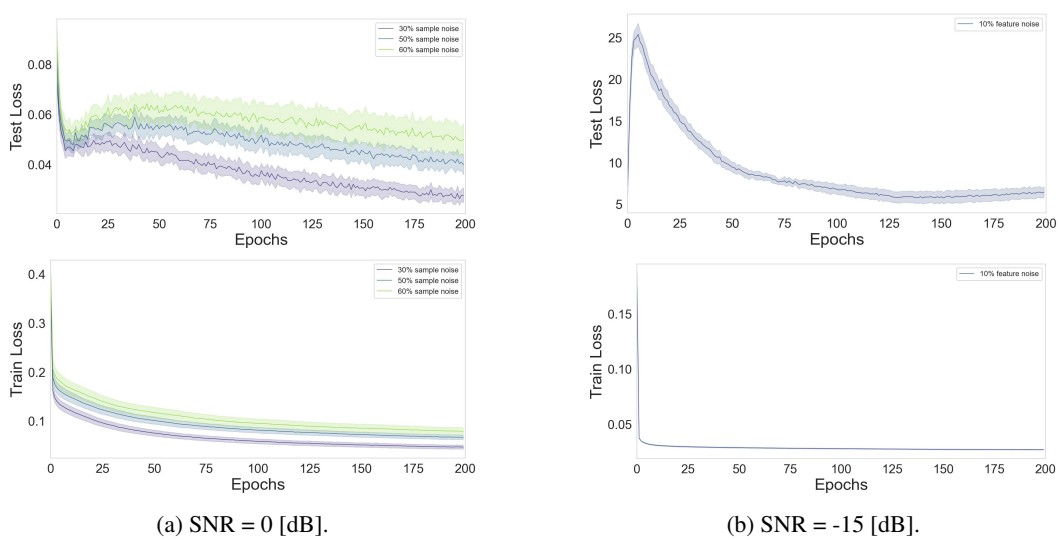

(a) SNR = 0 [dB].

(b) SNR = -15 [dB].

Figure 48: Epoch-wise double descent and non-monotonic behavior for varying levels of sample noise (left) and feature noise (right). For the scenario of feature noise, we mostly noticed the non-monotonic curve at 10%.

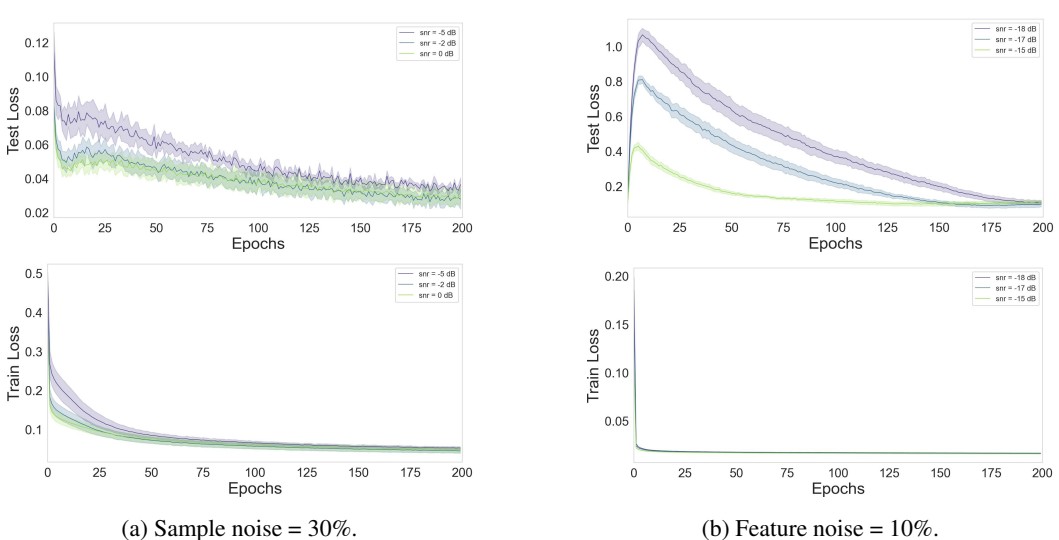

(a) Sample noise = 30%.

(b) Feature noise = 10%.

Figure 49: Epoch-wise double descent and non-monotonic behavior for varying levels of SNR. Left: sample noise, right: feature noise.

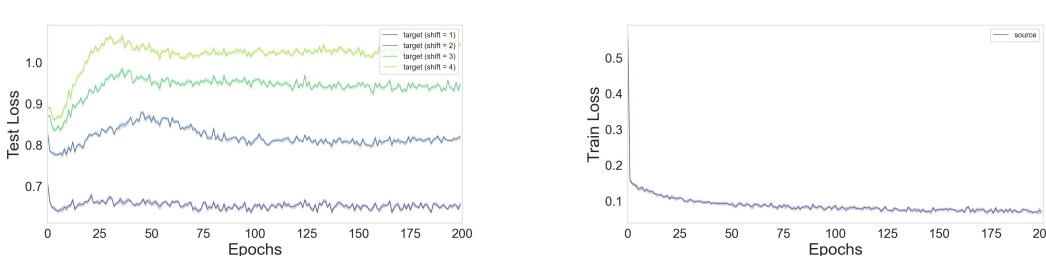

Figure 50: Epoch-wise double descent for when a domain shift is present between train and test sets.

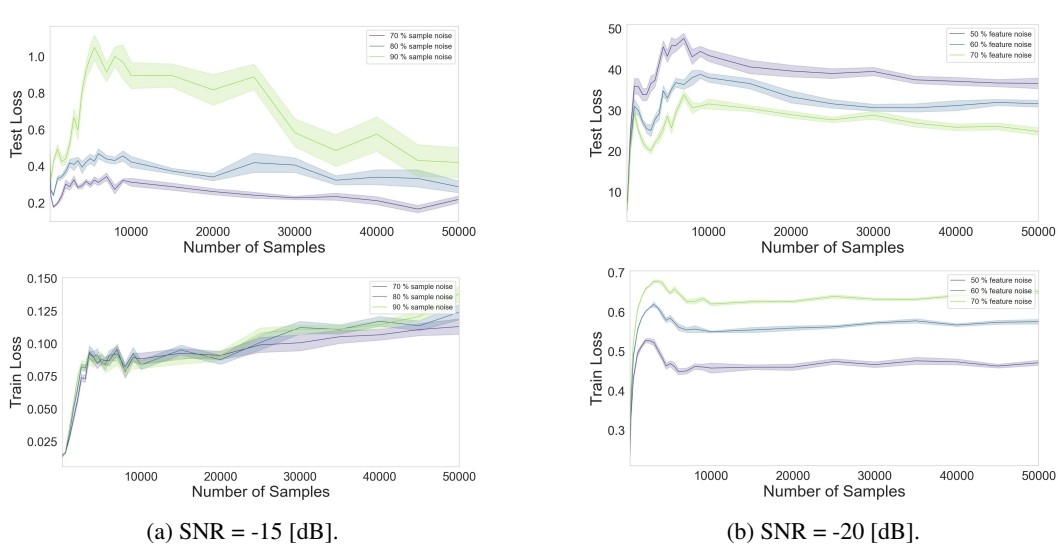

(a) SNR = -15 [dB].

(b) SNR = -20 [dB].

Figure 51: Sample-wise double descent and non-monotonic behavior for varying levels of sample noise (left) and feature noise (right).

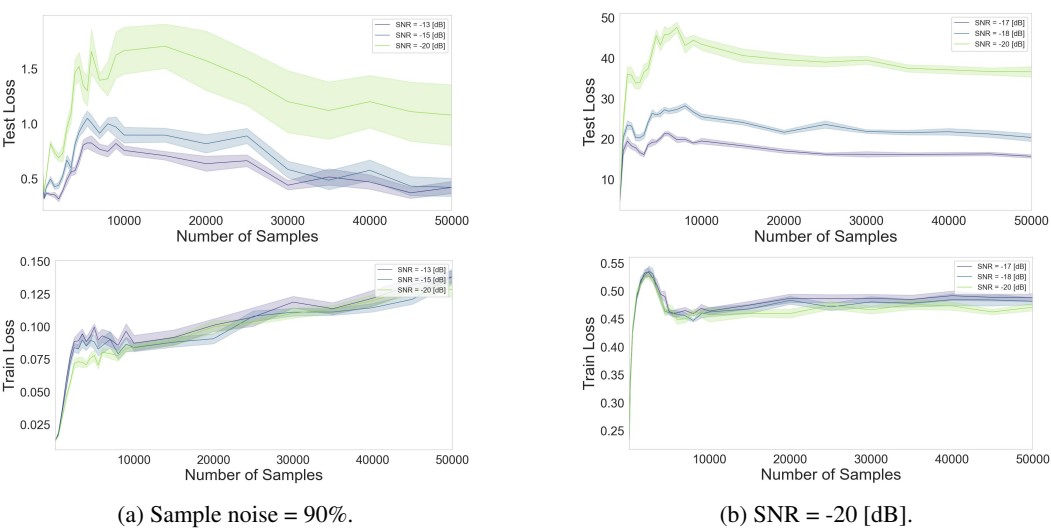

(a) Sample noise = 90%.

(b) SNR = -20 [dB].

Figure 52: Sample-wise double descent and non-monotonic behavior for varying levels of SNR. Left: sample noise, right: feature noise.

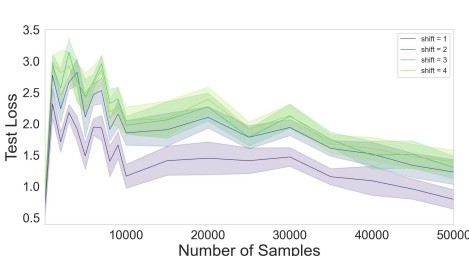 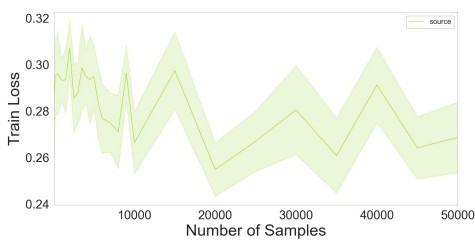

Figure 53: Sample-wise non-monotonic behavior for the domain shift scenario.

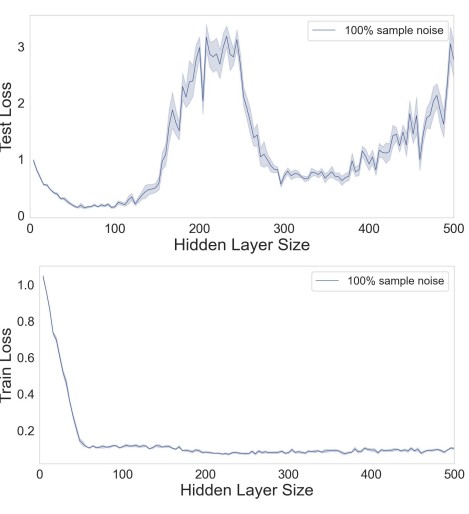 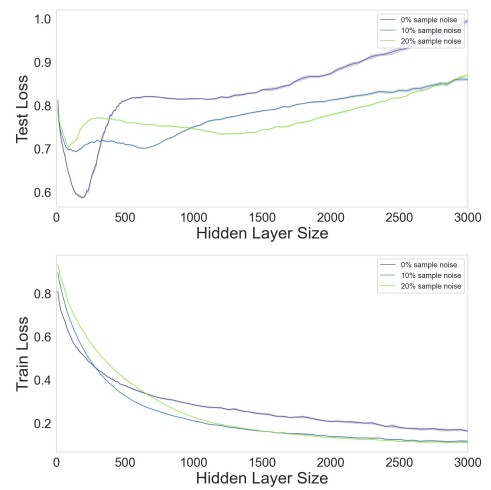

(a) **Linear subspace data** and SNR = -15 [dB].  (b) **Single-cell RNA data** and SNR = -10 [dB].

Figure 54: Test loss exhibits model-wise double descent followed by a final ascent for the scenario of varying **sample noise**.

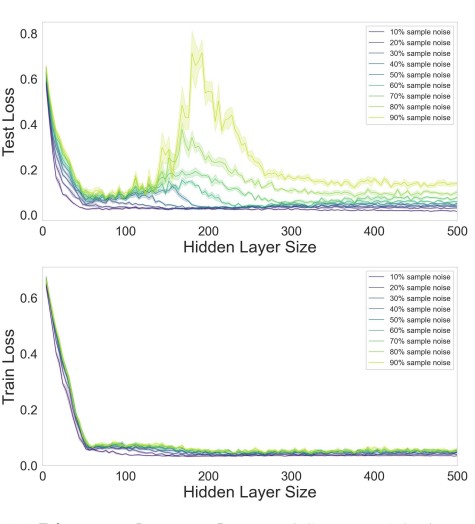 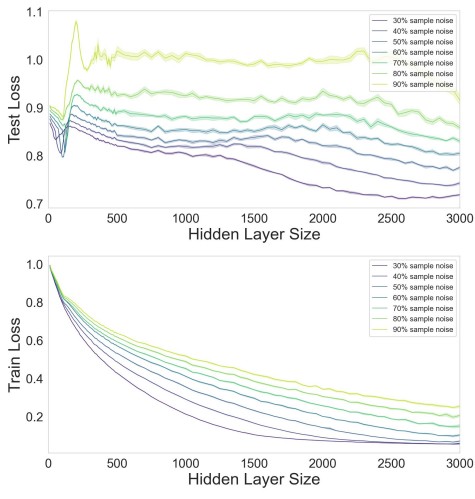

(a) **Linear subspace data** and SNR = -15 [dB].  (b) **Single-cell RNA data** and SNR = -17 [dB].

Figure 55: Test loss exhibits model-wise double descent for the case of Laplace noise.

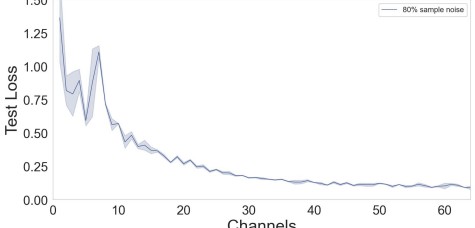 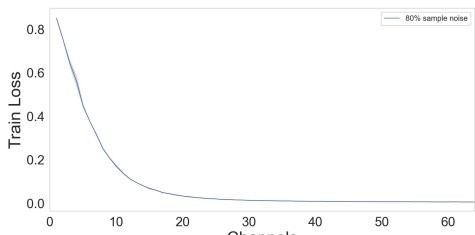

Figure 56: Test loss exhibits model-wise double descent for sparse CNN AEs trained on MNIST with bottleneck layer size of 550 and k = 500.

