# OpenReview forum: "Multiple Descents in Unsupervised Auto-Encoders: The Role of Noise, Domain Shift and Anomalies"
_ICLR.cc/2025/Conference — Submitted to ICLR 2025_

### Official Review · Reviewer_BmzS · 2024-10-28

**Soundness:** 4
**Presentation:** 3
**Contribution:** 3
**Rating:** 3
**Confidence:** 4

**Summary:**

The authors study the phenomenon of double descent appearing in over-parametrized machine learning models. Where recent findings focused on supervised learning problems, here the authors focus on unsupervised learning specifically studying autoencoders. Considering the evolution of training loss with respect to different data modifications (sample noise, feature noise, domain shift, and anomalies), they find multiple descent phenomena appearing under different settings, yielding insights into the behaviour of learning of autoencoders studying both synthetic and real data.

**Strengths:**

- The presented paper shows some rigorous analysis of the double descent phenomenon under different settings
- The paper is well written and easy to follow.

**Weaknesses:**

- The takeaway of this paper is not exactly clear. What can we do with these insights on Autoencoders, also given that the CNN AE -- probably the more relevant model in nowadays application -- presented in the appendix does not show the typical double descent with progressing training? The discovered connection between model size and performance does not seem so surprising to me. An extensive discussion and recent application of AE s would strongly benefit the paper. As a constructive suggestion, the authors could look into Sparse Autoencoders, which recently are in heavy use in XAI [1] and even have been adapted by Google in their Gemini project [2], and insights into their training dynamics would be of great use. Note that a smaller scale example on a standard vision benchmark could suffice here.
- The paper lacks a more theoretical discussion of *why* what we see is happening. The recent works shedding light onto this and showing that double descent is not in conflict with classical statistical learning theory could serve as a starting point [3,4] and should also be appropriately addressed in intro and related work.

[1] L. Gao et al., Scaling and evaluating sparse autoencoders (https://cdn.openai.com/papers/sparse-autoencoders.pdf)

[2] https://deepmind.google/discover/blog/gemma-scope-helping-the-safety-community-shed-light-on-the-inner-workings-of-language-models/

[3] A Curth* et al., A U-turn on Double Descent: Rethinking Parameter Counting in Statistical Learning NeurIPS 2023

[4] A Curth* Classical Statistical (In-Sample) Intuitions Don’t Generalize Well: A Note on
Bias-Variance Tradeoffs, Overfitting and Moving from Fixed to Random Designs arXiv 2024

*no relation to the reviewer

**Questions:**

see weaknesses

---

> ### Author Response · Authors · 2024-11-20
> **Response to the reviewer**
>
> We appreciate the reviewer for highlighting the main strengths of our paper, especially our rigorous analysis of the double descent phenomenon under different settings and the fact that the paper is well-written and easy to follow.
>
> **Weakness 1:**
>
> We conducted extensive empirical research to examine how a model's behavior changes with variations in its size, number of training epochs, and training samples. This investigation demonstrated the impact of factors such as noise, domain shift, and anomalies on the test loss curve. Our findings revealed, for example, that over-parameterized models are more effective in minimizing distribution shifts and detecting anomalies, even without being explicitly trained for these tasks.
>
> In terms of vision tasks using CNNs, we believe tabular data offers fascinating opportunities for research, which remains relatively underexplored. We appreciate the suggestion to include recent applications of autoencoders (AEs), such as sparse AEs. As a response, we have added a discussion in Appendix E.5, lines 1615-1619 and Figure 56 showcasing the emergence of double descent in sparse AEs trained on noisy MNIST samples. Due to time constraints, we have verified results specifically for 80% sample noise, with a bottleneck size of 550 and \(k = 500\). Further experiments are underway to evaluate different noise levels, bottleneck sizes, and values of \(k\).
>
> Additionally, we have incorporated the citations recommended in the review. Thank you for your valuable suggestions, which have enriched our work.
>
> **W2 Theoretical results:**
>
> We appreciate the reviewer's insightful comments. We would like to emphasize that our work presents several novel findings that should be regarded as independent contributions to the field. Specifically, we have identified and thoroughly investigated multiple descent phenomena in the context of deep unsupervised learning, which had not been previously observed or systematically analyzed. While the concept of double descent has gained attention in supervised learning, its extension to unsupervised learning settings—particularly in autoencoders—represents a significant and original contribution. Our experiments revealed model-wise, epoch-wise, and sample-wise multiple descent behaviors under various conditions, including noisy data, domain shifts, and anomalies. These observations challenge existing assumptions about over-parameterization in unsupervised tasks and open new avenues for understanding the dynamics of generalization within this domain.
>
> In addition to empirically discovering these phenomena, we demonstrated their practical implications using real-world datasets. For example, over-parameterized autoencoders not only improved reconstruction performance but also showed considerable utility in downstream applications such as anomaly detection and domain adaptation. These findings highlight the real-world relevance of our work and its potential to inform the design of robust unsupervised models in complex data environments. While we acknowledge that a deeper theoretical analysis of these observations is a valuable direction for future research, the novelty and scope of our empirical contributions provide a strong foundation for advancing the understanding of over-parameterized models in unsupervised learning.
>
> However, as suggested by the reviewer, the theoretical foundations of these observations deserve further exploration. Building on the works cited in [3] and [4], future research could formalize the conditions under which multiple descent occurs in unsupervised settings. For instance, extending the parameter-counting frameworks from [3] to unsupervised tasks might reveal how the effective number of parameters changes with varying data distributions and model architectures (or latent dimension of the AE). Additionally, by leveraging insights from [4] regarding the limitations of the bias-variance tradeoff in fixed designs, theoretical studies could examine how random designs or data perturbations affect generalization and descent behaviors in unsupervised models. We have included these citations and ideas as potential directions for future work in the conclusions section, line 539.

---

> > ### Comment · Reviewer_BmzS · 2024-11-21
> > **Answer to rebuttal**
> >
> > Thank you for your response.
> >
> > Regarding Weakness 1:
> > It is still unclear to me what we can *do* with these findings. As of now, this result is just an observation. Given the observed behaviour under domain shift etc, how can we prevent this from happening, or leverage it for our benefit in training?
> >
> > The SAE results are unfortunately on MNIST, which as been extensively discussed as a problematic dataset in the community. I appreciate the effort, yet need to see more complex (standard) benchmark data for conclusions.
> >
> > Moreover, tabular data is for sure not the main interest for the neural network community, with a whole line of papers showing that classical statistical learning, especially (gradient boosted) tree-based methods, are still on par or even outperform NNs on tabular data*.
> >
> > * see e.g. Y. Gorishnyi et al. Revisiting deep learning models for tabular data NeurIPS 2021,
> > L. Grinsztajn et al. Why do tree-based models still outperform deep learning on typical tabular data? NeurIPS 2022
> > R. Shwartz-Zivm, A. Armon. Tabular data: Deep learning is not all you need. Information Fusion, 81:84–90, 2022
> >
> > Regarding Weakness 2:
> > The original double descent phenomena came as surprise to the community. Here, especially with all the preceding work that can serve as a blueprint, more is needed to appropriately discuss the new findings - after all the double descent was to be expected to happen in other settings, as it has been discussed for different model types.
> >
> > My suggestion to the authors is to consider (1) classical settings of neural networks with the corresponding benchmarks (e.g., vision or language), and (2) properly discuss their work in the context of what is known. Given all previous work in the field, the simple observation of double descent in unsupervised learning (also, in tabular data only) in my opinion does not warrant for a paper on its own.

---

> > > ### Author Response · Authors · 2024-11-21
> > > **Answer to response**
> > >
> > > We thank the reviewer for engaging in the discussion. Below we address the new comments:
> > >
> > > **“As of now, this result is just an observation.“**
> > >
> > > To the best of our knowledge, systematically demonstrating scientific phenomena (or observations) is a cornerstone of academic research. Our findings on multiple descents in unsupervised auto-encoders provide actionable insights for designing better architectures for anomaly detection and handling domain shifts.
> > >
> > > We demonstrate that over-parameterized models, when trained correctly, exhibit better generalization in the presence of noise and domain shifts. This implies that architectural designs that prioritize over-parameterization, combined with noise-aware loss functions or adaptive training methods, can improve robustness. Additionally, our results on anomaly detection underscore the significance of selecting an appropriate hidden dimension size for accurately identifying anomalies. These insights offer practical strategies for leveraging the observed behaviors to improve model performance in real-world scenarios involving noise and distribution shifts.
> > >
> > > **The SAE results are, unfortunately on MNIST, which as been extensively discussed as a problematic dataset in the community. I appreciate the effort, yet need to see more complex (standard) benchmark data for conclusions.**
> > >
> > > While there are arguments against using MNIST for evaluating classification models, we are unaware of any scientific critiques regarding its validity for unsupervised learning with autoencoders. MNIST remains a widely accepted benchmark for testing foundational concepts and behaviors in this context. In fact, MNIST has been used in multiple recent publication for evaluation of intepretability using sparse neural network (an AEs) [A,B,C,D]. This is exactly the task that the reviewer pointed out to: “could look into Sparse Autoencoders, which recently are in heavy use in XAI [1] “. If the reviewer is aware of academic papers or studies specifically questioning the suitability of MNIST for our reconstruction task, we would greatly appreciate such references to better contextualize and address these concerns.
> > >
> > > [A] Balın, M.F.,  et al., 2019, May. Concrete autoencoders: Differentiable feature selection and reconstruction. In International conference on machine learning (pp. 444-453). PMLR.
> > >
> > > [B] Kim, C., et al., Discovering Features with Synergistic Interactions in Multiple Views. In Forty-first International Conference on Machine Learning. 2024
> > >
> > > [C] Jethani, N., et al. "Have We Learned to Explain?: How Interpretability Methods Can Learn to Encode Predictions in their Interpretations." International Conference on Artificial Intelligence and Statistics. PMLR, 2021.
> > >
> > > [D] Svirsky, J., et al.. Interpretable Deep Clustering for Tabular Data. In Forty-first International Conference on Machine Learning. 2024
> > >
> > > Additionally, we would like to emphasize that our decision to use MNIST was guided by practical constraints and prior reviewer feedback. Conducting double descent experiments, which require training a large number of models across varying configurations, is computationally intensive and not feasible for larger image datasets within the scope and timeline of this work. To address concerns about dataset complexity, we specifically followed the reviewer’s earlier suggestion: **“Note that a smaller scale example on a standard vision benchmark could suffice here.”**
> > >
> > > Using MNIST allowed us to systematically explore the double descent phenomenon under various noise levels. The results provide valuable insights into the behavior of unsupervised autoencoders, which can be further validated on larger datasets in future work. We hope this clarifies the rationale behind our dataset choice and the feasibility constraints of our experimental design. Additionally, we will work towards adding results on CIFAR-10.

---

> > > > ### Comment · Reviewer_BmzS · 2024-11-21
> > > > **Answer pt 1**
> > > >
> > > > Again, I do not understand the motivation of why to study this double descent, as other reviewers also point out. What you carry over to practice is only indirectly related to the double descent. For example, "over-parametrized model exhibit better performance, when trained correctly, exhibit better generalization in the presence of noise and domain shifts." where does the double, descent come in to play here, especially the second descent? And what about the main, classical double descent? As also other reviewers pointed out, the defined setting of domain-shift and noise seems a bit arbitrary and does not necessarily generalize.
> > > > And again, why does it not happen in CNNs?
> > > >
> > > > MNIST is not anymore an accepted *image benchmark* in Computer Vision, as its easily solved by even very small networks and by memorization, not requiring an image structure. In fact, the papers you bring up here are treating the images as linearized input. As you correctly cited, I asked for “...a smaller scale example on a standard **vision** benchmark...” as vision and language models are the (primary) area of application of SAEs.

---

> > > > > ### Author Response · Authors · 2024-11-21
> > > > > **Thanks for the response**
> > > > >
> > > > > We sincerely thank the reviewer for their engagement with our work. However, we would like to gently point out a few inaccuracies in the response, which may indicate that some key information from our paper was not fully considered.
> > > > >
> > > > > **where does the double, descent come in to play here, especially the second descent? And what about the main, classical double descent?**
> > > > >
> > > > > There is no such thing as a “classical double descent”. The second descent *is directly* related (and not indirectly as pointed out by the reviewer) to the better generalization in these setting. The second descent is directly linked to better generalization under conditions of noise and domain shifts. Specifically, it demonstrates that highly overparameterized models, despite their capacity to memorize the training set, are less prone to overfitting and can generalize effectively in these challenging scenarios.This insight highlights a practical guideline for users: adopting highly overparameterized models can mitigate the adverse effects of noise and domain shifts, leveraging the second descent phase to achieve robust performance. This connection is central to our findings and supports the argument that overparameterization, rather than being a drawback, is a critical tool for addressing these challenges.
> > > > >
> > > > > **And again, why does it not happen in CNNs?**
> > > > >
> > > > >  It **does** happen in CNNs as we have demonstrated in the paper where we use CNNs to demonstrate double descent (enrtire Section E2).
> > > > >
> > > > > **MNIST is not anymore an accepted image benchmark in Computer Vision, as its easily solved by even very small networks and by memorization, not requiring an image structure.**
> > > > >
> > > > > This comment focuses **exclusively on supervised learning**, which differs fundamentally from the memorization dynamics in autoencoders (AEs). To the best of our knowledge, there are no studies arguing against the use of MNIST for evaluating the reconstruction capabilities of AEs. Furthermore, the reviewer seems to have overlooked that MNIST is widely accepted for explainability studies, as we demonstrated by citing several references, including [A], which uses a CNN trained on MNIST to showcase interpretability.
> > > > > [A] Senetaire, Hugo Henri Joseph, et al. "Explainability as statistical inference." International Conference on Machine Learning. PMLR, 2023.
> > > > >
> > > > > **It seems that they are on supervised learning.**
> > > > >
> > > > > Yes, they are in response to a **direct argument by the reviewer** on “tree-based methods” which are **supervised**.
> > > > > In unsupervised tasks such as clustering or anomaly detection, neural networks also outperform other schemes. Here are examples:
> > > > >
> > > > > [B] Svirsky, J., et al. “Interpretable Deep Clustering for Tabular Data”. In Forty-first International Conference on Machine Learning. 2024
> > > > >
> > > > > [C] Shenkar, T. er al. "Anomaly detection for tabular data with internal contrastive learning." International conference on learning representations. 2022.
> > > > >
> > > > > [D] Rauf, H. et al. "TableDC: Deep Clustering for Tabular Data." arXiv preprint arXiv:2405.17723 (2024).

---

> > > > > > ### Comment · Reviewer_BmzS · 2024-11-22
> > > > > > **Answer to response**
> > > > > >
> > > > > > **Classical Double Descent**
> > > > > >
> > > > > > There is the "classical double descent", which is essentially all key literature also cited in your paper, comparing the generalization error along model size (parameter count). What was said is what the authors *carry over to practice* is not directly related to double descent. Over-parametrization was the road to success of modern LLMs, we do not need the double descent plot to start using over-parametrized models. People used over-parametrized before the authors' paper. So, what of the double descent, specifically, are the authors suggesting to be helpful for downstream applications?
> > > > > >
> > > > > > **CNNs**
> > > > > >
> > > > > > The only investigation on CNNs regarding model size versus generalization error is Figure 35 and 36, showing a double descent in the completely under-specified case (#of channel < # of output classes), and only for MNIST. Please point to more Figures on CNN double descent if this is wrong.
> > > > > >
> > > > > > **MNIST**
> > > > > >
> > > > > > Working in Explainability myself, I am sure that it is not a common, exhaustive benchmark for this field. While using more complex derivations of MNIST (for example with disentangled additional features such as color or morphology), this is usually only  a first investigative finding, complemented with results on other benchmarks following that. These includes most the works that were cited in the rebuttal. I strongly suggest to use a dataset that is a common *application* of sparse autoencoders (proper vision or language benchmarks), to assess the relationship between model size and generalization and ensure that findings are not due to low data complexity (as e.g. in MNIST).
> > > > > >
> > > > > > **Supervised vs unsupervised learning**
> > > > > >
> > > > > > The provided references and corresponding architectures are not being used here in the experiments. To make the point for tabular data, the authors should study the behaviour of such (complex) models, e.g., one of the provided references. The experiments are extremely small scale, according to the description of models in Appendix A.
> > > > > >
> > > > > >
> > > > > > At this point, I would strongly encourage the authors to follow the suggestion of the reviewers asking for further experiments.
> > > > > > I will keep my score.

---

> > > ### Author Response · Authors · 2024-11-21
> > > **Answer cont.**
> > >
> > > **Moreover, tabular data is for sure not the main interest for the neural network community, with a whole line of papers showing that classical statistical learning, especially (gradient boosted) tree-based methods, are still on par or even outperform NNs on tabular data.**
> > >
> > > * see e.g. Y. Gorishnyi et al. Revisiting deep learning models for tabular data NeurIPS 2021, L. Grinsztajn et al. Why do tree-based models still outperform deep learning on typical tabular data? NeurIPS 2022 R. Shwartz-Zivm, A. Armon. Tabular data: Deep learning is not all you need. Information Fusion, 81:84–90, 2022
> > >
> > > We appreciate the reviewer’s comment and acknowledge the historical dominance of tree-based methods, particularly gradient-boosted trees, in tabular data tasks. However, we believe this argument is outdated. Recent advancements in neural network architectures designed explicitly for tabular data have demonstrated superior performance compared to tree-based models [F, G,H, I]. Even well-tuned MLPs nowadays are competitive with tree-based models.
> > >
> > > Our results contribute directly to this emerging direction by providing insights into how neural networks for tabular data can be optimized under challenging conditions, such as noise, domain shifts, and anomalies. By uncovering the behavior of unsupervised neural networks under these scenarios, our work lays the groundwork for designing robust architectures that generalize effectively across tasks. This is particularly relevant to the development of Tabular Foundation Models, which aim to generalize across a wide range of datasets and have been recognized as a top research priority [I].
> > >
> > > [E] Hollmann, Noah, et al. "Tabpfn: A transformer that solves small tabular classification problems in a second." arXiv preprint arXiv:2207.01848 (2022).
> > >
> > > [F] Gorishniy, Yury, Akim Kotelnikov, and Artem Babenko. "TabM: Advancing Tabular Deep Learning with Parameter-Efficient Ensembling." arXiv preprint arXiv:2410.24210 (2024).
> > >
> > > [G] Kim, Myung Jun, Léo Grinsztajn, and Gaël Varoquaux. "CARTE: pretraining and transfer for tabular learning." arXiv preprint arXiv:2402.16785 (2024).
> > >
> > > [H] Gorishniy, Yury, et al. "TabR: Tabular Deep Learning Meets Nearest Neighbors." The Twelfth International Conference on Learning Representations. 2024.
> > >
> > > [I] van Breugel, Boris, and Mihaela van der Schaar. "Why tabular foundation models should be a research priority." arXiv preprint arXiv:2405.01147 (2024).
> > >
> > > **Regarding Weakness 2: The original double descent phenomena came as surprise to the community. Here, especially with all the preceding work that can serve as a blueprint, more is needed to appropriately discuss the new findings - after all the double descent was to be expected to happen in other settings, as it has been discussed for different model types.**
> > >
> > > While the double descent phenomenon has been observed in various supervised learning contexts, it is important to highlight that prior research has shown the absence of double descent in unsupervised learning scenarios. The data models and underlying assumptions in unsupervised learning are fundamentally different from those in supervised learning, which means that double descent is not guaranteed to occur in this context.
> > >
> > > To our knowledge, no previous studies have systematically identified and analyzed the double descent phenomenon in unsupervised learning, specifically with autoencoders. By exploring this behavior and connecting it to factors such as noise, domain shifts, and anomalies, our study provides new insights into the training dynamics and generalization capabilities of over-parameterized unsupervised models.
> > >
> > > We also want to emphasize that novelty in academic research does not always need to be surprising. Discovering new phenomena in unexplored contexts or under different assumptions is equally important, as it deepens our understanding of the field and lays the groundwork for future research. Our findings offer just that—an extension of the understanding of double descent into a previously unstudied area, with practical implications for model design and robustness.

---

> > > > ### Comment · Reviewer_BmzS · 2024-11-21
> > > > **Answer pt2**
> > > >
> > > > Thank you for bringing up these interesting tabular data papers. It seems that they are on **supervised** learning.
> > > >
> > > > A more theoretical analysis on the double descent is in my opinion still necessary, also asked by Reviewer v9nh. The reviewers argued that there observation offers "...an extension of the understanding of double descent into a previously unstudied area", and indeed an observation that advances the *understanding* of a phenomena are important. But without a (theoretical) justification or more extensive and critical discussion on the settings (type of noise, domain shift, network architecturs) this understanding is simply not given.

---

> ### Author Response · Authors · 2024-11-22
> **Comment to the response**
>
> **Classical Double Descent:**
>
> **"People used over-parametrized before the authors' paper. So, what of the double descent, specifically, are the authors suggesting to be helpful for downstream applications?"**
>
> The reviewer suggests that since over-parameterized models were already used in practice prior to the authors' work, studying the double descent phenomenon is unnecessary. However, this perspective overlooks the significant impact that understanding double descent has had on the field. The discovery of double descent challenged traditional beliefs about the bias-variance tradeoff, which suggested that increasing model complexity would always lead to overfitting. Double descent revealed that beyond a certain level of over-parameterization, performance can improve again, defying these expectations.
>
> This insight has reshaped the understanding of model behavior, particularly for over-parameterized models like deep neural networks. It has influenced how we think about capacity, training data, and generalization, and has led to new research on optimizing model architectures and regularization strategies. In short, research on double descent has directly informed practical machine learning applications, advancing generalization performance and robustness.
>
> **CNNs:**
>
> **"Figure 35 and 36, showing a double descent in the completely under-specified case (#of channel < # of output classes)":**
>
> This comment further indicates a misunderstanding, as the reviewer appears unfamiliar with unsupervised learning using autoencoders, where the notion of 'output classes' does not apply in the traditional sense.
>
> Additionally, the reviewer precisely wrote, "And again, why does it not happen in CNNs?" which demonstrates either a lack of thorough reading or oversight of our work. We have explicitly addressed this point in the paper, showing that double descent indeed occurs in CNNs. Additionally, on top of Figures 35 and 36, Figure 34 also presents model-wise double descent for different levels of sample and feature noise, and Figure 37 for the presence of domain shift.
>
> Furthermore, we have conducted additional experiments based on the reviewer's earlier requests, yet the reviewer now moves the goalpost, implying that more experiments are still needed. We believe the current set of experiments sufficiently demonstrates the occurrence of double descent in CNNs across a range of settings, and we have made every effort to address previous concerns adequately.
>
> **MNIST:**
>
> To support the applicability of our work, we have not only included MNIST but also utilized complex datasets, including single-cell RNA sequencing data, which presents significant challenges such as high-dimensionality, sparsity, and noise. These datasets are representative of real-world tabular applications and align well with the scope of our study. Most of the works cited in our rebuttal also focus on such datasets, highlighting their relevance to sparse autoencoders and the broader research context.
>
> We respectfully disagree with the reviewer’s assertion that findings on MNIST are inherently limited by its data complexity. We have cited several recent publications from top-tier venues [A-D] to substantiate the fact that MNIST is commonly used for demonstrating interpretability and for evaluating autoencoders in literature. Additionally, our results on single-cell RNA data further validate the robustness of our approach and the relationship between model size and generalization in challenging, high-dimensional tabular domains.
>
> The suggestion to use vision or language benchmarks, while valuable in certain contexts, appears to shift the goalpost unnecessarily. Our study focuses on tabular data and interpretable sparse autoencoders, and the datasets chosen—MNIST and single-cell RNA data—are appropriate for addressing the goals of our work. We believe these choices provide a strong foundation for demonstrating the effectiveness of our methodology.
>
> **“ensure that findings are not due to low data complexity”**
>
> Our paper includes single cell RNA-seq data which is much more complex than all vision datasets.
> We also include results for anomaly detection on CelebA (attributes).
>
> [A] Balın, M.F.,  et al., 2019, May. Concrete autoencoders: Differentiable feature selection and reconstruction. In International conference on machine learning (pp. 444-453). PMLR.
>
> [B] Kim, C., et al., Discovering Features with Synergistic Interactions in Multiple Views. In Forty-first International Conference on Machine Learning. 2024
>
> [C] Jethani, N., et al. "Have We Learned to Explain?: How Interpretability Methods Can Learn to Encode Predictions in their Interpretations." International Conference on Artificial Intelligence and Statistics. PMLR, 2021.
>
> [D] Svirsky, J., et al.. Interpretable Deep Clustering for Tabular Data. In Forty-first International Conference on Machine Learning. 2024

---

> > ### Comment · Reviewer_BmzS · 2024-11-22
> > **Response**
> >
> > "The reviewer suggests that since over-parameterized models were already used in practice prior to the authors' work, studying the double descent phenomenon is unnecessary."
> > This statement is wrong and does not relate to what I wrote, as clearly evident above. I highly suggest the authors read my original comment word by word. to really understand its semantics.
> >
> > "This comment further indicates a misunderstanding, as the reviewer appears unfamiliar with unsupervised learning using autoencoders, where the notion of 'output classes' does not apply in the traditional sense."
> > This is a highly insulting comment, again misreading the words - as before. The number of conceptual classes in this dataset is 10 (the digits). Indeed when you visualize the data using e.g. tSNE (*unsupervised* on image space only, *without* labels) you do receive 10 clusters corresponding to digits. Having anything less than 10 neurons (or channels) to represent the data naturally leads to a loss of information. Clearly no classical double descent, just bad representation.
> >
> >
> > The authors clearly stopped arguing constructively or keeping up a proper discussion, and changed to audacious, often insulting, comments by writing wrong statements and labeling them as mine, with the ovious purpose of discrediting my review (also reflected in third person use). At this point I will stop this discussion thread and will for sure stick to a reject rating, also in line with the experimental and theoretical limitations other reviewers highlighted.

---

> > > ### Author Response · Authors · 2024-11-22
> > > **Proffesional reasoning for point reduction**
> > >
> > > For clarity, we would like to note that the reviewer has reduced their score from 5 to 3 at this stage of the discussion, seemingly not based on scientific reasoning but rather as an emotional response. While we sincerely apologize for any unintended offense caused during this exchange and acknowledge the possibility that we may have misinterpreted some of your arguments, we believe it is important to ensure that the evaluation reflects the scientific merits of our work. Regarding your comment on information loss in MNIST, we would like to clarify that this is an inherent characteristic of undercomplete autoencoders and is well-understood in the proper context of their application. Furthermore, in our MNIST example, we have also evaluated models with a number of channels exceeding the number of classes, but it is important to note that this does not necessarily align with the typical regimes used in this domain. For instance, anomaly detection using autoencoders are typically implemented with much less channels than the number of classes.
> > >
> > > Furthermore, we have addressed all the concerns raised by the reviewer throughout this discussion. These include demonstrating the importance of deep learning methods for tabular data, validating the appropriateness of MNIST as a benchmark for unsupervised autoencoders, providing evidence of the occurrence of double descent in sparse autoencoders, and clarifying the practical implications of our results. Each of these points was supported by clear arguments and relevant references, reinforcing the robustness and relevance of our approach.
> > >
> > > Additionally, we have pointed out several factual inaccuracies in the review that suggest critical aspects of our paper may have been overlooked. In light of these clarifications, we respectfully request that the reviewer revisit our arguments in this discussion and reconsider the decision to lower the score.

---

> ### Author Response · Authors · 2024-11-22
> **Comment to the response cont.**
>
> **Supervised vs unsupervised learning**:
>
> **"To make the point for tabular data, the authors should study the behaviour of such (complex) models, e.g., one of the provided references. "**
>
> The reviewer’s claim is incorrect. Our architectures are, in fact, larger and more complex than the models employed in [E-G], which achieve state-of-the-art (SOTA) performance. While those specific architectures are not explicitly used in our work, the cited references were included to demonstrate that such architectures have been effective in related contexts, contradicting the reviewer's argument. Furthermore, our approach employs unsupervised autoencoders (AEs), which are well-suited for tabular data and align with the goals of our study.
>
> **“tabular data is for sure not the main interest for the neural network community, with a whole line of papers showing that classical statistical learning, especially (gradient boosted) tree-based methods, are still on par or even outperform NNs on tabular data”**
>
> [E] Gorishniy, Yury, Akim Kotelnikov, and Artem Babenko. "TabM: Advancing Tabular Deep Learning with Parameter-Efficient Ensembling." arXiv preprint arXiv:2410.24210 (2024).
>
> [F] Gorishniy, Yury, et al. "TabR: Tabular Deep Learning Meets Nearest Neighbors." The Twelfth International Conference on Learning Representations. 2024.
>
> [G] Holzmüller D, Grinsztajn L, Steinwart I. Better by default: Strong pre-tuned mlps and boosted trees on tabular data. arXiv preprint arXiv:2407.04491. 2024 Jul 5.

---

### Official Review · Reviewer_TSMF · 2024-11-03

**Soundness:** 2
**Presentation:** 2
**Contribution:** 2
**Rating:** 3
**Confidence:** 3

**Summary:**

This paper experimentally discovers the double descents (multiple descents) phenomenon in autoencoder models and explains this phenomenon to sample noise, feature noise, domain shift, and outliers. The double descent phenomenon represents a specific pattern in the generalization performance of a model based on its size. Notably, this phenomenon has provided significant motivation for studying the generalization performance of overparameterized models.

**Strengths:**

This study also provides empirical examples related to the generalization performance of overparameterized models in unsupervised learning. These findings will encourage further research on model complexity in unsupervised learning.

**Weaknesses:**

Although the paper claims to present extensive simulation results demonstrating the double descent phenomenon in unsupervised learning, the experiments primarily seem to involve regression tasks. This interpretation arises because the study examines the performance changes in autoencoders by adding noise to data or features, which can essentially be understood as fitting an arbitrary target variable, similar to a regression model. In this sense, the results could be viewed within the same context as previous findings observed in supervised learning.

Moreover, discovering multiple descent patterns may not offer a new perspective based solely on the current simulation results. Additional discussion and analysis would likely be necessary to substantiate this claim.

Before discussing the double descent phenomenon in unsupervised learning, it is essential first to define the performance metric in the context of unsupervised learning. If unsupervised learning is limited to distribution learning, I suggest defining the performance metric based on the distance to the target distribution and examining how performance varies with model size. Suitable models for unsupervised learning would include those capable of distribution learning, such as VAE, GAN, or diffusion models. Alternatively, a simpler experimental approach could involve using a clustering task.

**Questions:**

Is it possible to define the double descent phenomenon for unsupervised learning models other than autoencoders?

How can the use of test loss to define predictive performance in unsupervised learning be justified? From the perspective of distribution learning, shouldn't distributional similarity serve as the performance metric?

---

> ### Author Response · Authors · 2024-11-20
> **Response to the reviewer**
>
> We thank the reviewer for acknowledging the main strengths of our paper, particularly the encouragement for further research on model complexity in unsupervised learning.
>
> **P1:** To clarify the distinction between our study of double descent in unsupervised learning and traditional regression tasks:
>
> Our experiments focus on **autoencoders (AEs)**, which fundamentally differ from regression models in both objective and structure. Unlike regression, where models map inputs to a specific target variable, AEs are designed to learn a compressed representation of the input data itself without supervision, with the goal of capturing inherent structure rather than fitting an external target. The "target" in an AE is simply the input itself, and the model's reconstruction loss reflects how well it retains underlying patterns rather than directly mapping to a response variable as in regression. By examining AEs under varying noise conditions, domain shifts, and anomalies, our study provides insights specific to the unsupervised setting, where multiple descent arises due to challenges in learning these representations rather than fitting labels. Moreover, regression is a completely supervised task, which assumes access to the target data, while our setup is completely unsupervised.
>
> The fact that AE training and regression are completely different tasks is exemplified by the extensive theoretical and practical research that studies these two tasks separately. There is no use case of regression in which the target variable is $X$ ($X$ is the data). This difference involves not just a specific choice of target variable but also emphasizes that all types of contamination should be applied to both the input and output. This contrasts significantly with label/ target noise in a supervised setting, which is only applied to the output variable (y). Here is a list of studies discussing double descent in regression tasks, mostly linear regression: [1, 2, 3, 4, 5].
>
> We would like to highlight that research into similar or related phenomena across different data models and architectures is a common practice in machine learning. For example, the behavior of L1-norm regularization has been studied separately in regression [6] and in autoencoders (AEs) [7]. These independent investigations underline the fundamental distinctions between these tasks; if they were inherently identical, distinct analyses of L1-norm behavior would not be necessary.
>
> Our findings extend the understanding of double descent to unsupervised tasks and highlight unique implications for model selection in scenarios like anomaly detection and domain adaptation, which are not covered by standard regression-based double descent observations.
>
> [1] Harmless interpolation of noisy data in regression.
>
> [2] Benign overfitting in linear regression.
>
> [3] On the number of variables to use in principal component regression.
>
> [4] Dimensionality reduction, regularization, and generalization in overparameterized regressions.
>
> [5] More data can hurt for linear regression: Sample-wise double descent, 2019.
>
> [6] Regression shrinkage and selection via the lasso.
>
> [7] Sparse L^1- Autoencoders for Scientific Data Compression
>
>
> **P2 - discovering multiple descent patterns may not offer a new perspective based solely on the current simulation results:**
>
> In contrast to the reviewer's claim, our work provides, for the first time, evidence of double descent in unsupervised learning using real-world data, moving beyond purely synthetic simulations. Specifically, we demonstrated the occurrence of double descent not only in synthetic settings but also on established benchmark datasets like MNIST, as well as in several real-world single-cell RNA-seq datasets. This extension to real-world datasets significantly enhances the relevance of our findings, as it validates the existence of multiple descent phenomena under realistic conditions that practitioners often encounter.
>
> The inclusion of datasets such as MNIST and single-cell RNA-seq highlights the generalizability of our results across different types of data, ranging from image data to highly complex biological data with inherent noise and domain shifts. This real-world evidence contrasts with prior work that primarily relied on simulations and highly controlled environments, which may not fully capture the complexities of practical applications. Our work addresses this gap by systematically exploring these double descent behaviors in settings where noise, anomalies, and domain shifts are naturally present, providing valuable insights that are directly applicable to real-world scenarios.
>
> Additionally, our analysis illustrates the implications of double descent for practical tasks like anomaly detection and domain adaptation, particularly in unsupervised settings.

---

> ### Author Response · Authors · 2024-11-20
> **Response cont.**
>
> **P2 - It is essential first to define the performance metric in the context of unsupervised learning:**
>
> We appreciate the reviewer's emphasis on defining performance metrics when discussing the double descent phenomenon in unsupervised learning. In our work, we explicitly define and employ key metrics suitable for the unsupervised context. Specifically, we use the test loss, measured by the Mean Squared Error (MSE) between the input and the reconstructed output, as a central metric. This is particularly appropriate for autoencoders, as it directly reflects the model’s ability to capture the underlying data structure rather than merely overfitting noise—crucial for understanding double descent behavior.
>
> Beyond MSE, we also include other relevant metrics to demonstrate the practical implications of double descent. For anomaly detection, we use the ROC-AUC score to evaluate the model's ability to differentiate between normal and anomalous samples. In our domain adaptation experiments with single-cell RNA-seq data, we use the K-nearest Neighbors Domain Adaptation Test (KNN-DAT) to assess how well latent representations align across different domains. By leveraging these well-defined performance metrics, we ensure a comprehensive characterization of double descent patterns that is relevant to real-world unsupervised learning tasks.
>
> **P3 - If unsupervised learning is limited to distribution learning, I suggest defining the performance metric based on the distance to the target distribution:**
>
> Unsupervised learning is by no means limited to distribution learning. Our paper is specifically focused on under-complete autoencoders (AEs), which are non-generative models. AEs serve multiple purposes in machine learning, including dimensionality reduction, denoising, compression, anomaly detection, feature selection, and domain adaptation. Unlike Variational Autoencoders (VAEs) or Generative Adversarial Networks (GANs), which are often used for density estimation, our focus is on leveraging AEs for practical applications that do not require direct distribution modeling.
>
> We have chosen performance metrics that align with the objectives of these tasks, such as Mean Squared Error (MSE) for reconstruction quality, ROC-AUC for anomaly detection, and KNN-DAT for assessing domain adaptation. Details about these metrics and their relevance to our analysis are provided in Subsections 4.1 and 5.1 of the paper and addressed in our response to point 2 above.
>
> **Q1 - Is it possible to define the double descent phenomenon for unsupervised learning models other than autoencoders?**
>
> A study has investigated the phenomenon of double descent in Generative Adversarial Networks (GANs). The researchers demonstrated double descent in both supervised and semi-supervised GANs; however, they did not observe a double descent curve for unsupervised GANs [1]. This study is also referenced in our paper.
>
> [1] Luzi, Lorenzo, Yehuda Dar, and Richard Baraniuk. "Double descent and other interpolation phenomena in gans." arXiv preprint arXiv:2106.04003 (2021).
>
> **Q2 - How can the use of test loss to define predictive performance in unsupervised learning be justified?**
>
> Please read P3. Unsupervised learning is not limited to distribution learning and we are working with under-complete AEs, which is not a part of distributional learning.

---

> > ### Comment · Reviewer_TSMF · 2024-11-24
> >
> > Thank you for your detailed response.
> >
> > First, I would like to share my thoughts on why the double descent property is an important concept and pose some questions. I find it particularly fascinating that the double descent phenomenon in supervised learning demonstrates that the traditional model selection criterion, the bias-variance trade-off, is not absolute. Theoretically, this phenomenon has led to subsequent research, enhancing our understanding of data interpolation and showing that unconventional methods, such as negative regularization, can perform well. The double descent property has provided an observable explanation for the success of neural network models in various real-world tasks. It has also become a new belief that replaces the traditional bias-variance trade-off in neural networks.
> >
> > **(Q1)** My question is: what existing concept or assumption does the double descent property in unsupervised learning replace, as argued in this paper? For example, in kernel bandwidth selection, the bias-variance trade-off is often invoked as an explanation. Can the choice of kernel bandwidth be explained using the double descent property in unsupervised learning?
> >
> > **(Q2)** It is widely agreed that autoencoders are a form of unsupervised learning. However, autoencoders can also be viewed as supervised learning models if we assume that latent variables are observable. Although latent variables are not observed in practice, theoretically, this is akin to assuming a perfect encoder is known. Under this assumption, training the decoder of an autoencoder can be seen as a regression problem, where the existing double descent property explains the complexity of the regression model. Moreover, if the inverse map of the trained decoder is the estimation target, this, too, could be explained by the double descent property in supervised learning. I would like to hear your opinion on this perspective.
> >
> > **(Q3)** While the paper claims to address the double (or multiple) descent property in unsupervised learning, it seems to focus specifically on the double (or multiple) descent property of autoencoders. Using MSE as the evaluation metric appears insufficient to establish the double (or multiple) descent property in unsupervised learning.

---

> > > ### Author Response · Authors · 2024-11-24
> > > **Response to the reviewer**
> > >
> > > We thank the reviewer for engaging in the rebuttal and for sharing their perspective on the importance of research on double descent. Below, we address your questions.
> > >
> > > **Q1 - My question is: what existing concept or assumption does the double descent property in unsupervised learning replace, as argued in this paper? For example, in kernel bandwidth selection, the bias-variance trade-off is often invoked as an explanation. Can the choice of kernel bandwidth be explained using the double descent property in unsupervised learning?**
> > >
> > > The classic bias-variance trade-off has been a fundamental concept in understanding and designing regularization techniques in supervised learning. These ideas have also influenced unsupervised learning, where methods like autoencoders and kernel PCA rely on regularization [1, 2] or reduced complexity (e.g., limiting the number of neurons or constraining kernel bandwidth) to balance reconstruction accuracy and generalization. However, our paper challenges this conventional wisdom by demonstrating that the double descent phenomenon—previously observed in supervised and semi-supervised settings—also occurs in unsupervised autoencoders, contrary to prior assumptions. This finding contrasts with earlier studies [3, 4], which did not identify double descent in unsupervised models and PCA.
> > >
> > > Analogous to the role of kernel bandwidth in supervised learning, our results suggest that increasing the complexity of kernels in kernel PCA can enhance generalization performance. Specifically, in tasks such as anomaly detection and domain adaptation, which are typically sensitive to overfitting, we show that over-parameterized models can achieve improved generalization. This second phase of performance improvement, associated with the double descent regime, supports the potential of over-parameterized models to effectively learn high-dimensional data representations without overfitting.
> > >
> > > **Q2 - It is widely agreed that autoencoders are a form of unsupervised learning. However, autoencoders can also be viewed as supervised learning models if we assume that latent variables are observable. Although latent variables are not observed in practice, theoretically, this is akin to assuming a perfect encoder is known. Under this assumption, training the decoder of an autoencoder can be seen as a regression problem, where the existing double descent property explains the complexity of the regression model. Moreover, if the inverse map of the trained decoder is the estimation target, this, too, could be explained by the double descent property in supervised learning. I would like to hear your opinion on this perspective.**
> > >
> > > We appreciate the reviewer's exciting perspective on the connection between autoencoders and supervised learning. Your view of the autoencoder as a regression problem with a perfect encoder provides valuable insights. However, we would like to clarify two key points about our work:
> > >
> > > 1. **No Assumption of a Perfect Encoder:** Our study does not assume a perfect encoder or observable latent variables. Instead, we simultaneously train both the encoder and decoder, which allows us to account for reconstruction errors and noise—key factors in our investigation of the double descent property. While assuming a perfect encoder is theoretically valid, it significantly alters the autoencoder's behavior.
> > >
> > > 2. **Noise in Input and Target:** Even assuming a perfect encoder, the data model would not allow for noise addition during training since the latent representation would be noiseless. In our approach, we add noise to both X (the input) and the reconstruction target (also X). This differs from traditional supervised regression, where double descent has been examined. To the best of our knowledge, double descent is not observed in noiseless regression. Our setup, with noise in both input and target, aligns more with unsupervised learning characteristics and differs from the standard supervised regression framework.
> > >
> > > 3. **Relation to PCA Regression:** The model the reviewer is referring to is most similar to PCA regression, where principal components serve as inputs for regression tasks. In such models, it has been theoretically shown that double descent does not occur in the linear case [5]; over-parameterization usually leads to overfitting without the advantages observed in double descent scenarios.
> > >
> > > In contrast, our research demonstrates that when transitioning from linear models to nonlinear neural
> > > networks in autoencoders, the double descent phenomenon does appear. This finding suggests that exploring nonlinear models, such as kernel PCA regression [6], could be an intriguing direction for future research on double descent. However, such an exploration would necessitate a supervised data model, which, while potentially similar to ours, would not be identical due to differing assumptions regarding the relationship between inputs and targets.

---

> > > ### Author Response · Authors · 2024-11-24
> > > **Response cont.**
> > >
> > > **Q3 - While the paper claims to address the double (or multiple) descent property in unsupervised learning, it seems to focus specifically on the double (or multiple) descent property of autoencoders. Using MSE as the evaluation metric appears insufficient to establish the double (or multiple) descent property in unsupervised learning.**
> > >
> > > We appreciate the reviewer’s comment. As indicated in our title and throughout the paper, our research focuses explicitly on demonstrating the existence of the double (or multiple) descent phenomenon in unsupervised autoencoders. Autoencoders are fundamental to unsupervised learning and are utilized for various tasks, including anomaly detection, clustering, denoising, dimensionality reduction, interpretability, and feature selection, among others. While our findings offer valuable insights into this class of models, we do not claim to address the double descent phenomenon across the entire field of unsupervised learning.
> > >
> > >
> > > We recognize that unsupervised learning covers a wider range of models, including generative approaches such as GANs and diffusion models, which were not the focus of our study. Exploring the presence of double descent in these models presents an exciting opportunity for future research, as mentioned in our discussion section. We hope that our contribution lays the groundwork for investigating this property in other areas of unsupervised learning.
> > >
> > >
> > > [1] Zhao, Junbo, et al. "Adversarially regularized autoencoders." International conference on machine learning. PMLR, 2018.
> > >
> > >
> > > [2] Vo, Hai X., and Louis J. Durlofsky. "Regularized kernel PCA for the efficient parameterization of complex geological models." Journal of Computational Physics 322 (2016): 859-881.
> > >
> > >
> > > [3] Lupidi, Alisia, Yonatan Gideoni, and Dulhan Jayalath. "Does Double Descent Occur in Self-Supervised Learning?." arXiv preprint arXiv:2307.07872 (2023).
> > >
> > >
> > > [4] Gedon, Daniel, Antonio H. Ribeiro, and Thomas B. Schön. "No double descent in PCA: Training and pre-training in high dimensions, 2023." URL https://openreview. net/forum.
> > >
> > >
> > > [5] Gedon, Daniel, Antonio H. Ribeiro, and Thomas B. Schön. "No Double Descent in Principal Component Regression: A High-Dimensional Analysis." Forty-first International Conference on Machine Learning. 2024.
> > >
> > >
> > > [6] Rosipal, Roman, et al. "Kernel PCA for feature extraction and de-noising in nonlinear regression." Neural Computing & Applications 10 (2001): 231-243.

---

> > > > ### Comment · Reviewer_TSMF · 2024-11-28
> > > >
> > > > Thank you for your response. I will maintain my score.
> > > >
> > > > - Observing and simulating the multiple descent property phenomenon from various perspectives could provide new insights for future research.
> > > > - However, the study lacks generality as the model used to explain the multiple descent property phenomenon is limited to AE.

---

### Official Review · Reviewer_v9nh · 2024-11-03

**Soundness:** 3
**Presentation:** 3
**Contribution:** 2
**Rating:** 5
**Confidence:** 4

**Summary:**

This paper investigates the phenomenon of multiple descent in unsupervised learning paradigms, specifically using under-complete auto-encoders (AEs). Contrary to prior studies that suggest double descent does not occur in unsupervised learning, this work empirically demonstrates the presence of multiple descent under conditions of low SNR, significant anomalies, and domain shifts. Additionally, it identifies scenarios where multiple descent affect test loss, providing new insights to model selection and training with over-parameterized networks.

**Strengths:**

* This paper addresses an important and timely topic in understanding the generalization behavior of over-parameterized models.
The paper is well-written and easy to follow. It is interesting to see that under low SNR, over-parameterized models’ ability to fit (or memorize) noise can actually improve generalization when encountering domain shifts.
* The experiments are thorough, and the analysis of results provides insightful observations.

**Weaknesses:**

-  While the paper highlights the roles of noise, domain shift, and anomalies in multiple descent, providing further theoretical insights into the underlying mechanisms could enrich the understanding of why this phenomenon occurs.
- Defining critical regions for multiple descent more explicitly, possibly through some kind of scaling law like function characterizations, could make the findings more actionable and help readers choose appropriate model complexity and stopping condition in practical settings.

**Questions:**

- In practice, noise is not always Gaussian. How would different noise distributions impact the behavior of multiple descent?
- Could the paper provide theoretical insights or hypotheses on why multiple descent occurs under conditions of low SNR, domain shift, and anomalies, and how critical regions might be characterized more systematically?
- Are there practical ways to predict the occurrence of multiple descent based on data characteristics or noise conditions, making it more applicable for real-world scenarios?
- Given that SNR is challenging to compute from data without specific model assumptions about signal and noise, what are some methods or approximations to determine SNR in real-world data?
- For the KNN-DAT experiment (Fig 12), what would be the relationship of test loss curve with different levels of source-target distribution shift? (e.g. measured by distribution distance such as KL divergence or Wasserstein distance) Does the relationship confirm with the simulated experiment?

---

> ### Author Response · Authors · 2024-11-20
> **Response to the reviewer**
>
> We thank the reviewer for acknowledging the main strengths of our paper, particularly the importance of understanding the generalization behavior of over-parameterized models, how well the paper is written, and the thorough experiments providing insightful observations.
>
> **W1 - Providing further theoretical insights:**
>
> We agree that additional theoretical insights would deepen the understanding of this phenomenon. Our primary goal, however, was to provide comprehensive empirical evidence demonstrating the existence of the double descent phenomenon in unsupervised learning and to address the inconsistencies in the literature. We believe that our extensive empirical experiments, covering various contamination setups such as sample noise, feature noise, domain shift, and anomalies, help offset the absence of theoretical analysis.
>
> Furthermore, significant contributions like [1] and [2] have advanced the understanding of double descent in supervised learning despite lacking theoretical backing. We hope that our extensive empirical findings across different contamination setups and the confirmation of double descent in unsupervised tasks will inspire researchers to investigate the theoretical aspects of this phenomenon in greater depth, both qualitatively and quantitatively.
>
> [1] Mikhail Belkin, Daniel Hsu, Siyuan Ma, and Soumik Mandal. Reconciling modern machine learning practice and the classical bias–variance trade-off. Proceedings of the National Academy of Sciences, 116(32):15849–15854, 2019.
>
> [2] Preetum Nakkiran, Gal Kaplun, Yamini Bansal, Tristan Yang, Boaz Barak, and Ilya Sutskever. Deep double descent: Where bigger models and more data hurt. Journal of Statistical Mechanics: Theory and Experiment, 2021(12):124003, 2021.
>
> **W2 - Defining critical regions for multiple descents more explicitly:**
>
> As mentioned in the first section addressing the weaknesses, our research is entirely empirical. While some papers explore the mathematical relationships governing the double descent curve in supervised learning—often identifying the critical regime based on the number of training samples and noise energy—these studies typically focus on linear, sparse models with few layers, substantiating their findings through synthetic data whose characteristics they can carefully control [1, 2, 3, 4]. In contrast, our study employs non-linear models in an unsupervised context using real-world datasets.
>
> Our results indicate that the critical regimes deviate from the scaling laws reported in those theoretical papers. Moreover, we observe that data characteristics significantly influence the double descent behavior. For instance, while single-cell RNA data uniquely displays a triple descent pattern, other datasets consistently exhibit double descent.
>
> Thank you for suggesting a definition of scaling law, which we are currently working on.
>
> [1] Jacot, Arthur, et al. "Implicit regularization of random feature models." International Conference on Machine Learning. PMLR, 2020.
>
> [2] Chen, Lin, et al. "Multiple descent: Design your own generalization curve." Advances in Neural Information Processing Systems 34 (2021): 8898-8912.
>
> [3] Liang, Tengyuan, Alexander Rakhlin, and Xiyu Zhai. "On the multiple descent of minimum-norm interpolants and restricted lower isometry of kernels." Conference on Learning Theory. PMLR, 2020.
>
> [4] Belkin, Mikhail, Daniel J. Hsu, and Partha Mitra. "Overfitting or perfect fitting? risk bounds for classification and regression rules that interpolate." Advances in neural information processing systems 31 (2018).
>
> **Q1 - Noise is not always Gaussian:**
>
> For your request, we have conducted an experiment involving the single-cell RNA and linear subspace datasets for the case of sample noise with Laplace distributed noise. The experiment was held exactly as the one in Figure 3 in order to compare the results with different types of noise distribution. We have added a discussion in lines 249-250 in the main paper, and in the new Appendix E.5 (lines 1610-1613) and Figure 55.

---

> > ### Comment · Reviewer_v9nh · 2024-12-02
> > **Thank you for the detailed response.**
> >
> > I appreciate your effort to add the KL divergence analysis and the extra experiment results in the appendix.  The references that you provided are also helpful.
> >
> > I acknowledge the value of providing empirical evidence for the double descent phenomenon in unsupervised learning. However, the analysis in this work primarily focuses on whether certain factors influence the qualitative features of multiple descent. I still believe that a more in-depth investigation into the underlying mechanisms is needed, with a clearer connection to model selection or training strategies. For example, instead of discussing relative values like "low" or "high," it would be more informative to describe specific correlations, such as linear or quadratic relationships. I maintain that a careful examination of theoretical results in related fields could lead to more systematic and unified findings. For these reasons, I maintain my original scoring. (In its current form, this work could be more suitable for a workshop paper.)

---

> ### Author Response · Authors · 2024-11-20
> **Response cont.**
>
> **Q2 - Hypothesis on why multiple descents occur:**
>
> We observed that for the double descent curve to manifest in the presence of sample noise, the noise must be predominant—characterized by a high proportion of noisy samples and low SNR values. In such cases, models in the critical regime, which have sufficient capacity, prioritize memorizing the noise over learning the underlying signal, leading to increased test loss. In contrast, when the SNR is high, even if all samples are noisy, the model behaves similarly to one trained on clean data. With high SNR or a small fraction of noisy samples, model-wise double descent does not occur, as minimal noise allows models with sufficient capacity to learn the underlying signal rather than memorize noise, thus avoiding the critical regime. Over-parameterized models exhibit a second descent because they have enough capacity to memorize noisy samples while still capturing the characteristics of the signal. A similar pattern holds for anomalies with low SAR, as illustrated in Figure 6.a, since noisy samples can be interpreted as a form of anomaly.
>
> In the context of domain shift, models operating in the critical regime may memorize the source domain in order to reduce training loss. This can lead to an increased test loss when applied to target data. Over-parameterized models, on the other hand, can overfit the source data even as they minimize the model’s weight norm. These models, characterized by a lower L2 norm, tend to generalize better across different domains, which helps to lower the test loss. For a visual representation of this concept, refer to [1] Figure 3 and [2] Figures 2.d and 2.g. An explanation of the critical regions can be found in W2.
>
> [1] Belkin, Mikhail, et al. "Reconciling modern machine-learning practice and the classical bias–variance trade-off." Proceedings of the National Academy of Sciences 116.32 (2019): 15849-15854.
>
> [2] Dar, Yehuda, Vidya Muthukumar, and Richard Baraniuk. "A farewell to the bias-variance tradeoff." An Overview of the Theory of Overparameterized Machine Learning. ArXiv abs/2109.02355 (2021).
>
> **Q3 - Practical ways to predict the occurrence of multiple descents based on data characteristics or noise conditions:**
>
> We appreciate the reviewer’s question regarding practical ways to predict multiple descent phenomena. We first note that it is possible to estimate key data characteristics, such as intrinsic dimensionality and structural properties (explained in the next answer), which can provide initial indications of whether multiple descent is likely to occur. For instance, when noise levels are low and the dimensionality of the data is high, the phenomenon is less likely to arise. Conversely, high noise levels and low dimensionality will increase the likelihood of observing a double descent. However, validating these hypotheses requires further theoretical analysis.
>
> Our study highlights that the critical factors influencing multiple descent include the number of training samples, signal-to-noise ratio (SNR), noise levels, the presence of anomalies, and domain shifts. Identifying and quantifying these dependencies can help refine predictions about when double descent behavior might emerge. Additionally, practical approximations, such as using proxy metrics for SNR estimation or monitoring training dynamics in real-time, could provide valuable insights for practitioners dealing with noisy or shifted datasets. We believe that these are great questions for future work.

---

> ### Author Response · Authors · 2024-11-20
> **Response cont. 2**
>
> **Q4 - Determine SNR in real-world data:**
>
> One approach for estimating SNR from noisy data without making specific assumptions is using the Marchenku Pastur (MP) law [1]. First, we calculate the covariance matrix of our data. second, we perform eigenvalue decomposition to obtain the eigenvalues of the covariance matrix. Thirdly, we identify the eigenvalues that fall within the MP distribution (eigenvalues probably associated with noise) and the ones that do not (considered to be associated with the signal). Finally, we use the ratio of the signal-related eigenvalues to the noise-related eigenvalues to estimate the SNR. This approach assumes that the noise is i.i.d.
>
> Another approach is using manifold learning, where the assumption is that the signal lies on a lower-dimensional manifold. This method can estimate the intrinsic dimension of the data (d) using existing schemes such as [2, 3]. After that, we can estimate the power of the signal by calculating the variance retained by the lower-dimensional manifold (for example, in PCA, we will calculate the variance based on the first k eigenvalues), and the rest of the eigenvalues will be proportional to the variance of the noise. Finally, the ratio between them yields the SNR.
>
> Other methods can be applied for time signals, such as power spectrum analysis. This involves applying a Fourier transform to calculate the power spectral density (PSD) of the signal. By separating the power associated with the signal from that of the noise, we can compute their ratio to estimate the SNR. However, this method provides an approximation since some noise components may overlap with the signal's frequencies. Using techniques like low-pass filtering to separate the signal from the noise can also be effective. It’s essential to understand the data’s characteristics, domain knowledge, and relevant model assumptions for better accuracy.
>
> [1] Gavish, Matan, and David L. Donoho. "The optimal hard threshold for singular values is $4/\sqrt {3} $." IEEE Transactions on Information Theory 60.8 (2014): 5040-5053.
>
> [2] Ceruti, Claudio, et al. "DANCo: dimensionality from angle and norm concentration." arXiv preprint arXiv:1206.3881 (2012).
>
> [3] Facco, Elena, et al. "Estimating the intrinsic dimension of datasets by a minimal neighborhood information." Scientific reports 7.1 (2017): 12140.
>
> **Q5 - KNN-DAT experiment (Figure 12):**
>
> Thank you for raising this important question. As you suggested, we utilized the KL-divergence metric to assess the distribution shift between our source data ('Baron') and the target datasets ('Segerstolpe', 'Xin', 'Mutaro', and 'Wang'). We have added a discussion about this in lines 449-452 of the revised manuscript and modified the legend of Figure 12 to include the KL-divergence results. Our findings indicate a relationship between the height of the test loss and the domain shift of the data, which aligns with the results from the simulated experiment.

---

> ### Author Response · Authors · 2024-11-24
> **Notification**
>
> Dear reviewer, we appreciate the time and effort you have dedicated to reviewing our paper. As the discussion period is limited, we kindly request that the reviewer evaluate the new information provided in the rebuttal. We are eager to improve our paper and resolve all the concerns raised by the reviewer. If there are any remaining concerns that have not been addressed, we would be happy to provide further explanations.

---

### Official Review · Reviewer_NdzY · 2024-11-04

**Soundness:** 2
**Presentation:** 2
**Contribution:** 2
**Rating:** 5
**Confidence:** 4

**Summary:**

This paper investigates the double descent phenomenon in relation to three key training parameters, training epochs, model size, and data sample size, within an unsupervised learning framework. To illustrate the phenomenon, consider training epochs: initially, test error decreases as the number of training epochs increases. However, after reaching a certain threshold, test error begins to increase. After a period of error rising, the test error decreases again, known as, double descent. Additionally, the authors conduct experiments using synthetic data and small-scope models to empirically examine the impact of customized noise, domain shifts, and anomalies on double descent behavior. Finally, the study assesses how model size influences domain adaptation and anomaly detection performance.

**Strengths:**

1. This work provides empirical evidence that the double descent phenomenon exists in various training parameters under unsupervised learning settings, and the non-linear relationship observed between test error and training parameters may be of interest.
2. Unlike previous studies, to verify the influential factor model size, this work proposes utilizing specific layers in different dimensions to simulate the changing of model size. This modification is well-suited for experiments with small-scope neural networks by avoiding introducing extra variables.

**Weaknesses:**

$\textbf{Weakness in method motivation and novelty}$

1. This work needs more motivation to study the phenomenon of double descent. Particularly, this work does not provide possible theoretical improvement or help better understand the unsupervised learning problems. Additionally, the patterns identified lack a compelling connection to practical real-world (deep) model training. To summarize, this paper does not show enough evidence that the revealed pattern can improve existing optimization problems theoretically or practically.
2. While the study emphasizes that its unsupervised learning focus is distinct from prior work, the observed double descent patterns align with those documented in supervised learning. This difference in setup, while noted, does not seem to substantively alter the properties of the optimization process.
3. The empirical verification of double descent with respect to model size (via specific layer dimensions), sample size, and training epochs lacks novelty.


$\textbf{Weakness in evaluation}$

A substantial portion of the paper, particularly Section 4, examines double descent behavior under conditions of customized noise, domain shift, and anomalies in the training data. However, these customized conditions appear to have been selected without a clear rationale, and the resulting findings are unsurprising.

**Questions:**

1. Could you elaborate on the fundamental differences between supervised and unsupervised learning that warrant separate investigations into the double descent phenomenon in each scenario?

2. What insights or practical applications do you anticipate arising from studying double descent under different data conditions, such as Gaussian noise and domain shift?

3. Section 5 is somewhat unclear. The conclusion drawn here—that over-parameterized models improve training performance under varying conditions—seems straightforward. Additionally, how does this finding relate to the double descent phenomenon?

---

> ### Comment · Reviewer_NdzY · 2024-11-15
> **In correct review**
>
> I apologize. When submitting, I put the review of a different paper. Here is the correct review:
>
> **Summary**:
> This paper investigates the double descent phenomenon about three key training parameters, training epochs, model size, and data sample size, within an unsupervised learning framework. To illustrate the phenomenon, consider training epochs: initially, test error decreases as the number of training epochs increases. However, after reaching a certain threshold, test error begins to increase. After a period of error rising, the test error decreases again, known as, double descent. Additionally, the authors conduct experiments using synthetic data and small-scope models to empirically examine the impact of customized noise, domain shifts, and anomalies on double descent behavior. Finally, the study assesses how model size influences domain adaptation and anomaly detection performance.
>
> **Soundness**: 2: fair
> **Presentation**: 2: fair
> **Contribution**: 2: fair
>
> **Strengths**
> 1. This work provides empirical evidence that the double descent phenomenon exists in various training parameters under unsupervised learning settings, and the non-linear relationship observed between test error and training parameters may be of interest.
>
> 2. Unlike previous studies, to verify the influential factor model size, this work proposes utilizing specific layers in different dimensions to simulate the changing of model size. This modification is well-suited for experiments with small-scope neural networks by avoiding introducing extra variables.
>
> **Weaknesses**
>
> $\textbf{Weakness in method motivation and novelty}$
>
> 1. This work needs more motivation to study the phenomenon of double descent. Particularly, this work does not provide possible theoretical improvement or help better understand the unsupervised learning problems. Additionally, the patterns identified lack a compelling connection to practical real-world (deep) model training. To summarize, this paper does not show enough evidence that the revealed pattern can improve existing optimization problems theoretically or practically.
>
>
> 2. While the study emphasizes that its unsupervised learning focus is distinct from prior work, the observed double descent patterns align with those documented in supervised learning. This difference in setup, while noted, does not seem to substantively alter the properties of the optimization process.
>
> 3. The empirical verification of double descent with respect to model size (via specific layer dimensions), sample size, and training epochs lacks novelty.
>
>
> $\textbf{Weakness in evaluation}$
>
> A substantial portion of the paper, particularly Section 4, examines double descent behavior under conditions of customized noise, domain shift, and anomalies in the training data. However, these customized conditions appear to have been selected without a clear rationale, and the resulting findings are unsurprising.
>
> **Questions**
>
> 1. Could you elaborate on the fundamental differences between supervised and unsupervised learning that warrant separate investigations into the double descent phenomenon in each scenario?
>
> 2. What insights or practical applications do you anticipate arising from studying double descent under different data conditions, such as Gaussian noise and domain shift?
>
> 3. Section 5 is somewhat unclear. The conclusion drawn here—that over-parameterized models improve training performance under varying conditions—seems straightforward. Additionally, how does this finding relate to the double descent phenomenon?
>
> **Flag For Ethics Review**
>
> No ethics review needed.
>
> **Rating**
>
> 3: reject, not good enough
>
> **Confidence**
> 4: You are confident in your assessment, but not absolutely certain. It is unlikely, but not impossible, that you did not understand some parts of the submission or that you are unfamiliar with some pieces of related work.

---

> ### Author Response · Authors · 2024-11-20
> **Response to the reviewer**
>
> We thank the reviewer for recognizing the key strengths of our paper, particularly the empirical evidence demonstrating the existence of the double descent phenomenon across various training parameters in unsupervised settings and the relationship between test error and these parameters. Additionally, we appreciate the acknowledgment of our analysis on model size, which highlights the influence of both the bottleneck and hidden layers on the double descent curve.
>
> **Weakness 1 - motivation:**
>
> Our primary motivation lies in extending the understanding of double descent—previously studied mainly in supervised settings—to unsupervised learning scenarios, which are critical in many real-world applications. While the phenomenon of double descent has provided substantial insights into the generalization capabilities of over-parameterized models in supervised tasks, similar insights for unsupervised learning remain limited. Our work aims to address this gap by demonstrating how double descent behaviors occur with under-complete autoencoders across various challenging scenarios, including noise, domain shifts, and anomalies [1, 2, 3]. These scenarios are highly relevant in practical deep-learning tasks involving real-world data, particularly where labeled data is unavailable.
>
> Regarding practical impact, our findings provide important empirical insights into optimizing unsupervised model training. For instance, we demonstrate that over-parameterized autoencoders can achieve lower test loss and better generalization, even in the presence of substantial noise or domain shift, which directly affects model selection in tasks like anomaly detection and domain adaptation. Our results suggest that over-parameterized models can mitigate the adverse effects of noise and improve feature extraction, thereby enhancing downstream performance in anomaly detection and domain adaptation.
>
> While our work is primarily empirical (similar to the impactful works in [4, 5], it lays a foundation for future theoretical exploration by characterizing double descent patterns in unsupervised contexts. The identified behaviors provide an empirical basis for further theoretical analysis, particularly regarding how model capacity interacts with noise and domain shifts in unsupervised learning. Thus, our study not only contributes practical insights but also sets the stage for future theoretical advancements that can deepen our understanding of optimization and generalization in unsupervised models.
>
> [1] Vincent, Pascal, et al. "Extracting and composing robust features with denoising autoencoders." Proceedings of the 25th international conference on Machine learning. 2008.
>
> [2] Zhou, Chong, and Randy C. Paffenroth. "Anomaly detection with robust deep autoencoders." Proceedings of the 23rd ACM SIGKDD international conference on knowledge discovery and data mining. 2017.
>
> [3] Deng, Jun, et al. "Autoencoder-based unsupervised domain adaptation for speech emotion recognition." IEEE Signal Processing Letters 21.9 (2014): 1068-1072.
>
> [4] Mikhail Belkin, Daniel Hsu, Siyuan Ma, and Soumik Mandal. Reconciling modern machine learning practice and the classical bias–variance trade-off. Proceedings of the National Academy of Sciences, 116(32):15849–15854, 2019.
>
> [5] Preetum Nakkiran, Gal Kaplun, Yamini Bansal, Tristan Yang, Boaz Barak, and Ilya Sutskever. Deep double descent: Where bigger models and more data hurt. Journal of Statistical Mechanics: Theory and Experiment, 2021(12):124003, 2021.

---

> ### Author Response · Authors · 2024-11-20
> **Response cont.**
>
> **Weakness 2 - This difference in setup, while noted, does not seem to substantively alter the properties of the optimization process:**
>
> Our work differs from prior research in several important ways. Firstly, while most existing studies focus on supervised learning, our research is entirely unsupervised, specifically utilizing autoencoders. In supervised learning, noise is typically added to the labels or targets, commonly referred to as "label noise." Mathematically, this can be represented as perturbing the labels $y$, such that $y' = y + \epsilon$, where $\epsilon$ is random noise. In contrast, our unsupervised setup involves adding noise directly to the input data, represented by $x' = x + \epsilon$, where $x$ is the input and $\epsilon$ is noise. This model distinction means that the optimization process in our work centers on reconstructing noisy input data rather than minimizing a loss with noisy target labels.
>
> Secondly, we investigate scenarios that extend beyond standard noise settings, such as the presence of anomalies in the training data. This scenario, characterized by introducing anomalous samples alongside regular data, is fundamentally different from typical supervised noise settings and, to our knowledge, has not been explored in the context of double descent in supervised learning. The impact of such anomalies on the double descent behavior, particularly in an unsupervised context, provides new insights that are absent from prior work.
>
> Finally, we identify the unique role of the latent dimension in determining the emergence of double descent in our unsupervised autoencoder setting. Unlike prior work that focuses on model width or parameter count, we show that the size of the latent space significantly influences the balance between learning meaningful representations and overfitting noise. Smaller latent dimensions constrain model capacity, leading to higher test loss, especially with noise or anomalies. As the latent dimension increases, the model transitions to an over-parameterized regime, resulting in the characteristic double descent curve, where the test loss first rises due to noise memorization and then decreases as generalization improves. This unique focus on the latent dimension highlights its critical impact on performance in unsupervised learning, offering insights that are particularly useful for tasks like anomaly detection and domain adaptation.
>
> **Weakness 3 - The empirical verification of double descent lacks novelty:**
>
> We believe that empirically verifying double descent concerning model size, number of epochs, and number of samples is novel for four main reasons:
>
> 1. [1] argues that double descent does not occur in unsupervised learning using AEs. Our work disproves this claim by demonstrating multiple instances of double and triple descent curves across various contamination setups. Additionally, our paper addresses and resolves the debate on the existence of double descent in unsupervised learning, as discussed in the related work section. The works in the related work section show that this phenomenon is of interest in unsupervised learning.
>
> 2. To expand on the second weakness: while model-wise, epoch-wise, and sample-wise double descent have been observed in supervised learning, one of our key contributions is demonstrating that this phenomenon also occurs beyond the supervised regime—a point that was previously uncertain.
>
> 3. We decompose the model's size into two interconnected components, the hidden and bottleneck layers, demonstrating how each layer influences the test loss and the critical regime. Figure 2 displays interesting results: as the bottleneck dimension increases, a larger hidden layer size is required to overfit (i.e., the critical regime shifts to larger hidden dimensions as the bottleneck layer size gets bigger). Intuitively, one might expect the opposite: that the critical regime would shift downwards as the bottleneck size increases, allowing an interpolating model to be achieved with a smaller hidden layer size while keeping the overall model size the same. This is a richer evaluation than supervised examples (see [2]) since we display that certain layers of an autoencoder are very influential for the reconstruction results and the double descent curve.
>
> 4. We show that anomalies also influence the double descent curve, an observation that was not observed in supervised learning.
>
> [1] Lupidi, Alisia, Yonatan Gideoni, and Dulhan Jayalath. "Does Double Descent Occur in Self-Supervised Learning?." arXiv preprint arXiv:2307.07872 (2023).
>
> [2] Preetum Nakkiran, Gal Kaplun, Yamini Bansal, Tristan Yang, Boaz Barak, and Ilya Sutskever. Deep double descent: Where bigger models and more data hurt. Journal of Statistical Mechanics: Theory and Experiment, 2021(12):124003, 2021.

---

> > ### Comment · Reviewer_NdzY · 2024-11-26
> > **Thanks for the rebuttal**
> >
> > After carefully reviewing all the feedback, I will keep my score for the following reasons:
> > 1). The reasons warrant separate investigations into the double descent phenomenon in supervised and unsupervised learning scenarios are still unclear. The distinctions between those two scenarios, explicit supervision, and implicit supervision, are self-evident but do not directly address the question: Do explicit supervision and implicit supervision truly imply different phenomena or mechanisms for double descent?
> > 2). Regarding experiment design, the customized conditions appear to have been selected without a clear rationale.

---

> ### Author Response · Authors · 2024-11-20
> **Response cont. 2**
>
> **Weakness in evaluation:**
>
> Our study introduces four contamination setups: sample noise, feature noise, domain shift, and anomalies, each with varying noise levels, domain shifts, and SNRs.
>
> Section 4 presents results using synthetic or semi-synthetic datasets, while Section 5 focuses on real-world data. These experiments are both relevant and novel, as real-world data often contains contamination, and our findings illustrate how this impacts test loss.
>
> For instance, in many cases, datasets are inherently noisy, and we identify a critical regime that should be avoided. Additionally, in anomaly detection applications, it is not straightforward to receive pre-sorted data divided into clean and anomalous samples. We find that the ROC-AUC and test loss of the clean samples display non-monotonic behavior and double descent, respectively, as the training data is contaminated with anomalies.
>
> Additionally, the double descent phenomenon itself is particularly complex in the unsupervised setting, especially when the training set contains more noisy samples than clean ones. Intuitively, one might expect the test loss to keep increasing as bigger models memorize the noise instead of learning the underlying structure of the signal.
>
> **Q1 - elaborate on the fundamental differences between supervised and unsupervised learning:**
>
> The fundamental differences between supervised and unsupervised learning indeed justify separate investigations into the double descent phenomenon, primarily due to the nature of the tasks, the types of assumptions involved, and how noise and data structure influence learning behavior in each regime.
>
> 1. Task Objectives: Supervised learning tasks typically involve predicting discrete or continuous labels, such as classification or regression. The model aims to map input features to a known target output. In contrast, unsupervised learning, such as autoencoders (AEs), aims to learn meaningful representations of the input data without labeled targets, often focusing on tasks like reconstruction. This difference fundamentally alters how errors and model performance are evaluated. In supervised learning, performance is measured in terms of predictive accuracy, while in unsupervised learning, metrics such as reconstruction loss are used.
>
> 2. Data Assumptions: Supervised learning commonly introduces noise as "label noise," where the labels may be flipped or altered, affecting the model's predictive capability. Unsupervised learning deals with noise in the input data itself, influencing the model's ability to learn useful representations. For instance, adding noise to an AEr’s training data can lead to memorization of noise rather than learning the underlying structure. This distinction changes the conditions under which double descent manifests, as unsupervised models need to balance learning useful features against overfitting to noise, while supervised models focus on the trade-off between bias and variance in predicting labels.
>
> 3. Model Complexity: Supervised and unsupervised learning differ in their underlying model assumptions. Supervised models often use assumptions about the relationship between input features and labels, such as linear separability. In unsupervised settings, models like autoencoders must capture complex data structures without explicit supervision, making them more sensitive to changes in data distribution and model complexity. As a result, the double descent behavior in unsupervised models may arise under different conditions, such as varying noise levels, domain shifts, or data anomalies.
>
> **Q2 - insights or practical applications arising from studying double descent under different data conditions:**
>
> Studying double descent under different data conditions, such as Gaussian noise and domain shift, can lead to several valuable insights and practical applications:
>
> 1. Noise Resilience: Understanding how models behave under different noise levels can help in designing more robust models that maintain performance despite noisy data.
>
> 2. Generalization: Insights into how over-parameterized models can generalize better even when trained on noisy data, challenging traditional views on overfitting.
>
> 3. Double Descent Curves: Identifying the points of double descent can help in selecting the optimal model complexity that balances bias and variance effectively.
>
> 4. Guidelines for Practitioners: Providing guidelines on selecting model sizes and training regimes based on the expected noise levels and domain shifts in the data.
>
> 5. Denoising: Insights from studying noise can lead to better denoising techniques in image and signal processing.
>
> 6. Transfer Learning: Improved understanding of how models handle domain shifts can enhance transfer learning techniques, making models more adaptable to new, unseen data.
>
> 7. Anomaly detection: Understanding the behavior of model capacity under different anomaly levels in the training data can help enhance anomaly detection capabilities.

---

> ### Author Response · Authors · 2024-11-20
> **Response cont. 3**
>
> **Q3 - The conclusion drawn here—that over-parameterized models improve training performance under varying conditions:**
>
> If you are referring to: “Over-parameterized models also lead to reduced train and test losses, resulting in improved reconstruction of both the source and target data”, we have modified these lines to be: ”They also lead to reduced test loss, resulting in improved reconstruction of the target data.”, lines 465-466. You are right, the improvement in train loss as the model’s size rises is straightforward.

---

> ### Author Response · Authors · 2024-11-24
> **Notification**
>
> Dear reviewer, we appreciate the time and effort you have dedicated to reviewing our paper. As the discussion period is limited, we kindly request that the reviewer evaluate the new information provided in the rebuttal. We are eager to improve our paper and resolve all the concerns raised by the reviewer. If there are any remaining concerns that have not been addressed, we would be happy to provide further explanations.

---

### Official Review · Reviewer_S8BP · 2024-11-04

**Soundness:** 3
**Presentation:** 2
**Contribution:** 2
**Rating:** 5
**Confidence:** 3

**Summary:**

The paper investigates the double descent phenomenon in the unsupervised setting. The experiments on synthetic datasets explore the relation between double descent and four variables: sample noise, feature noise, domain shift, and anomalies. Moreover, some observations remain for real-world datasets.

**Strengths:**

The paper investigates double descent in the unsupervised setting, which did not receive attention as much as supervised learning or self-supervised learning.

The paper has the following contributions:
1. The paper shows how the four different variables (sample noise, feature noise, domain shift, and anomalies) affect the double descent.
2. The paper connects observations of the synthetic and real-world datasets.
3. It is interesting to see that although there are four different variables, many of them share the same intuition (the magnitude or portion of the noise controls the vertical height and the horizontal position).

**Weaknesses:**

The paper is not smooth to read. Some details are not mentioned or explained. For example, providing some details regarding the under-complete AEs could help the readers understand why the experiments should be based on under-complete AEs. Besides, the experiment sections contain many figures and they could be rearranged for a smoother reading experience.

As for the experiment section specifically:
1. How is Fig.2 different from the traditional figure of test loss against the number of parameters? They seem to convey the same information. Does it mean the relation between the two in unsupervised learning is the same as supervised learning?
2. Some curves are missing. For example, in Fig.3a various levels of sample noises are shown, but many of them are missing in Fig.3b. Could you provide all the curves?
3. I recommend the authors put a paragraph or a table to summarize the relation between the double descent and the four variables as they share very similar behavior and intuition. It will be easier for the reader to access the experiment outcome.
4. For the paper to be more informative, the author could compare the double descent behaviors in unsupervised learning against the supervised learning regime. Could you conclude if there is any different behavior in the unsupervised setting than the ones observed in the other settings?

**Questions:**

Please refer to the weakness section for the questions.

---

> ### Author Response · Authors · 2024-11-20
> **Response to the reviewer**
>
> We appreciate the reviewer for highlighting the main strengths of our paper, especially our discovery of double descent in unsupervised learning and the analysis of various contamination setups, including sample noise, feature noise, domain shifts, and anomalies. We are grateful for the recognition that although there are four contamination setups, most of them share the same intuition regarding the height and horizontal position of the test loss and critical regime, as well as the connections we established between synthetic and real-world data observations.
>
> **Weaknesses - the paper is not smooth to read:**
>
> Due to the numerous experiments involving different datasets and contamination setups, the main paper contains many figures, which may impact the flow of reading. We have made efforts to reduce the number of figures, including moving all training loss figures to the appendix section. For your request, we have arranged some of the figures so the paper will be easier to follow. Figures 7-13 in our revised manuscript were modified to be at the top and bottom of the paper. **Details regarding under-complete AEs:** The rationale for using under-complete AEs (instead of over-complete ones) is explained in lines 203-205 in the “Results” section.
>
> Additionally, in response to your suggestion, we have included more details explaining our choice of under-complete AEs in lines 70-72 of the attached manuscript, adjacent to the first mention of the term “under-complete AEs”. Appendix A, Table 2 details the model sizes, including the bottleneck layer size, hidden layer size for MLPs, and the number of channels for CNNs. Additionally, Figures 17 and 18 illustrate the MLP and CNN models and their parameter counts.
>
> **Q1 - How is Fig.2 different from the traditional figure of test loss against the number of parameters? Does it mean the relation between the two in unsupervised learning is the same as supervised learning?**
>
> In contrast to supervised learning, the bottleneck layer plays a significant role in many applications that use AEs (i.e., dimensionality reduction, denoising, compression, anomaly detection, feature selection, and domain adaptation). This motivates breaking down the model’s size into the bottleneck layer and other layers.
>
> Figure 2 differs from traditional test loss versus number of parameters plots because it also presents the variation in the bottleneck size. In all the other figures in our paper, the bottleneck size remains constant, and we vary the hidden layer size. The bottleneck and hidden layer sizes play a crucial role in the outcomes of an autoencoder, which is why we broke down the model’s size into these parts. For example, suppose we have two models, one with a larger bottleneck layer and smaller hidden layers compared to the second, and both of them result in the same number of parameters. The first model might achieve better reconstruction results since the bottleneck layer is more significant, leading to less information loss in the latent space, making it easier for the decoder to extract a better output.
>
> Looking only at the model’s size and not breaking it down to these two factors will result in noisy test loss curves as a function of the number of parameters since two models with similar capacities result in completely different test loss outcomes. Figure 2 shows that it’s not enough just to select the model’s capacity; we also need to consider the hidden and bottleneck layer size, as they influence the position of the critical regime. In all the figures in which the bottleneck layer size is constant, we reveal relations between the test loss and the model’s size that are similar to the supervised regime.
>
> Moreover, Figure 2 displays interesting results: as the bottleneck dimension increases, a larger hidden layer size is required to overfit (i.e., the critical regime shifts to larger hidden dimensions as the bottleneck layer size gets bigger). Intuitively, one might expect the opposite: that the critical regime would shift downwards as the bottleneck size increases, allowing an interpolating model to be achieved with a smaller hidden layer size while keeping the overall model size the same. This is a richer evaluation than supervised examples (see [1]) since we display that certain layers of an autoencoder are very influential for the reconstruction results and the double descent curve.
>
> [1] Preetum Nakkiran, Gal Kaplun, Yamini Bansal, Tristan Yang, Boaz Barak, and Ilya Sutskever. Deep double descent: Where bigger models and more data hurt. Journal of Statistical Mechanics: Theory and Experiment, 2021(12):124003, 2021.

---

> ### Author Response · Authors · 2024-11-20
> **Response cont.**
>
> **Q2 - Some curves are missing:**
>
> In Figure 3(b), we only showed noise levels ranging from 30-60% since the figure was too cluttered. We followed your suggestion and also added 70-90% noise levels. Lines 246-248 discuss the final ascent phenomenon, which we also observed in single-cell RNA data for cases with 0-20% sample noise. These lower noise levels are presented in Appendix E.4, Figure 54(b), resulting in all the noise levels also presented in Figure 3(a). Please note that the single-cell RNA data is acquired from real-world experiments that already include natural noise, so even if we did not add any synthetic noise (presented by 0%), the data is still affected by noise.
>
> **Q3 - A paragraph to summarize the relation between the double descent and the four variables:**
>
> That is a great suggestion. In the revised paper, we have added a paragraph concluding the results of all the contamination setups in lines 344-349. As contamination setups become more severe, such as higher noise levels for sample and feature noise, significant domain shifts, many anomalies, or low SNR, the double descent phenomenon becomes more pronounced, and the test loss increases. In some instances, these noise levels also cause the critical regime to shift to the right.
>
> **Q4 - compare the double descent behaviors in unsupervised learning against the supervised learning regime:**
>
> Since double descent was initially observed in supervised learning, you are right that a comparison between our unsupervised results and those from supervised learning is essential. We have added Table 1 in Appendix A (page 17) in the revised paper, summarizing which contamination setups have been studied for their impact on the double descent curve in both unsupervised and supervised learning. Our findings indicate that the overall results are similar across the setups examined in both learning regimes [1, 2, 3]. The primary difference we observed is the necessity of low SNR values for the unsupervised regime.
>
> Although the double descent phenomenon appears in both supervised and unsupervised regimes, we would like to clarify that their tasks differ significantly. For instance, most research on double descent in supervised learning focuses on regression or classification tasks, which come with different assumptions, such as using linear models and distinct methods of introducing noise, like label noise (where some labels are randomly altered). In contrast, our work addresses unsupervised autoencoder tasks with a different set of assumptions, such as learning meaningful representations in a latent space, and the noise is added to the samples. Therefore, comparisons between the behaviors in supervised and unsupervised regimes should be made cautiously.
>
> [1] Nakkiran, Preetum, et al. "Deep double descent: Where bigger models and more data hurt." Journal of Statistical Mechanics: Theory and Experiment 2021.12 (2021): 124003.
>
> [2] Li, Zhu, Weijie J. Su, and Dino Sejdinovic. "Benign overfitting and noisy features." Journal of the American Statistical Association 118.544 (2023): 2876-2888.
>
> [3] Tripuraneni, Nilesh, Ben Adlam, and Jeffrey Pennington. "Overparameterization improves robustness to covariate shift in high dimensions." Advances in Neural Information Processing Systems 34 (2021): 13883-13897.

---

> > ### Comment · Reviewer_S8BP · 2024-11-30
> >
> > Thank you for your response. I would like to maintain my score. The paper has the potential to explore the double descent deeper in the unsupervised setting and provide more insights. Although both supervised and unsupervised learning have different tasks, supervised learning also requires a good representation for the downstream tasks. Thus, we cannot conclude their differences/similarities merely by considering differences in tasks. I recommend exploring the necessary condition of double descent further (e.g., why low SNR leads to double descent and how it connects unsupervised learning to supervised learning if it does?).

---

> ### Author Response · Authors · 2024-11-24
> **Notification**
>
> Dear reviewer, we appreciate the time and effort you have dedicated to reviewing our paper. As the discussion period is limited, we kindly request that the reviewer evaluate the new information provided in the rebuttal. We are eager to improve our paper and resolve all the concerns raised by the reviewer. If there are any remaining concerns that have not been addressed, we would be happy to provide further explanations.

---

### Author Response · Authors · 2024-11-20
**General Response**

We would like to express our gratitude to the Area Chair and the reviewers for their time and effort in evaluating our paper. We appreciate their constructive comments and their recognition of the strengths of our work. In particular, we are grateful for their acknowledgment of our discovery of the double descent phenomenon in fully unsupervised autoencoders (AEs). This finding emerged from a comprehensive empirical study that offers insightful observations based on both real-world and synthetic data.

While some reviewers noted that this result is "unsurprising," we want to emphasize that this problem has been previously studied by several authors who demonstrated a negative outcome. For instance, some argued that double descent does not occur in our setting [1], and got accepted for the workshop on High-Dimensional Learning Dynamics at ICML 2023: https://arxiv.org/abs/2307.07872 - check the comments.

To the best of our knowledge, our work is the first to identify model-wise, sample-wise, and epoch-wise double descent in a completely unsupervised setting. Additionally, we are the first to observe double descent in a real-world setting rather than in academic datasets with induced label noise. We also examine the role of various data contaminations and demonstrate that the latent and hidden dimensions in autoencoders (AEs) contribute independently to the double descent phenomenon. We believe these findings represent significant contributions to the field.

We are thankful for recognizing the impact of various contamination setups on the double descent curve and for acknowledging how our empirical study inspires future research on model complexity in unsupervised learning.

[1] Lupidi, Alisia, Yonatan Gideoni, and Dulhan Jayalath. "Does Double Descent Occur in Self-Supervised Learning?." arXiv preprint arXiv:2307.07872 (2023).

---

### Meta-Review · Area_Chair_YWot · 2024-12-08

**Metareview:**

This paper studies the double descent phenomenon in unsupervised learning by using undercomplete autoencoders. The paper empirically shows that double or multiple descents occur at the model-wise, epoch-wise, and sample-wise levels. The paper examined sample noise, feature noise, domain shift, and anomalies. The paper also shows how autoencoders can be used for detecting anomalies and mitigating the domain shift between datasets.

The main strength of the paper is to empirically show that double descent phenomenon can be shown with autoencoder experiments, which is a different result from a previous paper. Reviewers acknowledge the contribution of the extensive experiments with various variables (such as sample noise, feature noise, etc.), and mentions that the paper is easy to follow.

Although the results are novel (in authors' own words: "To the best of our knowledge, our work is the first to identify model-wise, sample-wise, and epoch-wise double descent in a completely unsupervised setting."), several reviewers pointed out that the paper does not offer new perspectives/insights to the literature, and that the takeaway is not exactly clear. I agree with the authors that the results can contribute a lot to the research community even if it is "not surprising", but in such cases, the contribution can be enhanced further by having theoretical discussions/analysis for the autoencoder case, which was suggested by multiple reviewers. A reviewer provided helpful reference to work along this line.

The paper was a borderline case. If the paper is unfortunately not accepted to the conference, I encourage the authors to refine their work based on the reviewers suggestions and consider resubmitting it to this or another conference or journal in the future.

**Additional Comments On Reviewer Discussion:**

A rebuttal was provided to each review. All reviewers read the rebuttal, and provided a response, and expressed that their concerns were not fully addressed (mostly for the reasons mentioned in the metareview above). One of the reviewer decreased the score without mentioning the reasoning, so I did not weigh in the final decreased score.

---

### Decision · Program_Chairs · 2025-01-22

Reject